# Cells of the human intestinal tract mapped across space and time

Rasa Elmentaite[1], Natsuhiko Kumasaka[1], Kenny Roberts[1], Aaron Fleming[2], Emma Dann[1], Hamish W. King[3], Vitalii Kleshchevnikov[1], Monika Dabrowska[1], Sophie Pritchard[1], Liam Bolt[1], Sara F. Vieira[1], Lira Mamanova[1], Ni Huang[1], Francesca Perrone[4], Issac Goh Kai'En[5], Steven N. Lisgo[5], Matilda Katan[6], Steven Leonard[1], Thomas R. W. Oliver[1,7], C. Elizabeth Hook[7], Komal Nayak[4], Lia S. Campos[1], Cecilia Domínguez Conde[1], Emily Stephenson[5], Justin Engelbert[5], Rachel A. Botting[5], Krzysztof Polanski[1], Stijn van Dongen[1], Minal Patel[1], Michael D. Morgan[8,9], John C. Marioni[1,8,9], Omer Ali Bayraktar[1], Kerstin B. Meyer[1], Xiaoling He[10], Roger A. Barker[10], Holm H. Uhlig[11,12,13], Krishnaa T. Mahbubani[14], Kourosh Saeb-Parsy[14], Matthias Zilbauer[4,15,16], Menna R. Clatworthy[1,2], Muzlifah Haniffa[1,5,17], Kylie R. James[1,19 ✉] & Sarah A. Teichmann[1,18 ✉]

The cellular landscape of the human intestinal tract is dynamic throughout life, developing in utero and changing in response to functional requirements and environmental exposures. Here, to comprehensively map cell lineages, we use single-cell RNA sequencing and antigen receptor analysis of almost half a million cells from up to 5 anatomical regions in the developing and up to 11 distinct anatomical regions in the healthy paediatric and adult human gut. This reveals the existence of transcriptionally distinct BEST4 epithelial cells throughout the human intestinal tract. Furthermore, we implicate IgG sensing as a function of intestinal tuft cells. We describe neural cell populations in the developing enteric nervous system, and predict cell-type-specific expression of genes associated with Hirschsprung's disease. Finally, using a systems approach, we identify key cell players that drive the formation of secondary lymphoid tissue in early human development. We show that these programs are adopted in inflammatory bowel disease to recruit and retain immune cells at the site of inflammation. This catalogue of intestinal cells will provide new insights into cellular programs in development, homeostasis and disease.

Intestinal tract physiology relies on the integrated contribution of multiple cell lineages, the relative abundance and cell networking of which fluctuate from embryonic development to adulthood. Further complexity is added because the intestinal tract is formed of distinct anatomical regions that develop at different rates and carry out diverse roles in digestion, nutrient absorption, metabolism and immune regulation.

The analysis of rare fetal tissues has resolved the formation of villi–crypt structures and the seeding of immune cells into the gut environment[1–3]. Similarly, our understanding of the cellular landscape of the adult gut is benefiting from single-cell technologies. Regional differences in immune-cell activation and microbiome composition in the

healthy human colon have previously been reported[4]. Studies that compare inflammatory bowel disease samples to healthy tissues have enabled the identification of disease-relevant stromal[5,6], tissue-resident CD8 T cell[7–9] populations and correlation between cellular response and clinical treatment[10]. Although extensive work has been carried out to profile the intestinal tract at single-cell resolution (Supplementary Table 1), a holistic analysis of the gut through space (anatomical location) and time (lifespan) is lacking. Building such a developmental roadmap would be invaluable for the scientific community[11].

Here we create a single-cell census of the healthy human gut, encompassing around 428,000 cells from the small and the large intestines as

[1]Wellcome Sanger Institute, Wellcome Genome Campus, Hinxton, UK. [2]Molecular Immunity Unit, Department of Medicine, University of Cambridge, MRC Laboratory of Molecular Biology, Cambridge, UK. [3]Centre for Immunobiology, Blizard Institute, Queen Mary University of London, London, UK. [4]Department of Paediatrics, University of Cambridge, Cambridge, UK. [5]Biosciences Institute, Faculty of Medical Sciences, Newcastle University, Newcastle upon Tyne, UK. [6]Structural and Molecular Biology, Division of Biosciences, University College London, London, UK. [7]Department of Histopathology, Cambridge University Hospitals NHS Foundation Trust, Cambridge, UK. [8]European Molecular Biology Laboratory, European Bioinformatics Institute, Wellcome Genome Campus, Cambridge, UK. [9]Cancer Research UK Cambridge Institute, University of Cambridge, Cambridge, UK. [10]John van Geest Centre for Brain Repair, Department of Clinical Neurosciences and Wellcome-MRC Cambridge Stem Cell Institute, University of Cambridge, Cambridge, UK. [11]Translational Gastroenterology Unit, John Radcliffe Hospital, University of Oxford, Oxford, UK. [12]Department of Paediatrics, University of Oxford, Oxford, UK. [13]NIHR Oxford Biomedical Research Centre, Oxford, UK. [14]Department of Surgery, University of Cambridge and NIHR Cambridge Biomedical Research Centre, Cambridge, UK. [15]Department of Paediatric Gastroenterology, Hepatology and Nutrition, Cambridge University Hospitals Trust, Cambridge, UK. [16]Wellcome-MRC Cambridge Stem Cell Institute, Anne McLaren Laboratory, University of Cambridge, Cambridge, UK. [17]Department of Dermatology and NIHR Newcastle Biomedical Research Centre, Newcastle Hospitals NHS Foundation Trust, Newcastle upon Tyne, UK. [18]Theory of Condensed Matter Group, Cavendish Laboratory/ Department of Physics, University of Cambridge, Cambridge, UK. [19]Present address: Garvan Institute of Medical Research, The Kinghorn Cancer Centre, Darlinghurst, New South Wales, Australia. ✉e-mail: k.james@garvan.org.au; st9@sanger.ac.uk

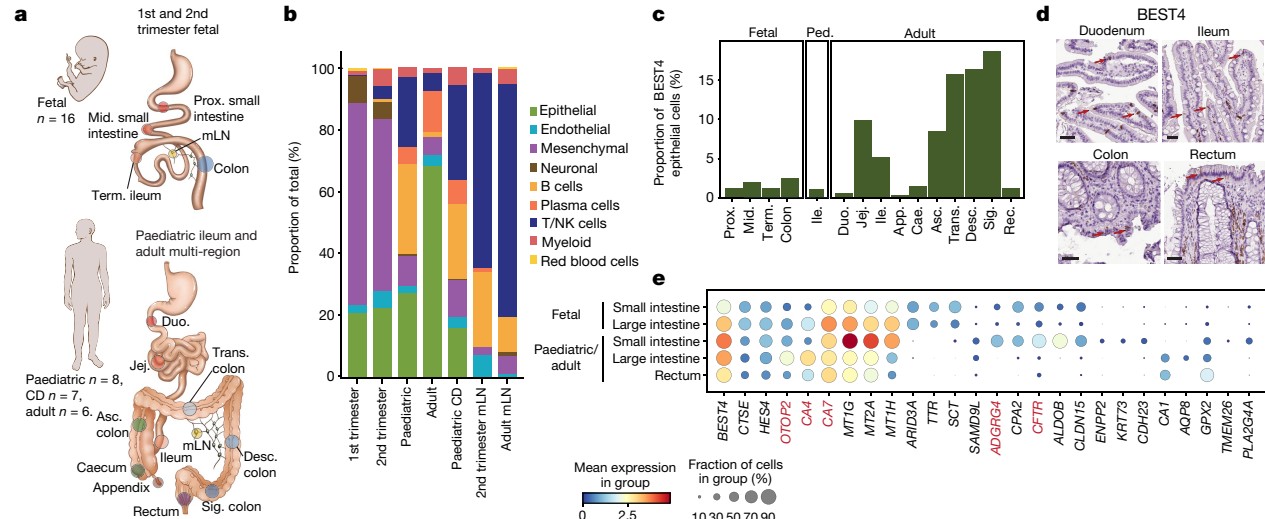

**Fig. 1 | Intestinal cellular census throughout life. a**, Schematic of human gut tissue sampling. Number of donors sampled for scRNA-seq is given. Mid., middle; prox., proximal; term., terminal; mLN, mesenteric lymph node; jej., jejunum; duo., duodenum; trans., transverse; asc., ascending; desc., descending; sig., sigmoid. **b**, Relative proportions of cell lineages at each developmental stage. NK, natural killer; CD, inflammatory bowel disease. **c**, Proportions of *BEST4*-expressing enterocytes among epithelial cells in scRNA-seq data of each tissue region and at each developmental stage. Ile., ileum; app., appendix; cae., caecum; rec., rectum. **d**, Expression of BEST4 in histological sections, from https://proteinatlas.org[48] (Supplementary Table 8, $n = 2$ biologically independent samples for each region). Scale bars, 50 μm. **e**, Dot plot with relative expression of selected genes within BEST4 epithelial cells from different locations and ages. Key genes are highlighted in red, and the full Milo analysis can be found in Extended Data Fig. 3d.

well as associated lymph nodes during in utero development, childhood and adulthood.

## Integrated map of human intestinal cells

To investigate cellular dynamics across the intestinal tract, we performed single-cell RNA sequencing (scRNA-seq) on distinct tissue regions of second-trimester (12–17 post-conception weeks (PCW)) and adult (29–69 years) intestines and draining mesenteric lymph nodes (mLN) (Fig. 1a, Extended Data Fig. 1a). Additionally, we integrated results from the scRNA-seq analysis of tissues from first-trimester (6–11 PCW) intestine, paediatric Crohn's disease and healthy ileum[1].

The dataset comprised more than 428,000 high-quality cells (Extended Data Fig. 1b, Supplementary Table 2). Leiden clustering and marker-gene analysis revealed major clusters of epithelial, mesenchymal, endothelial, immune, neural and erythroid cells (Fig. 1b, Extended Data Fig. 1c). Fetal gut samples were enriched for mesenchymal and neural cells, with increased abundance of immune cells from the second trimester onwards in gut and mLN (Fig. 1b, Extended Data Fig. 1d–f). Further sub-clustering of the cellular lineages enabled the identification of 133 cell types and states with specific transcriptional identities (Extended Data Fig. 2a, Supplementary Tables 3–7).

BEST4 epithelial cells, which have been previously observed in human small and large intestines[6,12,13], varied in abundance between intestinal regions (Fig. 1c, d). Using differential cell-type abundance analysis[14], we identified their region-specific expression signatures (Fig. 1e, Extended Data Fig. 3a–d). Notably, small-intestinal BEST4 cells were marked by high expression of the gene *CFTR*, which encodes a chloride channel and is mutated in cystic fibrosis (Fig. 1e); such high expression was also observed at the protein level (Extended Data Fig. 3e). In this staining and in previous work, BEST4 cells were in close proximity to cells that resembled goblet cells[12] (Extended Data Fig. 3f). Our analysis highlights a possible role of BEST4 enterocytes of the small intestine in aiding mucus production by goblet cells and biosynthesis of acids, in contrast to the functions of colonic BEST4 cells in the metabolism of small molecules (Extended Data Figs. 3g, 4a, b).

## Diversity of intestinal epithelial cells

In the epithelial compartment, secretory cells consisted of goblet, tuft, Paneth and microfold cells, as well as precursor states. Absorptive and goblet cells showed regional separation at all life stages (Fig. 2a, b, Extended Data Figs. 5a–c, 6a, b). Given that the gut epithelium represents an entry point for SARS-CoV-2[15], we also report that both *ACE2* and *TMPRSS2*—which encodes transmembrane serine protease 2—were expressed by enterocytes in early development (Fig. 2c, Extended Data Fig. 6c).

Subclustering of enteroendocrine cells (EECs) revealed *NEUROG3*-expressing precursor cells enriched in the first-trimester fetal gut, and multiple mature subsets resembling populations described in intestinal organoids[16] (Fig. 2d, Extended Data Fig. 6d). Although neuropeptide W (encoded by *NPW*) is known to stimulate food intake[17] and is broadly expressed by EECs[16,18], we found a subpopulation of *NPW*-expressing enterochromaffin cells that are also specific for *PRAC1* and *RXFP4* (Extended Data Fig. 4d). We delineated genes involved in the differentiation of *NEUROG3* precursors to enterochromaffin cells, including recently described genes (marked by arrows) such as *FEV*[19,20] (Fig. 2e, Extended Data Fig. 6e, f).

Notably, among the top differentially expressed genes in tuft cells was *PLCG2* (Extended Data Fig. 6g, h), a phospholipase that is typically associated with haematopoietic cells. To explore the relevance of *PLCG2* in tuft cells, we screened for the expression of upstream receptors (Fig. 2f, Extended Data Fig. 6i). FCGR2A, which is activated in response to IgG and expressed by selected epithelial cells in immunized mice[21], was specifically expressed by approximately 2.75% of tuft cells (Fig. 2f). We confirmed the expression of the protein FCGR3 (the mouse orthologue of human FCGR2A) by approximately 5% of small intestinal tuft cells in mice (Fig. 2g, Extended Data Fig. 6j). Receptor tyrosine kinases were expressed across tuft cells and other epithelial cell types. Because these are known to be mainly linked to PLCG1 activation, whether they are also responsible for PLCG2 activation in tuft cells is difficult to delineate (Fig. 2h). *PLCG2* expression by tuft cells was at higher levels than in B and myeloid lineages, and was confirmed in both in vivo and in vitro models (Extended Data Fig. 7a–f) and—together

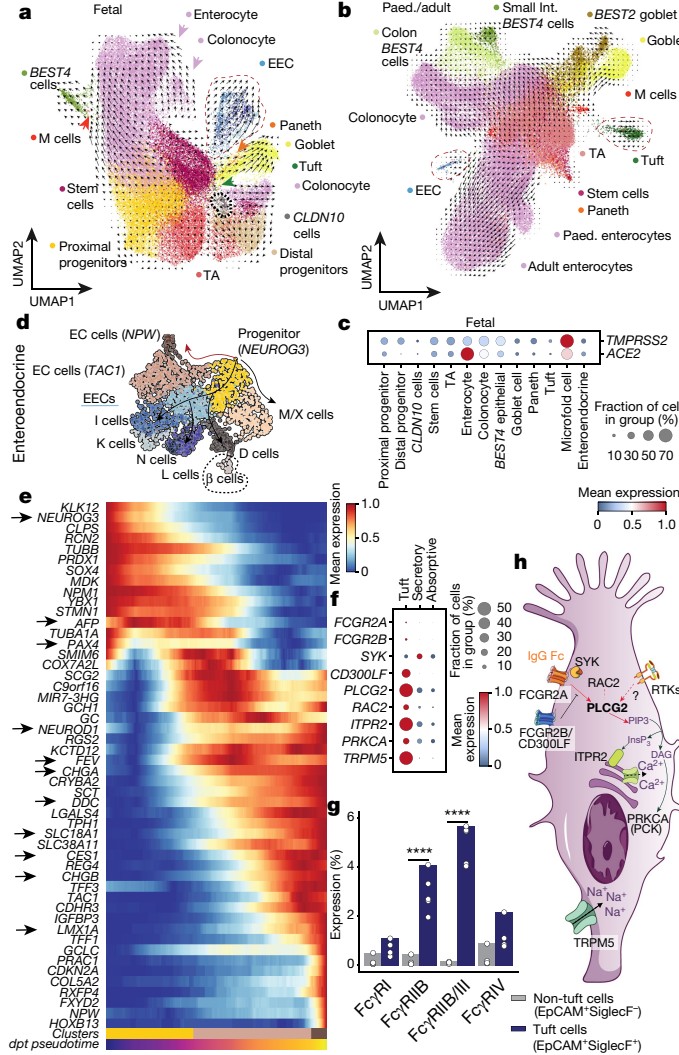

**Fig. 2 | Epithelial cells and FCGR2A signalling in tuft cells. a, b,** Uniform manifold approximation and projection (UMAP) of fetal (**a**) and postnatal (**b**) epithelial cell types. Key cell types are circled with a dashed line and arrows depict paths of differentiation towards secretory and absorptive enterocytes as determined by scVelo. M cells, microfold cells; TA, transit-amplifying. **c,** Dot plot of *TMPRSS2* and *ACE2* expression in epithelial cells in the fetal intestine as in **a. d,** UMAP of enteroendocrine (EEC) and enterochromaffin (EC) cell subsets. Arrows depict summarised scVelo differentiation trajectories. *C9orf16* is also known as *BBLN*. **e,** Heat map of genes that change along the differentiation trajectory from *NEUROG3*-expressing progenitors to enterochromaffin cells (red arrow in **d**). Arrows indicate genes that have known associations with enterochromaffin cell differentiation. **f,** Dot plot with expression of molecules upstream or downstream of the PLCG2 pathway in tuft cells and pooled absorptive (TA and enterocytes) and secretory (Paneth, goblet and EEC) cells. **g,** Per cent expression of Fcγ receptor by SiglecF⁺EpCAM⁺ and SiglecF⁻EpCAM⁺ cells in wild-type mice determined by flow cytometry (individual points represent biological replicates; $n = 4$). Using a two-way ANOVA we observe significant interaction between Fcγ receptor expression ($F(3, 39) = 42.29$, $P = 3.05 \times 10^{-11}$). Post-hoc analysis showed significant differences between non-Tuft epithelial cells and Tuft cells for FcγRIIB (mean difference = 2.80, 95% CI [1.80, 3.81], $P < 0.0001$) and FcγRIIB/III (mean difference = 4.88, 95% CI [3.88, 5.89], $P < 0.0001$) expression. ****$P_{adj} < 0.0001$ values corrected with Tukey's test for multiple comparisons. **h,** Schematic of proposed signalling pathways in tuft cells. RTK, receptor tyrosine kinases.

with downstream signalling mediators including *RAC2*, *ITPR2*, *PRKCA* and *TRPM5* (Fig. 2f, h)—suggested the ability of tuft cells to respond to immune-cell signalling.

## Development of the enteric nervous system

Next we investigated the differentiation of neural cells from enteric neural crest cell (ENCC) progenitors (Fig. 3a, b) that were present in the dataset from 6.5 PCW (Extended Data Fig. 8a). ENCCs balance proliferation and differentiation into glia and neurons, while maintaining a progenitor reserve. To capture discrete processes of ENCC differentiation, we analysed early (6–11 PCW) and late (12–17 PCW) development separately (Extended Data Fig. 8b). In early development, ENCCs differentiated primarily to neurons via neuroblasts, giving rise to two distinct branches: branch A (*ETV1*) and branch B (*BNC2*) (Fig. 3a, Extended Data Figs. 8c, 9a–d), as has been observed in mice[22]. At this stage, branch A further differentiated to inhibitory motor neurons (iMN, resembling ENC8–ENC9[22]) and two subsets that had characteristics of intrinsic primary afferent neurons (IPANs) or interneurons (resembling ENC12[22]) with similarity to cells observed in the human fetal gut[3] (Extended Data Fig. 8d). Branch B further differentiated to immature excitatory motor neuron (eMN) subsets (branches B1 and B2, resembling ENC1–ENC3[22]) (Fig. 3a).

At later development, branch A differentiated into *NEUROD6*-expressing interneurons (resembling ENC10[22]), whereas branch B differentiated into IPANs (Fig. 3b, Extended Data Figs. 8c, 9a–d) similar to previously described adult human IPAN A cells[23]. We visualize opposing expression of *SCGN* (branch A1) and *GRP* (branch A2 and A3) and *BNC2* (branch B1 and B2) in the developing and adult human myenteric plexus (Fig. 3c, Extended Data Fig. 8e). The expression of transcription factors, such as ETV1, was previously validated in situ in a complementary resource of the human gut[24].

Although differentiated neurons were abundant at 6–11 PCW, glial cells were enriched at later development. Three types of enteric neural and a subset of differentiating glia (*COL20A1*) were present at 12–17 PCW (Fig. 3b, Extended Data Fig. 9a–d). Colonic glia 1 cells expressed posterior HOX genes and *TFAP2B*, which suggests that they originated in the sacrum or trunk (Extended Data Fig. 8g). We visualized *BMP8B*-expressing cells in the myenteric plexus, whereas *DHH*-expressing cells were found both in the mesentery and the myenteric plexus (Fig. 3d, Extended Data Fig. 8f).

To identify neural cells involved in Hirschsprung's disease (HSCR), we screened for the expression of known HSCR-associated genes[25–27]. The majority of HSCR-associated genes were expressed across multiple differentiating populations with varying intensity (Fig. 3e), and varied between neuron branches A and B. For example, *RET* was highly expressed by branch A, but not by branch B, neurons. Notably, *ZEB2* and *EDNRB* were more highly expressed across colonic glia and neuroblast subsets compared to equivalent small intestinal subsets (Fig. 3e). Any differences in expression between regions might also be due to the developmental lag of the large intestines. In addition, key ligands that are implicated in HSCR—including *GDNF*, *NRTN* and *EDN3*—were primarily expressed by mesothelium, smooth muscle cells and interstitial cells of Cajal (ICC) (Extended Data Fig. 8h).

## Formation of secondary lymphoid organs

Gut-associated lymphoid tissues and mLN are key sites of gut immune surveillance. We observed mLN emergence at around 12 PCW, with structures disectable from 15 PCW (Extended Data Fig. 10a)—consistent with previous observations[28]. Interactions between mesenchymal and endothelial lymphoid tissue organizers (mLTo and eLTo, respectively) and lymphoid tissue inducers (LTi) are central to initiating the formation of secondary lymphoid organs[29]. To better understand this process in humans, we assessed our dataset for the key cell types involved.

Sub-clustering of fetal and adult T and innate lymphoid cells revealed three clusters that matched published characteristics of LTi cells (Fig. 4a, b). This included high expression of *RORC*, *KIT*, *TNF*,

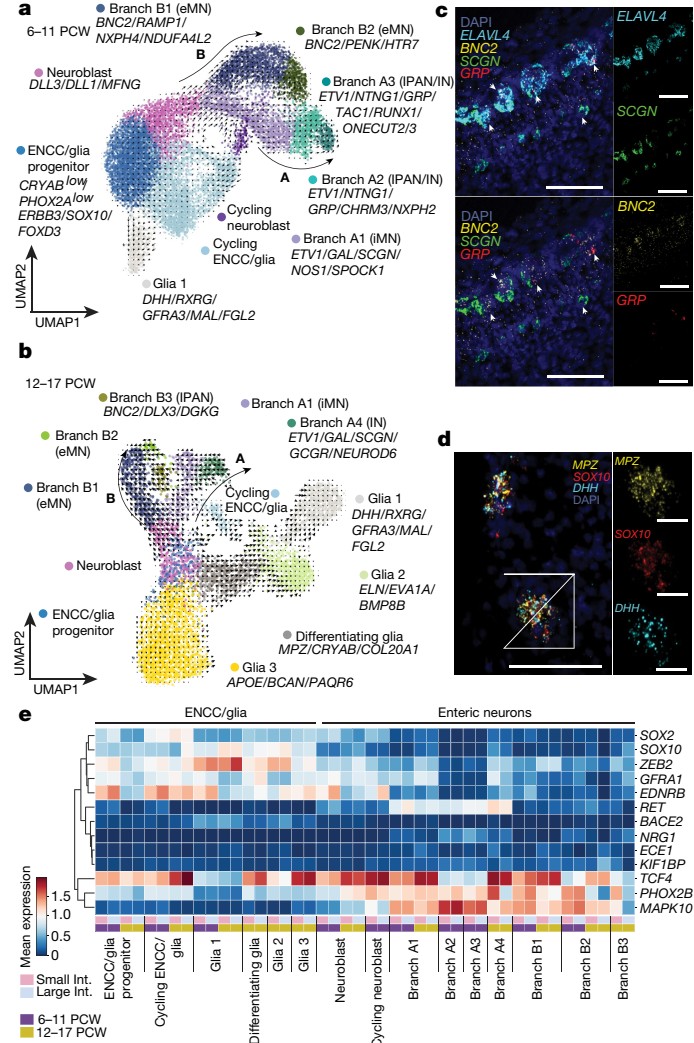

**Fig. 3 | Cells of the developing enteric nervous system. a**, **b**, UMAP of enteric neural crest cells (ENCC) and their progeny at 6–11 (**a**) and 12–17 (**b**) PCW. Overlaid arrows depict scVelo trajectories, with major neuronal branches shown as A and B. Marker genes for populations are listed. Branch A2 and A3 subsets were not observed at 12–17 PCW, possibly because they were outnumbered by the glial populations. **c**, **d**, Multiplex smFISH staining of *SCGN* branch A1, *GRP* branch A2/A3 and *BNC2* branch B1/2 developing *ELAVL4* neurons (arrows, n = 2) in the 15 PCW ileum (scale bars, 100 μm) (**c**) and glia 1 (*DHH*, *MPZ*, *SOX10*) cells in the mesentery (scale bars: main, 100 μm; expansion, 30 μm. n = 2 (**d**)). *n* represents the number of biological replicates across regions. **e**, Heat map showing the mean expression of genes associated with HSCR across intestinal regions and developmental stages. iMN, inhibitory motor neuron; IPAN, primary afferent neurons; IN, interneurons, int., intestine.

*LTA*, *LTB*, *IL7R* and *ITGB7* (beta chain of gut-specific α4β7 integrin) as well as the absence of productive αβ TCR (Extended Data Figs. 10b–d, 11a–c). Innate lymphoid cell progenitors (ILCPs) were transcriptionally comparable to fetal liver ILCPs[30] (Extended Data Fig. 10e), and were found both in fetal mLN and in embryonic gut, whereas NCR⁺ and NCR⁻ type 3 innate lymphoid cells (ILC3s) were expanded across gut regions throughout 6–17 PCW (Extended Data Fig. 10f). This suggests that LTi-like ILC3 subsets are expanded during the development of gut-associated lymphoid tissues, but not during the development of mLN. Single-molecule fluorescence in situ hybridization (smFISH) staining identified all three LTi-like subtypes (Extended Data Fig. 10g) and placed *CXCR5 and RORC*-expressing LTi cells adjacent to *CXCL13*-expressing LTo cells in proximal gut mucosa (Fig. 4c), supporting

the concept of congregation of these cells in the developing gut. These observations—together with the expression of genes that encode key chemokines (*CXCR5*, *CCR7* and *CCR6*) and RNA velocity analysis (Extended Data Fig. 10h)—suggests that ILCPs are the first LTi-like cells in the developing gut, and represent a progenitor state to ILC3s.

We observed arterial, venous, capillary and lymphatic endothelial cells (LECs) (Extended Data Figs. 2a, 12a, b). LECs separated into six clusters (labelled LEC1–LEC6; Extended Data Fig. 12c). LEC2 cells expressed *TNFRSF9*, *THY1*, *CXCL5* and *CCL20*—as described in human lymph nodes[31]—as well as targets of the NF-κB pathway and adhesion molecules including *MADCAM1*, *VCAM1* and *SELE*, suggesting their involvement in lymphocyte trafficking (Extended Data Fig. 12d, e). We confirmed that the presence of high endothelial venule structures was required for lymphocyte entry and for proximity of *PROX1*⁺ vessels to *CXCL13*⁺ mLTo and *RORC*⁺ LTi cells (Extended Data Fig. 12f, g).

Within the stromal compartment, we identified subtypes of myofibroblast, smooth muscle cells, pericyte, interstitial cells of Cajal, mesothelium and populations resembling stromal cells previously described in the colon[5] (labelled as stromal 1–4) (Fig. 4d, Extended Data Fig. 2a). We further identified fibroblast populations typically defined in mouse lymph nodes[32], including T reticular cells and follicular dendritic cells (Extended Data Fig. 13a). In the prenatal intestine and mLN, we observed a stromal population—marked by the expression of *CCL19*, *CCL21* and *CXCL13* as well as adhesion and NF-κB pathway molecules (Extended Data Fig. 13b–f)—that resembled the mLTo described in mouse lymph nodes[33]. We further determined cell–cell interactions that governed early leukocyte recruitment across LEC2, mLTo and LTi-like cells (Extended Data Fig. 13g, h), and differences in B cell activation status between fetal and adult samples (Extended Data Fig. 14a–l).

To visualize the recruitment of naive immune cell subsets to activated mLTo, we used cell2location[34] to perform spatial mapping of single-cell transcriptomes to 10x Genomics Visium spatial zones in 17 PCW fetal ileum (Fig. 4e, Methods). In this analysis, we captured tissue zones with expression of mLTo marker genes (*CCL19*, *CCL21* and *CXCL13*) (Extended Data Fig. 15a–c) that were likely to correspond to developing secondary lymphoid organs. We found that LEC2, mLTo and LTi-like subsets mapped to the same tissue zones as naive immune subsets (for example, *SELL* CD4 T cells, T regulatory cells, immature B cells; tissue zone or fact_11 (Extended Data Fig. 15c)). M cells characteristic of secondary lymphoid organs were present in the adjacent zone (tissue zone or fact_5 (Extended Data Fig. 15c)).

Following our observations, we compared programs of lymphoid organogenesis with the formation of ectopic lymphoid structures that is observed in patients with Crohn's disease[35]. We found that ILC3s in tissues from patients with Crohn's disease matched fetal NCR⁺ ILC3s with more than 60% probability, whereas T reticular cells and stromal 4 cells from patients with Crohn's disease transcriptionally reassembled fetal mLTo (Extended Data Fig. 15d, e) and were expanded in four out of seven Crohn's disease samples (Extended Data Fig. 13b). Finally, from a genome-wide association study across cell types in our dataset, we calculated the enrichment score for genes associated with Crohn's disease and ulcerative colitis (with a false-discovery rate of 10%; Methods). Adult ILC3s and fetal ILCPs and NCR⁺ ILC3s were among the top cells that were enriched for the expression of genes associated with Crohn's disease (Fig. 4f, Extended Data Fig. 15f).

## Discussion

Here we present an integrated dataset of more than 428,000 single cells from multiple anatomical regions of the human gut throughout life. This dataset can be browsed at https://gutcellatlas.org.

We found that *FCGR2A*, which encodes a receptor activated by the Fc fragment of IgG upstream of *PLCG2*, is expressed by a subset of tuft cells. In the context of development, IgG has been shown to traverse the placenta, and so could provide a potential route for tuft cell activation

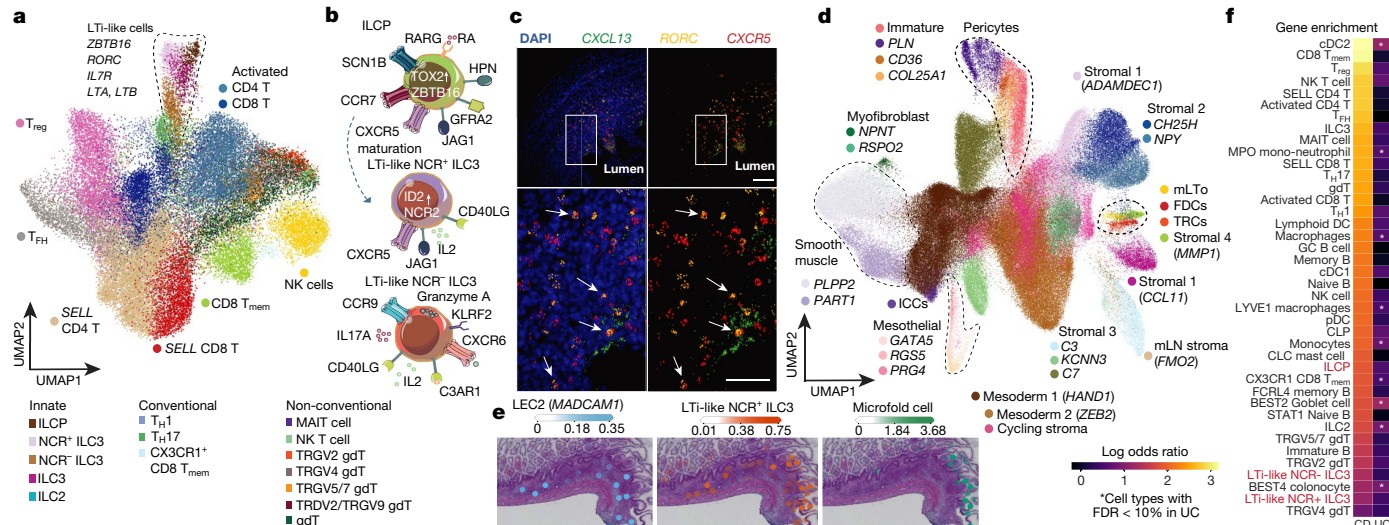

**Fig. 4 | Lymphoid tissue organogenesis programs adopted in Crohn's disease. a**, UMAP of T and innate cells in scRNA-seq data across development. Dotted line denotes LTi-like cells and listed are characteristic genes. **b**, Schematic showing expression signatures of identified LTi-like states. **c**, Multiplex smFISH of 15 PCW ileum showing proximity of *RORC CXCR5*-expressing LTi-like cells to *CXCL13*-expressing mLTo cells (*n* = 2 biological replicates across regions). Arrows highlight cells of interest. Scale bars: main, 100 μm; expansion, 50 μm. **d**, UMAP of stromal cell types across development. The dotted line highlights key lineages. **e**, Spatial mapping of cell types from

the scRNA-seq data to spatial transcriptomics data of 17 PCW terminal ileum using cell2location[34]. Estimated abundance for cell types (colour intensity) across locations (dots) is overlaid on a histology image for LEC2 (left), LTi-like ILC3 (middle) and microfold (right) cells. **f**, Heat map showing top cell types across fetal, paediatric (healthy and Crohn's disease) and adult data that are enriched for gene expression associated with either Crohn's disease or ulcerative colitis (UC). All cell types listed are FDR < 10% for Crohn's disease. Asterisks denote cell types with FDR < 10% for ulcerative colitis. CLP, common lymphoid progenitor.

in utero. Two missense variants of *PLCG2* have been linked to aberrant B cell responses in early-onset inflammatory bowel disease[36,37] and primary immune deficiency[38]. Here we show increased expression of FcγRIIB (human *FCGR2B*)—which encodes an inhibitory receptor—by tuft cells in a mouse model of colitis, suggesting another possible involvement for this pathway in inflammatory bowel disease through tuft cells. Overall, these data suggest a potentially impactful immune-sensing role for intestinal tuft cells, and a topic for further investigation in future.

Immune sensing in the intestines also occurs in secondary lymphoid organs, and although there are substantial differences in the size and location of secondary lymphoid organs between species[39], our understanding of the formation of these structures is mostly derived from animal models[29]. Single-cell studies have begun to elucidate the diversity and activation of immune cells in the developing human gut[3,40,41]. Here we identify cell types in the developing intestines that have transcriptional signatures and signalling pathways matching LTi, mLTo and eLTo (LEC2) cells. In vitro studies have shown the ability of *RORC*-expressing CD56[+]CD127[+]IL17A[+] cells[42] and ILC3s[43] to activate mesenchymal cells. This leads us to propose that all three LTi-like subsets defined in this study act in the initiation of secondary lymphoid organs, but probably at different stages of the process. Future studies using complementary approaches will shed light on these cell states and their tissue architecture. Notably, we also observe two specialized fibroblast populations in paediatric Crohn's disease—follicular dendritic cells and T reticular cells—similar to cells described in mouse lymph nodes[32], Peyer's patches[44–47] and those expanded in patients with ulcerative colitis[5] as well as patients with Crohn's disease[1,10].

Overall, our work provides clarity on the complex interplay between intestinal cell types throughout time and space, and has potential implications for disease and for the engineering of in vitro systems.

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

## Methods

No statistical methods were used to predetermine sample size. The experiments were not randomized and the investigators were not blinded to allocation during experiments and outcome assessment.

### Patient samples

The research complies with all relevant ethical regulations and guidelines. Informed consent was obtained from all human participants and includes consent to publish photographs included in figures. Maternal consent and fetal gut samples were obtained through Newcastle upon Tyne Hospitals NHS Foundation Trust, the Human Developmental Biology Resource (HDBR, https://www.hdbr.org), or through Addenbrooke's Hospital, Cambridge in collaboration with R. A. Barker. Procurement and study of fetal samples were approved by REC18/NE/0290- IRAS project ID: 250012, North East - Newcastle & North Tyneside 1 Research Ethics Committee and REC-96/085, East of England - Cambridge Central Research Ethics Committee.

Paediatric patient material used in intestinal organoid culture was obtained with informed consent from either parents and/or patients using age-appropriate consent and assent forms as part of the ethically approved research study (REC-12/EE/0482, NRES Committee East of England, Hertfordshire and REC-17/EE/0265- IRAS project ID: 222907, East of England - Cambridge South Research Ethics Committee).

Human adult tissue was obtained by the Cambridge Biorepository of Translational Medicine from deceased transplant organ donors after ethical approval (reference 15/EE/0152, East of England - Cambridge South Research Ethics Committee) and informed consent from the donor families. Details of the ages and genders of donors are included as Supplementary Table 2. Samples were collected from 11 distinct locations, including the duodenum (two locations, DUO1 and DUO2, which were pooled in the analysis), jejunum (JEJ), ileum (two locations, ILE1 and ILE2, which were pooled in the analysis), appendix (APD), caecum (CAE), ascending colon (ACL), transverse colon (TCL), descending colon (DCL), sigmoid colon (SCL), rectum (REC) and mesenteric lymph nodes (mLN). Fresh mucosal intestinal tissue and lymph nodes from the intestinal mesentery were excised within 1 h of circulatory arrest; intestinal tissue was preserved in University of Wisconsin organ-preservation solution (Belzer UW Cold Storage Solution; Bridge to Life) and mLN were stored in saline at 4 °C until processing. Tissue dissociation was conducted within 2 h of tissue retrieval.

### Mouse samples

C57BL/6 mice were obtained from Jackson Laboratories and maintained in specific-pathogen-free conditions at a Home-Office-approved facility at the University of Cambridge. Female mice aged 10–14 weeks were used; the numbers of mice used are included in the relevant figure legends. All procedures were carried out in accordance with ethical guidelines with the United Kingdom Animals (Scientific Procedures) Act of 1986 and approved by The University of Cambridge Animal Welfare and Ethical Review Body.

### Isolation of intestinal cells from fetal tissue

The fetal gut mesentery was cut to lengthen out the tissue and the gut was dissected into proximal ileum (PIL), middle ileum (MIL), terminal ileum (TIL), colon or large intestine (LI) and appendix. Samples, except appendix, were washed twice with Hanks Buffered Saline Solution (HBSS; Sigma-Aldrich, 55021C) and minced into pieces using a scalpel. The samples were incubated in 2 ml HBSS solution containing 0.21 mg ml$^{-1}$ Liberase TL (Roche, 5401020001) or DH (Roche, 5401089001) and 70 U ml$^{-1}$ hyaluronidase (Merck, 385931-25KU) for up to 50 min at 37 °C, shaking every 5 min, and homogenized every 15 min using a pipette. The single cells were passed through a 40–100 μm sieve and spun down at 400$g$ at 4 °C for 10 min. Red cell lysis solution (eBioscience10X RBC Lysis Buffer (Multi-species)) was used according to the manufacturer's guidelines

to remove red blood cells, and the remaining cells were collected in FACS buffer (1% (v/v) FBS in PBS) by centrifugation at 400$g$ at 4 °C for 5 min. All gut region samples (except mLN) proceeded to enrichment by magnetic-activated cell sorting (MACS).

### Isolation of intestinal cells from adult tissue

Adult tissue sections were weighed before being washed in cold D-PBS (Gibco, 14190094) and diced with a scalpel. Samples were dissociated in 1–2 ml of digestion mix (D-PBS, 250 μg ml$^{-1}$ Liberase TL (Roche, 5401020001), 0.1 mg ml$^{-1}$ DNaseI (Sigma, 11284932001)) or DH (Roche, 5401089001) and 70 U ml$^{-1}$ hyaluronidase (Merck, 385931-25KU) for up to 30 min at 37 °C. Digested lysates were then passed through a 70 μm cell strainer, followed by 10 ml of neutralization medium (RPMI 1640 medium with HEPES (Gibco, 42401042), 20% (v/v) FBS (Sigma, 25200-056)). The samples were then centrifuged at 700$g$ for 5 min at 4 °C. Cells were resuspended in 1 ml 0.04% (w/v) BSA in D-PBS and counted using a NucleoCounter NC-200 and Via1-Cassette (ChemoMetec). All gut region samples (except mLN) proceeded to MACS enrichment.

### MACS enrichment

Samples with a cell yield of greater than 500,000 were enriched by MACS. Dissociated cells were centrifuged for 5 min at 300$g$ at 4 °C and resuspended in 80 μl chilled MACS buffer (D-PBS, 0.5% (w/v) BSA (Sigma-Aldrich, A7906-10G), 2 mM EDTA (Thermo Fisher, 15575020)) with 20 μl CD45 MicroBeads (Miltenyi Biotech, 130-045-801) and incubated for 15 min at 4 °C. Cells were washed with 2 ml MACS buffer and centrifuged as above and resuspended in 500 μl MACS buffer. Cells were passed through a pre-wetted MS column (Miltenyi Biotech, 130-042-201) on a QuadroMACS Magnetic Cell Separator (Miltenyi Biotech) followed by four rounds of 500 μl of MACS buffer. Flow-through was collected as the CD45$^-$ fraction. The column was removed from the magnet and the CD45$^+$ fraction was eluted with 1 ml of MACS buffer. CD45$^-$ and CD45$^+$ fractions were centrifuged as above and resuspended in 0.5–1 ml of 0.04% (w/v) BSA in D-PBS. Cell count and viability was determined using a NucleoCounter NC-200 and Via1-Cassette (ChemoMetec) or haemocytometer and resuspended in 0.04% (w/v) BSA in D-PBS. Fetal CD45$^+$ and CD45$^-$ fractions were combined at a 1:1 ratio.

### Organoid culture

Intestinal organoids from paediatric patients were cultured in Matrigel (Corning). During organoid culture, the medium was replaced every 48–72 h. Organoids were passaged using mechanical disruption with a P1000 pipette and re-seeded in fresh growth-factor-reduced Matrigel (Corning). When comparing culture media, multiple wells were seeded from a single dissociated sample. The organoids were then allowed to grow for 5 days followed by 24 h treatment with recombinant human protein TNF (H8916, Sigma Aldrich) at 40 ng ml$^{-1}$ or IFNγ (PHC4031, Life Technologies) at 20 ng ml$^{-1}$. Organoids were in vitro differentiated for 4 days by culturing in a differentiation medium[49] and then collected for RNA extraction. Bright-field images were taken using an EVOS FL system (Life Technologies).

Processing for single-cell sequencing analysis was performed by removing the organoids from Matrigel at passage 3–4 using incubation with Cell Recovery Solution at 4 °C for 20 min, pelleting the cells, and re-suspending in TrypLE enzyme solution (Thermo Fisher) for incubation at 37 °C for 10 min. Cells were pelleted again and re-suspended in DMEM/F12.

### RNA extraction, reverse transcription and quantitative PCR

Total RNA was extracted with the GenElute Mammalian Total RNA Miniprep kit (Sigma) and 1 μg of RNA was reverse-transcribed using the QuantiTect Reverse Transcription Kit (Qiagen). Complementary DNA corresponding to 5 ng RNA was used for real-time PCR performed by QuantiFast SYBR Green PCR Master Mix (Qiagen) on the 7500 Fast real-time PCR system, 7500 software v.2.0.6 (Applied Biosciences by Thermo Fisher Scientific). Primers for the specific target

amplification were: *FCGR2A* fwd 5′-CCATCCCACAAGCAAACCACAG-3′ and rev 5′-AGCAGTCGCAATGACCACAG-3′; *FCGR2B* fwd 5′-CCATCCC ACAAGCAAACCACAG-3′ and rev 5′-ACAGCAATCCCAGTGACCACAG-3′; *PLCG2* fwd 5′-AACCCATCTGACCCTCCTCTTG-3′ and rev 5′-AGACTGCTGTT CCCTGTGTTCC-3′; *POU2F3* fwd 5′-TTCAGCCAGACCACCATCTCAC-3′ and rev 5′-GGACTCTGCATCATTCAGCCAC-3′; *MUC2* fwd 5′-GATTCGA AGTGAAGAGCAAG-3′ and rev 5′-CACTTGGAGGAATAAACTGG-3′; *LGR5* fwd 5′-CTCCCAGGTCTGGTGTGTTG-3′ and rev 5′-GAGGTCTAGGTAGG AGGTGAAG-3′. Relative quantifications were normalized against GAPDH and calculated applying the $\Delta\Delta C_t$ method.

Statistical analysis was performed using GraphPad Prism 7 software (GraphPad Software) by multiple *t*-test analysis.

## 10x Genomics Chromium GEX library preparation and sequencing

MACS-enriched and total cell fractions were loaded for droplet-based scRNA-seq according to the manufacturer's protocol for the Chromium Single Cell 5′ gene expression v.2 (10x Genomics) to obtain 8,000–10,000 cells per reaction. Library preparation was carried out according to the manufacturer's protocol. Pools of 16 libraries were sequenced across both lanes of an Illumina NovaSeq 6000 S2 flow cell with 50 bp paired-end reads.

Intestinal organoids were prepared using Chromium Single Cell 3′ gene expression v.2 (10x Genomics) to obtain 8,000 cells per reaction. Intestinal organoid cDNA libraries were sequenced on a single lane of an Illumina HiSeq 4000 with 50 bp paired-end reads.

## V(D)J sample preparation

10x Genomics V(D)J libraries were generated from the 5′ 10x Genomics Chromium complementary DNA (cDNA) libraries as detailed in the manufacturer's protocol. B cell receptor (BCR) and T cell receptor (TCR) libraries for relevant samples were pooled and sequenced on a single lane of an Illumina HiSeq 4000 with 150 bp paired-end reads.

## Plate-based Smart-seq2

Plate-based scRNA-seq was performed with the NEBNext Single Cell/ Low Input RNA Library Prep Kit for Illumina (E6420L; New England Biolabs). Total cell fractions from dissociated gut sections of donors BRC2033–2034, F67, F72, F78 were snap-frozen in 10% (v/v) DMSO in 90% (v/v) BSA. Cells were thawed rapidly in a 37 °C water bath and diluted slowly with a pre-warmed FACS buffer (2% (v/v) FBS in D-PBS). Cells were pelleted by centrifugation for 5 min at 300*g*, washed with 300 μl of D-PBS and pelleted as before. Cells were resuspended in 100 μl of Zombie Aqua Fixable Viability Kit (1:200 dilution; 423101) and incubated at room temperature for 15 min. Cells were washed with 2 ml of FACS buffer followed by 300 μl of FACS buffer and resuspended in a total of 100 μl of Brilliant Violet 650 mouse anti-human CD45 (dilution 1:200; BioLegend, 304043), Alexa Fluor 700 mouse anti-human CD4 (dilution 1:200; BioLegend, 300526) and APC-H7 mouse anti-human CD19 (dilution 1:200; BD Biosciences, 560727) and incubated for 20 min in the dark at room temperature. Cells were washed twice with 300 μl of FACS buffer. Single, live, CD45[+] cells were sorted by fluorescence-activated cell sorting (FACS) into wells of a 384-well plate (0030128508; Eppendorf) containing 2 μl of 1× NEB Next Cell Lysis Buffer (New England Biolabs). FACS sorting was performed with a BD Influx sorter (BD Biosciences) with the indexing setting enabled. Plates were sealed and spun at 100*g* for 1 min then immediately frozen on dry ice and stored at −80 °C. cDNA and sequencing library generation was performed in an automated manner on the Bravo NGS Workstation (Agilent Technologies)[4]. The purified pool was quantified on an Agilent Bioanalyzer (Agilent Technologies) and sequenced on one lane of an Illumina HiSeq 4000. Raw reads were aligned to the human transcriptome v.GRCh38-3.0.0 using STAR aligner (v.2.5.1b).

## Pre-processing of 10x Genomics scRNA-seq data

10x Genomics scRNA-seq gene expression raw sequencing data were processed using the CellRanger software v.3.0.0–v.3.0.2 and the 10X human transcriptome GRCh38-3.0.0 as the reference. The 10x Genomics V(D)J Ig heavy and light chains were processed using cellranger vdj v.3.1.0 and the reference cellranger-vdj-GRCh38-alts-ensembl-3.1.0 with default settings.

## scRNA-seq quality control and processing of 10x sequencing data

Pandas (v.1.1.2), NumPy (v.0.25.2), Anndata (v.0.6.19), ScanPy (v.1.4) and Python (v.3) were used to pool single-cell counts and for downstream analyses. Single-cell transcript counts for fetal and adult samples were handled separately to control for anticipated differences in cell expression and sample quality. For each run, we apply the SoupX algorithm[50] with default parameters and function adjustCounts() to remove ambient mRNA from the count matrix. Cells for each dataset were filtered for more than 500 genes and less than 50% mitochondrial reads, and genes were filtered for expression in more than 3 cells. A Scrublet (v.0.2.1) score cut-off of 0.25 was applied to assist with doublet exclusion. Additional doublet exclusion was performed throughout downstream processing based on unexpected co-expression of canonical markers such as CD3D (component of the TCR) and EpCAM. Gene expression for each cell was normalized and log-transformed. Cell cycle score was calculated using the expression of 97 cell cycle genes listed in ref. [51]. Cell cycle genes were then removed for initial clustering. Cell cycle score, the percentage of mitochondrial reads and unique molecular identifiers (UMIs) were regressed before scaling the data.

## Cell-type annotation

Batch correction of fetal and adult datasets was performed with bbknn (v.1.3.9, neighbours=2-3, metric='euclidean', n_pcs=30-50, batch_key='donor_id' or 'batch'). Dimensionality reduction and Leiden clustering (resolution 0.3–1.5) was carried out and cell lineages were annotated on the basis of algorithmically defined marker gene expression for each cluster (sc.tl.rank_genes_groups, method='wilcoxon'). Cell lineages were then subclustered and batch correction and Leiden clustering were repeated for annotation of cell types and states. Annotated fetal and adult datasets were then merged and annotations adjusted for concordance. A brief description of cell-type annotation for each lineage is provided below.

*Epithelial lineage cells* (*EPCAM*-positive) shared between fetal, paediatric and adult datasets were stem cells (*LGR5*, *ASCL2*, *SMOC2*, *RGMB*, *OLFM4*), Paneth (*DEFA5*, *DEFA6*, *REG3A*), transit-amplifying (TA; *MKI67*, *TOP2A*, *PCNA*), goblet cells (*CLCA1*, *SPDEF*, *FCGBP*, *ZG16*, *MUC2*), BEST4 enterocytes (*BEST4*, *OTOP2*, *CA7*), enterocytes (*RBP2*, *ANPEP*, *FABP2*) and colonocytes (*CA2*, *SLC26A2*, *FABP1*), enteroendocrine cells (*CHGA*, *CHGB*, *NEUROD1*), Microfold cells (*SPIB*, *CCL20*, *GP2*), Tuft cells (*POU2F3*, *LRMP* (also known as *IRAG2*), *TRPM5*), BEST2 goblet cells were observed in adult colonic samples (Extended Data Fig. 5). Amongst the genes enriched in BEST2 goblet cells were Kallikreins *KLK15* and *KLK3*, as well as protease inhibitors *WFDC2* and *WFDC3*. Among fetal epithelial cells, 43 large intestinal goblet cells expressed *BEST2*. Fetal BEST2-expressing cells clustered together with small intestinal goblet cells, possibly due to the small number of these cells present in the data. Enteroendocrine cells were further sub-clustered and annotated based on key hormones expressed (M/X cells (*MLN/GHRL*), D cells (*SST*), β cells (*INS*, possibly from developing pancreatic bud), L cells (*GCG*), N cells (*NTS*), K cells (*GIP*), I cells (*CCK*) and enterochromaffin cells (*TPH1*) either expressing neuropeptide W (*NPW*) or *TAC1*).

Fetal-specific subsets included proximal progenitors (*FGG*, *BEX5*) as described in refs. [1–3,52], distal progenitors enriched for colon genes (*CKB*,

*AKAP7, GPC3*) and CLDN10 cells that are possibly pancreatic progenitors based on expression of *DLK1, PDX1, RBPJ, CPA1* and *SOX9*[53]. This population also highly expressed *CLU*, a marker for intestinal revival stem cells that has previously been described[54].

*Endothelial lineage cells* (*PECAM1, CDH5*) were subdivided into arterial (*GJA4, HEY1, HEY2, EFNB2*), venous (*ACKR1, VWF*) and lymphatic endothelium (*PROX1, LYVE1, CCL21*). Arterial and venous cells from fetal and adult donors formed separate clusters and differed in gene expression, possibly reflecting fetal cell immaturity. Among age-shared arterial genes were *GJA5, SEMA3G, HEY1, HEY2* and age-shared venous genes were *ACKR1, ADGRG6, CPE, APLNR*. Capillary clusters were defined based on expression of RGCC and VWA1. Arterial capillaries specifically expressed *CA4* and *FCN3*.

Lymphatic endothelium further separated into six clusters, including LEC1 (*ACKR4, OTC*), resembling cells lining the subcapsular sinus ceiling in the human lymph nodes;[31] LEC2 (*GP1BA, UBD, ANO9, FIBIN, PAPLN*) and only LEC expressing MADCAM1 and reassembled LECs lining subcapsular sinus floor in human lymph nodes[31]. We further define LEC3 (*ADGRG3^{hi}*) present mostly in the paediatric and adult intestinal regions, and LEC4 (SATB2^{hi}, *PTX3^{hi}, CXADR^{hi}*) specific to developing gut (Extended Data Fig. 7c) that may represent differentiated and immature lymphatic vessels, respectively. LEC5 (*CLDN11, DEGS2, SBSPON, ANGPT2^{hi}, GJA4^{hi}*) reassembled collecting lymphatic valves in lymph nodes[31].

*Neural lineage cells* were defined on the basis of observations in mouse embryos[55]. Neuronal branches were named branch A or B on the basis of expression of *ETV1* or *BNC2*, respectively. Subpopulations of the branches were named A1–A4 and B1–B3 and combinatorial gene markers are provided in Fig. 3a, b. Branch A1 was functionally named inhibitory motor neurons (iMN) based on expression of *GAL, NOS1* and *VIP*; branch A2 was a mixed IPAN/IN population based on *NTNG1* and *NXPH2*. Branch A3 was a second subset of IPAN/IN equivalent to adult PIN3 and PSN3;[22] Branch A4 were annotated as interneurons (IN) based on *NEUROD6* expression[22]. Branch B1 were annotated as immature excitatory motor neurons (eMN) based on *NXPH4* and *NDUFA4L2*;[22] branch B2 was a second cluster of eMN based on expression of *BNC2* and *PENK*;[22,55] branch B3 (IPAN) was assigned the IPAN label based on *DLX3, ANO2, NOG* and *NTRK3*. Branch B3 also showed the expression of *CALB2* and *SST* described in the human IPAN A population[23].

All terminal glial cells expressed high levels of ENCC progenitor/terminal glial marker genes including *FOXD3, MPZ, CDH19, PLP1, SOX10, S100B* and *ERBB3*, but lacked *RET*. Three types of enteric glia (*S100B, CRYAB, MPZ*) were observed: glia 1 (*DHH, RXRG, NTRK2, MBP*), glia 2 (*ELN, TFAP2A, SOX8, BMP8B*), glia 3 (*BCAN, APOE, CALCA, HES5, FRZB*) and a subset of differentiating glia (*COL20A1*) were present at 12–17 PCW (Fig. 3b). Differentiating glia (*COL20A1*) cluster annotation was based on expression of glial markers, positioning in between differentiating subsets and scVelo results.

*Mesenchymal lineage cells* included previously described mesodermal populations mesoderm 1 (*HAND1, HAND2, PITX2*) and mesoderm 2 (*ZEB2*);[1] stromal 1 (*ADAMDEC1*) subset either highly expressing *ADAM28* or *CCL11, CCL8, CCL13;* stromal 2 (*PDGFRA, BMP4*) either enriched for *F3, NPY* or *CH25H, MMP1* expression; stromal 3 (*C7*) expressing *KCNN3, LRRC3B* or *C3, CLEC3B, SEMA3E*. We also observe the stromal 4 population (*MMP1, MMP3, PDPN, COL7A1, CHI3L1*) most recently described in the fetal gut[3]. In addition, we observe populations consistent with lymph-node immune-organizing fibroblasts including T reticular cells (*CCL19, GREM1^{hi}, TNFSF13B*)[32], follicular dendritic cells (*CXCL13, CR1, CR2*)[32], a population of FMO2 stromal cells (*LMO3, RASD1, PODN^{hi}*) found in the mLN samples that may represent adipose stromal cell population based on the expression of *FOXO1, KLF15, ZBTB16, LMO3, HSD11B1*[56,57]. Myofibroblasts were identified on the basis of expression of actin (*ACTA2*) and transgelin (*TAGLN*), fibroblast characteristic decorin (*DCN*), but

lacked smooth muscle marker desmin (*DES*). Myofibroblast populations further differed in the expression of *HHIP, NPNT, SYT10* or *RSPO2, SYT1, PTGER1*. Smooth muscle cells were annotated on the basis of high expression of *DES*, calponin (*CNN1*)[58] and actin/myosin chain expression (*ACTA2, MYH11*) and subsets were further characterized following the annotation in ref. [3]. Interstitial cells of Cajal (ICC) were identified on the basis of expression of *KIT, ANO1* and *ETV1*[32,59]. Other ICC genes included *DLK1, CDH8, CDH10* and *PRKCQ*. Cycling stromal cells were defined based on expression of *MKI67, TOP2A, CDK1* and other cell cycle genes. Fetal mLTo cells were defined as discussed in the text. Additionally, mLTo cells were high for *UBD, CLSTN3, SLC22A3, TNFSF11* and *APLNR* expression.

Pericytes were identified on the basis of expression of *NOTCH3, MCAM* (CD146) and *RGS5*. Immature pericytes were annotated based on high expression of *PDGFRB, CSPG4* (encoding NG2)[58] and was marked by high *NDUFA4L2* expression. We further annotate contractile pericytes (*ACTA2^{hi}*) that were specifically marked by expression of *PLN, RERGL, KCNA5, KCNAB1, NRIP2*. Angiogenic pericytes were annotated based on high expression of *PRRX1* and *PROCR* among pericyte populations (also expressed in stromal cells) as described in ref. [3], and also more specifically marked by *ENPEP, ABCC8, COL25A1* and *TEX41*. We also observe a population of pericytes marked by CD36[60] (named 'Pericyte').

Mesothelial cells were defined on the basis of *KRT19, LRRN4* and *UPK3B* expression, as observed previously[1,61], and additionally observe the expression of *TNNT1, RASSF7* and *KLK11* in these cells. Mature pericytes captured in adult tissues highly expressed *PRG4, MT1F, MT1G, CP4BPA*, and *HAS1*. In addition, we observe a population of mesothelial cells expressing *RGS5, TMEM235* and *SERPINE3*.

*B lineage cells* were defined as the cluster having positive expression of *MS4A1* (encodes CD20) and *CD19*. Among fetal B cells, common lymphoid progenitor (CLP) cells were classified on the basis of minimal expression of the lineage markers and specific expression of *CD34, SPINK2* and *FLT3*. Pro-B cells had specific expression of *DNTT* (TdT), recombination-activating genes (RAG) 1 and 2, high expression of B lymphocyte antigen receptor *CD79B*, V-set pre-B cell surrogate light chains (*VPREB1, VPREB3*) and expression of immunoglobulin-lambda chain *IGLL1*. Pre-B cells specifically expressed *CD38* and had higher expression of *CD19* than pro-B cells. Immature B cells had the highest expression of *MS4A1* among B lineage cells and expression of immunoglobulin heavy chains M and D. Adult B lineage cells had distinct clusters of naive B cells (Fc fragment of IgM receptor- FCMR), memory B cells (*SELL* (encodes CD62L)) and a population of memory B cells that specifically expressed transmembrane immunoregulatory molecule *FCRL4* (encodes protein also known as FcRH4) that has been suggested to be tissue-restricted[62] Class switch IgA and IgG plasma cells showed expression of the syndecan, *SDC1*, and immunoglobulin heavy chains A and G, respectively. Cycling B cells were characterized by specific expression of *MKI67* (encodes Ki67) and genes involved in DNA replication, *HMGB2, TUBA1B* and *UBE2C*.

*T lineage cells* were determined as the cluster of fetal, paediatric and adult cells with expression of T cell receptor component *CD3D* and subpopulations were annotated as previously described[4]. CD4 T cells had expression of *CD4*, but not *CD8A*, and vice versa for CD8 T cells. For each of these T cell groups, a population of paediatric/adult *SELL* (encodes CD62L) naive/central memory cells and *CD69* activated cells was assigned. 'Activated T' are fetal cells that have productive TCRαβ chains as determined by paired V(D)J sequencing. In addition, among postnatal CD4 T cells are *CXCR3,IFNG* T_H1 and *RORA^{hi},IL22, CD20* T_H17 cells. Paediatric CD CD4 T cells labelled as T_H1 and T_H17 showed additional co-expression of *IL22* and *IL26*, matching previously reported expression of these molecules in dysfunctional CD8 T cells in ulcerative colitis[8]. T regulatory cells were defined by *FOXP3, CTLA4, TIGIT* expression and T follicular helper cells expressed *CXCR5* and *PDCD1*. Two subsets of CD8 T memory cells were observed: CD8 T_mem and

*CX3CR1*-expressing CD8 $T_{mem}$ cells that both expressed *FGFBP2, S1PR5, FCGR3A* and *TGFBR3*.

Adult gdT subsets differentially expressed gamma and delta variable chain genes. Paediatric gdT (*TRDC*) cells did not have TCR sequencing data and did not express specific variable chains (Extended Data Fig. 2a). Specific expression of *TRGV2, TGVG4* and *TRVG5/TRVG7*, which each encode a variable chain of the gamma TCR chain, further defined subpopulations respectively. We also observe a population of *TRDV2/ TRGV9* gdT cells[63] in the fetal gut. NK T cells expressed *CD3D* as well as NK genes *GZMA, NKG7* and *PRF1*. MAIT cells expressed *TRAV1, TRAV2* and *SLC4A10*. ILC2 expressed *PTGDR2, HPGDS, IL1RL1* and *KRT1*[64] and were found in the fetal samples.

NK cells were defined on the basis of *EOMES, PRF1, NKG7* and KIR receptor expression. Adult ILC3s were defined based on the expression of *RORC, IL1R1, IL23R, KIT*[64] and further expressed *TNFS4* and *PCDH9*. ILCP were defined as Lin⁻ *IL7R* (encodes CD127), *KIT* (CD117), *RORC* (RORγ) cells that further express *CCR6, NRP1*, but not *NCR2* (encodes NKp44), as described in the liver[65]. ILCPs also highly expressed chemokine receptors *CXCR5* and *CCR7*, cell adhesion molecule encoded by *SCN1B*[66] and serine protease encoded by *HPN*.

LTi-like NCR⁺ ILC3 cells had highest expression of *TNFRSF11A* (RANK) and its ligand *TNFSF11* (RANKL), as well as *NCR1* (NKp46) and *NCR2* (NKp44), whereas LTi-like NCR- ILC3s lacked *NCR2* (NKp44) and expressed *IL17A, ITGAE* and *CCR9*, and NK-associated genes including *NKG7, PRF1* and *GZMA*, consistent with NCR⁻ ILC3 cells described in mice[67]. We observed the expression of *IL22* in the adult compared to fetal NRC2⁺ ILC3s, suggesting that fetal gut ILC3 represents an immature ILC3 counterpart. Reassuringly, the LTi-like subtypes were identifiable in full-length scRNA-seq data from fetal ileum (Extended Data Fig. 11).

*Myeloid lineage cells* further sub-clustered to classical monocytes (*FCN1, S100A4, S100A6*), dendritic cells, macrophage, mast cells (*GATA2, CPA3, HPGDS*) and megakaryocytes (*GP9, LCN2*). Among dendritic cells (DCs) we observed cDC1 (*CLEC9A*) and cDC2 (*CLEC10A*), lymphoid DCs (*LAMP3*) and plasmacytoid DCs (pDCs; *CLEC4C, JCHAIN*). Among macrophage populations, we observe a subset of classical macrophages (*CD163, C1QB, C1QC*), LYVE1⁺ macrophages (*RNASE1, SPP1*) that have been previously observed in around heart vessels[68] and lung[69] and inflammatory macrophages (*MMP9*);[10] we also observe a fetal-specific population of mono-neutrophil progenitors (*MPO, AZU1, ELANE*) previously described in human fetal liver[30] and *CLC* mast cells that could represent immature fetal states.

### Cell-type scoring

For MHCII scoring of B cells we use the following genes: *SECTM1, CD320, CD3EAP* (also known as *POLR1G*) *CD177, CD74, CIITA, RELB, TAP2, HLA-DRA, HLA-DRB5, HLA-DRB1, HLA-DQA1, HLA-DQB1, HLA-DQB1-AS1, HLA-DQA2, HLA-DQB2, HLA-DOB, HLA-DMB, HLA-DMA, HLA-DOA, HLA-DPA1, HLA-DPB1*. Glial or neuronal cell signature score was calculated using the curated gene sets from ref.[70] (glia: *ERBB3, PLP1, COL18A1, SOX10, GAS7, FABP7, NID1, QKI, SPARC, MEST, WWTR1, GPM6B, RASA3, FLRT1, ITPRIPL1, ITGA4, POSTN, PDPN, NRCAM, TSPAN18, RGCC, LAMA4, PTPRZ1, HMGA2, TGFB2, ITGA6, SOX5, MTAP, HEYL, GPR17, TTYH1*; neuronal: *ELAVL3, ELAVL4, TUBB2B, PHOX2B, RET, CHRNA3*). NF-kB signalling pathway score was calculated using genes from http://www.gsea-msigdb.org/gsea/msigdb/genesets.jsp (*NFKB2, BIRC3, TNFAIP2, TNIP1, NOTCH2, TMEM173* (also known as *STING1*) *TIFA, PRDX4, CAMK4, BCL3, CHUK, IKBKB, IKBKE, NFKB1, NFKB2, NFKBIA, NFKBIB, NFKBIE, REL, RELA, RELB*). The scoring was done using sc.tl.score_genes() function with default parameters to calculate the average expression of selected genes substrated with the average expression of reference genes.

### Intestinal organoid analysis

Single-cell count matrices from three organoid growth conditions were combined together using Pandas (v.1.1.2) and NumPy (v.0.25.2) packages. Cells with fewer than 8,000 genes and with less than 20% mitochondrial reads were included in the analysis. Genes with expression in fewer than 3 cells were also excluded. For *PLCG2* expression comparison we use normalized (sc.pp.normalize_per_cell) and log-transformed (sc.pp.log1p) counts. The data were plotted using Seaborn package bar plot and swarmplot functions (v.0.11.0).

### Pre-processing and analysis of Smart-seq2 sequencing data

Cells with more than 6,000 genes and greater than 25% mitochondrial reads were excluded, before regression of 'n_counts', 'percent_mito' and 'G2M_score'. Cells positive for PTPRC expression (logTPM+1 > =0.2) were taken forward for downstream analysis. T cell receptor sequences generated using the Smart-seq2 scRNA-seq protocol were reconstructed using the TraCeR software (https://hub.docker.com/r/teichlab/tracer/) as described previously[71].

### Differential cell-state abundance analysis for BEST4 cells

To identify region-specific subpopulations, we performed compositional analysis between BEST4 enterocytes from small and large intestine tissue, using a tool for differential abundance testing on *k*-nearest neighbour (KNN) graph neighbourhoods, implemented in the R package miloR (v.0.99.8) https://github.com/MarioniLab/miloR[14].

In brief, we performed PCA dimensionality reduction and KNN graph embedding on the BEST4 enterocytes. We define a neighbourhood as the group of cells that are connected to a sampled cell by an edge in the KNN graph. Cells are sampled for neighbourhood construction using the algorithm proposed previously[72]. For each neighbourhood we then perform hypothesis testing between conditions to identify differentially abundant cell states whilst controlling the FDR across the graph neighbourhoods.

We test for differences in abundance between the cells from small and large intestine tissue in adult samples and fetal samples. To identify markers of small-intestine-specific and large-intestine-specific subpopulations, we performed differential gene expression (DGE) analysis between adult cells in neighbourhoods enriched for small intestine cells and in neighbourhoods enriched for large intestine cells (10% FDR for the differential abundance test). The DGE test was performed using a linear model implemented in the package limma[73] (v.3.46.0), using 10% FDR, and aggregating expression profiles by sample (implemented in the function findNhoodGroupMarkers of the miloR package, with option aggregateSamples = TRUE). Gene Ontology enrichment analysis was performed using the R package clusterProfiler (v.3.18.1).

### RNA velocity and diffusion map pseudotime analyses

For neural cell trajectory analysis we use scVelo 0.21 package implementation in Scanpy 1.5.1[74]. The data were sub-clustered on fetal neural cells and pre-processed using functions for detection of minimum number of counts, filtering and normalization using scv.pp.filter_and_normalize and followed by scv.pp.moments function. The gene-specific velocities were obtained using scv.tl.velocity(mode = 'stochastic') and scv.tl.velocity_graph() by fitting a ratio between unspliced and spliced mRNA abundances. The gene specific velocities were visualized using scv.pl.velocity_graph() or scv.pl.velocity_embedding_grid() functions. To visualize genes that change along the pseudotime we use sc.pl.paga_path() function with pseudotime set to monocle3 pseudotime. This function required calculation of PAGA parameters and dpt_pseudotime with functions as follows: sc.tl.paga(), sc.pl.paga(), sc.tl.draw_graph(init_pos = 'paga'), sc.tl.dpt()[75].

### BCR analysis

Single-cell BCR analyses were performed as described previously[76]. In brief, poor quality or incomplete V(D)J contig sequences were discarded and all IgH sequences for each donor were combined together. IgH sequences were annotated with IgBLAST[77] before isotype reassignment using AssignGenes.py (pRESTO[78]). Ambiguous V gene calls were

corrected using TIgGER v.03.1[79] before identifying clonally related sequences with DefineClones.py (ChangeO v.0.4.5[79]) using a threshold of 0.2 for nearest-neighbour distances. The germline IgH sequence for each clonal family was determined using CreateGermlines.py (ChangeO v.0.4.5) followed by using observedMutations (Shazam v.0.1.11[79]) to calculate somatic hypermutation frequencies for individual sequences. Finally, for integration with the single-cell gene expression object, the number of high quality and annotated contigs per Ig chain (IgH, IgK, IgL) was determined for each cell barcode. If multiple unique sequences for a given chain were detected, that cell was annotated as 'Multi' and not considered in further analysis. BCR metadata was combined with the scRNA object for downstream analysis and comparison of different B cell populations.

## Cell–cell communication analysis

To infer cell–cell communication and screen for ligands and receptors involved we applied the CellPhoneDB v.2.0 Python package[80,81] on the normalized raw counts and fine cell-type annotations from the second trimester intestinal samples (12–17 PCW). We use default parameters and set subsetting to 5,000 cells. To identify the most relevant interactions, we subset specific interactions on the basis of the ligand/receptor expression in more than 10% of cells within a cluster and where the $\log_2$ mean expression of the pair is greater than 0. The selected interactions were plotted as expression of both ligand and receptor in relevant cell types.

## Gene expression linked to enteric nervous system disease

HSCR-related genes were curated from The Human Phenotype Ontology website (https://hpo.jax.org/app/, Aganglionic megacolon HP:0002251) and ref. [25]. We selected genes with expression greater than or equal to 0.1 in neural lineage single cells and calculated mean expression per cluster and organ. Expression was visualized using seaborn.clustermap() function (v.0.11.0).

## Cell-type composition analysis using metadata

The number of cells for each sample ($n = 159$ samples in total with complete metadata) and coarse-grain cell type (9 different cell types in total) combination was modelled with a generalized linear mixed model with a Poisson outcome. The five clinical factors (age group, donor, biopsy, disease and gender) and the three technical factors (fraction, enzyme and 10x kit) were fitted as random effects to overcome the collinearity among the factors. The effect of each clinical or technical factor on cell-type composition was estimated by the interaction term with the cell type. The 'glmer' function in the lme4 package implemented on R was used to fit the model. The standard error of the standard deviation parameter for each factor was estimated using the numDeriv package. The effect size of each level of a clinical or technical factor was obtained by the posterior mean of the corresponding random effect coefficient and the local true sign rate (LTSR) was calculated from the posterior distribution. See Supplementary Note Section 1 for more details.

## Cell-type enrichment analysis for IBD-GWAS genes

Inflammatory bowel disease genome-wide association study (IBD GWAS) summary statistics of Crohn's disease and ulcerative colitis were obtained from the International Inflammatory Bowel Disease Genetics Consortium (IIBDGC) (https://www.ibdgenetics.org/). The GWAS enrichment analysis of 103 annotated gut cell types for Crohn's disease and ulcerative colitis was performed using a fGWAS approach[82,83], often used for fine-mapping and enrichment analysis of various functional annotations for molecular quantitative trait and GWAS loci. The association statistics (log odds ratios and standard errors) were converted into the approximate Bayes factors using the Wakefield approach[84]. A *cis*-regulatory region of 1 Mb centred at the transcription start site (TSS) was defined for each gene (Ensembl GRCh37 Release 101). The Bayes factors of variants existing in each *cis* region were weighted and averaged by the prior probability (an exponential function of TSS proximity) estimated from the distance distribution of regulatory interactions[85]. The likelihood of an fGWAS model was given by the averaged Bayes factors across all genome-wide genes multiplied by the feature-level prior probability obtained from a linear combination of cell-type-specific expression and the averaged expression across all cell types as a baseline expression. The enrichment of each cell type was estimated as the maximum likelihood estimator of the effect size for the cell-type-specific expression. The code of the hierarchical model (https://github.com/natsuhiko/PHM) was utilized for the enrichment analysis. The detailed model derivation is demonstrated in Supplementary Note section 2.

## Visium spatial transcriptomics sample preparation

10x Genomics Visium protocol was applied on optimal cutting temperature medium (OCT)-embedded fresh frozen samples. All tissues were sectioned using the Leica CX3050S cryostat and were cut at 10 μm. The samples were selected on the basis of morphology, orientation (based on H&E) and RNA integrity number that was obtained using Agilent 2100 Bioanalyzer. Tissue optimization was performed to obtain permeabilization time for fetal tissue (12 min) and after optimization the Visium spatial gene expression protocol from 10X Genomics was performed using Library Preparation slide and following the manufacturer's protocol. After transcript capture, Visium Library Preparation Protocol from 10x Genomics was performed. All images for this process were scanned at 40× on Hamamatsu NanoZoomer S60. Eight cDNA libraries were diluted and pooled to a final concentration of 2.25 nM (200 μl volume) and sequenced on 2× SP flow cells of Illumina NovaSeq 6000.

## 10x Genomics Visium data processing

10x Genomics Visium spatial sequencing samples were aligned to the human transcriptome GRCh38-3.0.0 reference (consistently with single-cell RNA-seq samples) using 10x Genomics SpaceRanger v.1.2.1 and exonic reads were used to produce mRNA count matrices for each sample. 10x Genomics SpaceRanger was also used to align paired histology images with mRNA capture spot positions in the Visium slide. The paired image was used to determine the average number of nuclei per Visium location in the tissue and used as a hyperparameter in the spatial mapping of cell types.

## Spatial mapping of cell types with cell2location

To spatially map developing gut cell types in situ, 10x Genomics Visium mRNA count matrices were integrated with scRNA-seq data using the cell2location method as described in detail previously[34]. In brief, the cell2location model estimates the abundance of each cell population in each location by decomposing mRNA counts in 10x Genomics Visium data using the transcriptional signatures of reference cell types. Reference signatures of 65 cell populations (neuronal and mesenchymal subtypes were grouped together where relevant to simplify interpretation of spatial mapping) from the 12–17 PCW small intestine samples were estimated using a negative binomial regression model provided in the cell2ocation package. We provide spatial cell abundance maps of all 65 cell subsets on GitHub (https://github.com/vitkl/fetal_gut_mapping/) and the distribution of all 65 cell subsets across tissue zones in Extended Data Fig. 9a–c.

Untransformed and unnormalized mRNA counts were used as input to both the regression model for estimating signatures (filtered to 13,904 genes and 124,049 cells) and the cell2location model for estimating spatial abundance of cell populations (filtered to 13,904 genes shared with scRNA-seq, 4,645 locations, 3 experiments analysed jointly).

Cell2location was used with the following settings (all other settings set to default values): training iterations: 40,000; cells per location $\hat{N} = 12$, estimated on the basis of comparison with histology image paired with 10x Genomics Visium; cell types per location $\hat{A} = 8$, assum-

ing that most cells in a given location are of a different type; co-located cell type groups per location $\hat{Y} = 4$.

To identify tissue zones and groups of cell types that belong to them (Extended Data Fig. 15c, dot plot), conventional non-negative matrix factorization (NMF) was applied to cell abundance estimated by cell-2location. The NMF model was trained for a range of factors and tissue zones R = {8,…,35} and the decomposition into 17 factors was selected as a balance between segmenting relevant tissue zones (lymphoid structures, blood vessel types) and over-separating known zones into several distinct factors. NMF weight for each factor and cell type is shown in Extended Data Fig. 15c.

## Cryosectioning, single-molecule fluorescence in situ hybridization and confocal imaging

Fetal gut tissue was embedded in OCT and frozen on an isopentane-dry ice slurry at −60 °C, and then cryosectioned onto SuperFrost Plus slides at a thickness of 10 μm. Before staining, tissue sections were post-fixed in 4% paraformaldehyde in PBS for 15 min at 4 °C, then dehydrated through a series of 50%, 70%, 100% and 100% ethanol, for 5 min each. Staining with the RNAscope Multiplex Fluorescent Reagent Kit v2 Assay (Advanced Cell Diagnostics, Bio-Techne) was automated using a Leica BOND RX, according to the manufacturers' instructions. After manual pre-treatment, automated processing included epitope retrieval by protease digestion with Protease IV for 30 min prior to RNAscope probe hybridization and channel development with Opal 520, Opal 570, and Opal 650 dyes (Akoya Biosciences). Stained sections were imaged with a Perkin Elmer Opera Phenix High-Content Screening System, in confocal mode with 1 μm z-step size, using a 20× water-immersion objective (NA 0.16, 0.299 μm per pixel). Channels: DAPI (excitation 375 nm, emission 435–480 nm), Opal 520 (ex. 488 nm, em. 500–550 nm), Opal 570 (ex. 561 nm, em. 570–630 nm), Opal 650 (ex. 640 nm, em. 650–760 nm). The fourth channel was developed using TSA-biotin (TSA Plus Biotin Kit, Perkin Elmer) and streptavidin-conjugated Atto 425 (Sigma-Aldrich).

## Flow cytometry validation of Fcgr on mice tuft cells

C57BL/6 mice received either normal drinking water or 2% (w/v) 36,000–50,000 MW dextran sodium sulfate (DSS) (MP Biomedicals) to induce colitis. For DSS treatment, mice received DSS water for 5 days followed by 14 days of normal drinking water, and then a final 5 days of 2% (w/v) DSS prior to being culled.

The small intestines of mice were flushed of faecal content with ice-cold PBS, opened longitudinally, cut into 0.5-cm pieces, and washed by vortexing three times with PBS with 10 mM HEPES. Tissue was then incubated with an epithelial stripping solution (RPMI-1640 with 2% (v/v) FCS, 10 mM HEPES, 1 mM DTT and 5 mM EDTA) at 37 °C for two intervals of 20 min to remove epithelial cells. The epithelial fraction was subsequently incubated at 37 °C for 10 min with dispase (0.3 U ml$^{-1}$, Sigma-Aldrich) and passed through a 100-μm filter to obtain a single-cell suspension. Cells were blocked for 20 min at 4 °C with 0.5% (v/v) heat-inactivated mouse serum followed by extracellular staining in PBS at 4 °C for 45 min with the following antibodies; EpCAM-FITC (1:400, G8.8, Invitrogen), CD45-Bv650 (1:200, 30-F11, BioLegend), CD11b-Bv421 (1:300, M1/70, BD Biosciences), Siglec-F-APC (1:200, 1RNM44N, Invitrogen), FcγRI-PE (1:200, X54-5/7.1, BioLegend), FcγRIIB-PE (1:200, AT130-2, Invitrogen), FcγRII/RIII-PE-Cy7 (1:200, 2.4G2, BD Biosciences), FcγRIV-PE (1:200, 9E9, BioLegend) and Rat IgG2b, κ isotype-PE-Cy7 (1:200, LOU/C, BD Biosciences). Cells were then stained with LIVE/DEAD Fixable Aqua Dead Cell Stain Kit (Thermo Fisher Scientific) for 20 min at room temperature, fixed with 2% PFA, and analysed on a CytoFLEX LX (Beckman Coulter) flow cytometer.

## Reporting summary

Further information on research design is available in the Nature Research Reporting Summary linked to this paper.

## Data availability

The expression data for fetal and adult regions is available on an interactive website: https://www.gutcellatlas.org/. Raw sequencing data are available at ArrayExpress (https://www.ebi.ac.uk/arrayexpress) with accession numbers E-MTAB-9543, E-MTAB-9536, E-MTAB-9532, E-MTAB-9533 and E-MTAB-10386. Previously published first trimester and paediatric data are available at ArrayExpress (E-MTAB-8901)[1]. For the purpose of Open Access, the authors have applied a CC BY public copyright licence to any Author Accepted Manuscript version arising from this submission. Source data are provided with this paper.

## Code availability

Processed single-cell RNA sequencing objects are available for online visualization and download at https://www.gutcellatlas.org/. The code generated during this study is available at Github: https://github.com/Teichlab/SpaceTimeGut, https://github.com/vitkl/fetal_gut_mapping/, https://github.com/natsuhiko/PHM.

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

**Acknowledgements** We acknowledge support from the Wellcome Sanger Cytometry Core Facility, Cellular Genetics Informatics team, Cellular Generation and Phenotyping (CGaP) and Core DNA Pipelines. This work was financially supported by the Wellcome Trust (WT206194, S.A.T.; 203151/Z/16/Z, R. A. Barker.); the European Research Council (646794, ThDefine, S.A.T.); an MRC New Investigator Research Grant (MR/T001917/1, M.Z.); and a project grant from the Great Ormond Street Hospital Children's Charity, Sparks (V4519, M.Z.). The human embryonic and fetal material was provided by the Joint MRC/Wellcome (MR/R006237/1) Human Developmental Biology Resource (https://www.hdbr.org/). K.R.J. holds a Non-Stipendiary Junior Research Fellowship from Christ's College, University of Cambridge. M.R.C. is supported by a Medical Research Council Human Cell Atlas Research Grant (MR/S035842/1) and a Wellcome Trust Investigator Award (220268/Z/20/Z). H.W.K. is funded by a Sir Henry Wellcome Fellowship (213555/Z/18/Z). A.F. is funded by a Wellcome PhD Studentship (102163/B/13/Z). K.T.M. is funded by an award from the Chan Zuckerberg Initiative. H.H.U. is supported by the Oxford Biomedical Research Centre (BRC) and the The Leona M. and Harry B. Helmsley Charitable Trust. We thank A. Chakravarti and S. Chatterjee for their contribution to the analysis of the enteric nervous system. We also thank R. Lindeboom and C. Talavera-Lopez for support with epithelium and Visium analysis, respectively; C. Tudor, T. Li and O. Tarkowska for image processing and infrastructure support; A. Wilbrey-Clark and T. Porter for support with Visium library preparation; A. Ross and J. Park for access to and handling of fetal tissue; A. Hunter for assistance in protocol development; D. Fitzpatrick for discussion on developmental intestinal disorders; and J. Eliasova for the graphical images. We thank the tissue donors and their families, and the Cambridge Biorepository for Translational Medicine and Human Developmental Biology Resource, for access to human tissue. This publication is part of the Human Cell Atlas: https://www.humancellatlas.org/publications.

**Author contributions** S.A.T., K.R.J. and M.H. initiated, designed and supervised the project; K.T.M. and K.S.-P. carried out adult tissue collection; K.R.J., R.E., M.D., S.P., L.B., S.F.V. and M.P. performed adult tissue processing and scRNA-seq experiments; R. A. Barker and X.H. supported collection of fetal tissue samples; S.N.L., R. A. Botting., I.G.K.'E., J.E. and C.D.C. supported fetal tissue processing and scRNA-seq experiments; F.P. and K.N. conducted organoid experiments; L.M., L.B. and E.S. performed library preparation; A.F. performed flow cytometry validation in mice and data interpretation; T.R.W.O., C.E.H. and L.S.C. provided pathology support; K.R., S.P. and O.A.B. performed tissue sectioning, staining and imaging; K.R.J. and R.E. analysed single-cell data and generated figures; S.L., N.K., K.P., S.V.D. and N.H. provided analysis support; V.K. analysed 10x Genomics Visium data; H.W.K. performed BCR analysis and contributed to data visualization; E.D. performed differential abundance analysis; J.C.M., M.D.M., M.K., K.B.M., M.Z., H.U. and M.R.C. contributed to interpretation of the results; and K.R.J., R.E. and S.A.T. wrote the manuscript. All authors contributed to the discussion and interpretation of results, as well as editing of the manuscript.

**Competing interests** In the past three years, S.A.T. has consulted for or been a member of scientific advisory boards at Roche, Qiagen, Genentech, Biogen, GlaxoSmithKline and ForeSite Labs. The remaining authors declare no competing interests.

**Additional information**
**Correspondence and requests for materials** should be addressed to Kylie R. James or Sarah A. Teichmann.

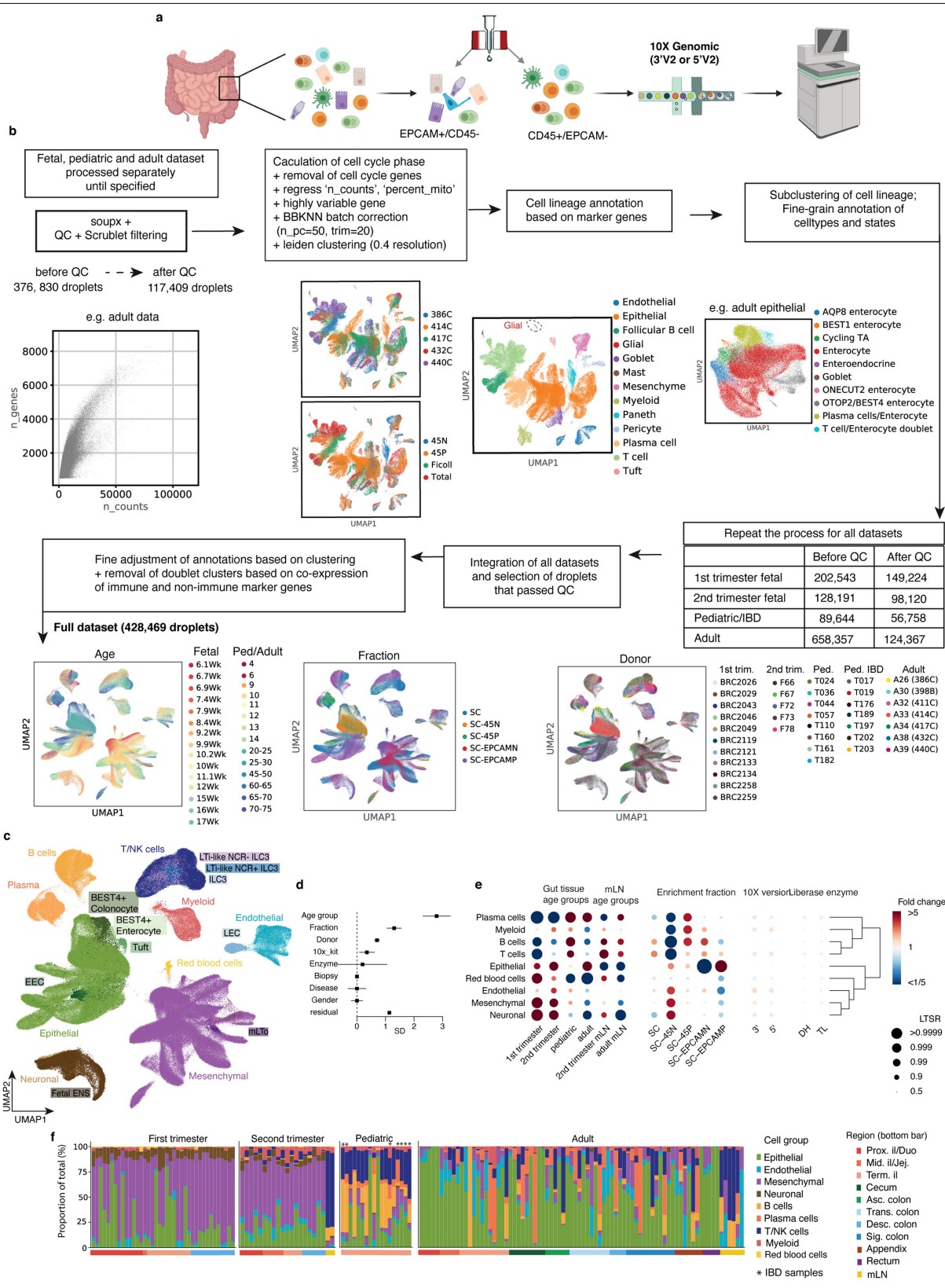

**Extended Data Fig. 1** | See next page for caption.

**Extended Data Fig. 1 | Data quality control. a**, Schematic showing tissue processing strategy for second trimester fetal, paediatric and adult scRNA-seq samples. After enzymatic dissociation, either total fraction was loaded onto a 10x Genomics Chromium chip or CD45$^{+/-}$ cell fractions were separated using magnetic cell sorting (MACS) and both fractions were loaded on the 10x Genomics Chromium chip separately. Lymph nodes were processed without enrichment. Second trimester fetal and adult cell samples were processed using 5′v2 10x Genomics Chromium kits (Methods). **b**, Pre-processing and quality control of single-cell RNA-seq data generated in this study and described previously[1]. In short, four datasets—namely first trimester fetal, second trimester fetal, paediatric healthy and Crohn's disease, and adult—were pre-processed separately (including quality control, soupX analysis and scrublet doublet removal). Firstly, dimension reduction, clustering and annotation by cell lineage was performed on each dataset separately. Each cell lineage was sub-clustered and a fine-grained cell type and cell state annotation was performed based on marker gene expression. The four datasets were then merged together and each lineage was sub-clustered to unify cell type labels where appropriate. UMAP visualizations show the combined dataset coloured by sample age, enrichment fraction and donor name. **c**, UMAP visualization of cellular landscape of the human intestinal tract coloured by cellular lineage. **d**, Forest plot showing the relative importance (explained standard deviation) of each technical/biological factor on the cell type proportion. The 95% confidence intervals were computed from $n = 1,431$ data points (9 cell types × 159 samples). See Method section for more details. **e**, Dot plot in which the fold change represents the enrichment (or depletion, low fold change in blue) of cells compared with baseline. The LTSR value represents statistical significance of the fold change estimate ranging from 0 to 1, where 1 represents a confident estimate. See Method section for more details. **f**, Bar plots with relative proportion of cell lineages in each 10x Genomics Chromium run grouped by anatomical region within the scRNA-seq dataset as in Fig. 1b.

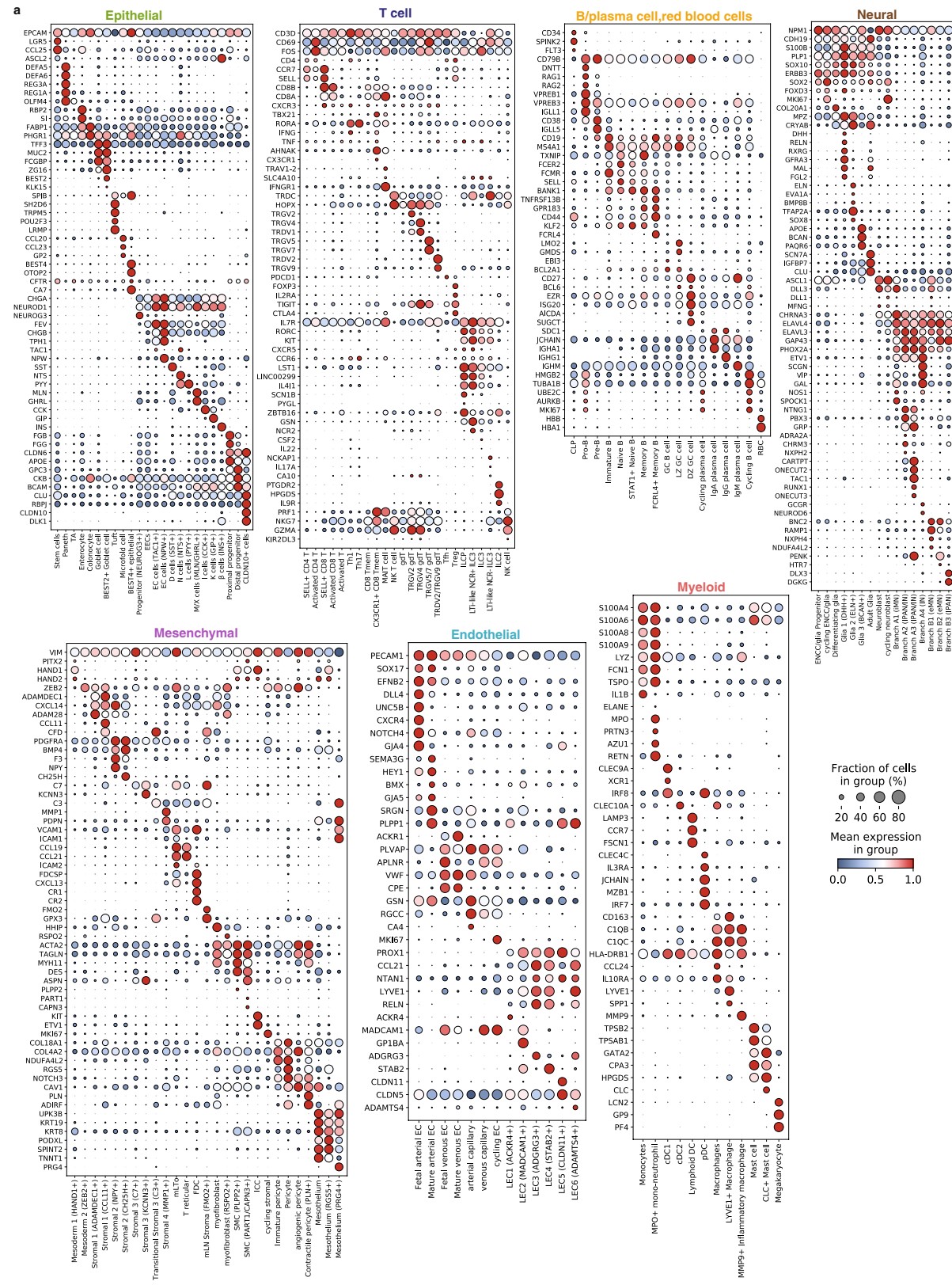

**Extended Data Fig. 2 | Cell types defined in the study. a,** Dot plot for expression of marker genes of cell types and states in each cell lineage in the scRNA-seq dataset. Relates to Supplementary Tables 3, 4.

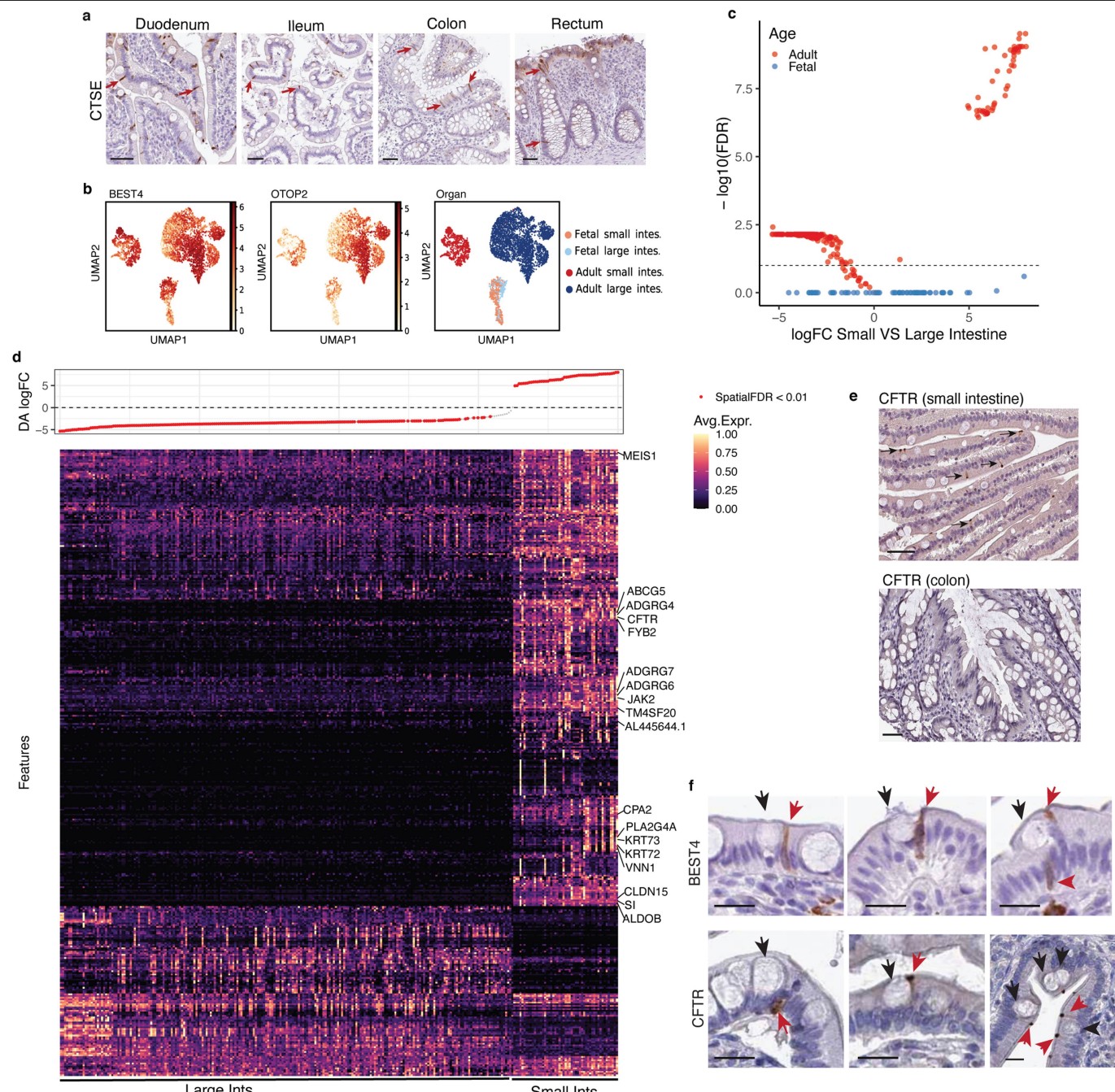

**Extended Data Fig. 3 | Region variability in BEST4 enterocytes.**
**a**, Expression of CSTE (antibody: CAB032687, $n = 3$ biologically independent samples for each region) in gut histological sections from proteinatlas.org. Scale bar = 50 μm. **b**, UMAP visualization of BEST4 enterocytes in scRNA-seq dataset coloured by key marker *BEST4/OTOP2* expression and region group (fetal and paediatric/adult). **c**, Volcano plot for differential abundance (DA) between cells from the small intestine and large intestine as in **b**. Each point represents a neighbourhood of BEST4 enterocytes (FDR: False Discovery Rate, logFC: log-Fold Change) for adult (red) and fetal samples (blue). The dotted line indicates the significance threshold of 10% FDR. **d**, Heat map showing the average neighbourhood expression of genes differentially expressed between

DA neighbourhoods in adult BEST4 enterocytes (1,502 genes) as in **c**. Expression values for each gene are scaled between 0 and 1. Neighbourhoods are ranked by log-fold change in abundance between conditions. Positive log-fold change is small intestine neighbourhoods and negative is large intestine neighbourhoods. **e**, Expression of CFTR (antibody: CAB001951/ HPA021939, $n = 3$ biologically independent samples for each region) in small intestinal (top) and colonic (bottom) histological sections from Human Protein Atlas (proteinatlas.org). Scale bar = 50 μm. **f**, Immunohistochemistry staining of BEST4 (HPA058564) and CFTR (HPA021939) in small intestinal sections as in **e**. Black arrows point to cells with goblet cell morphology and red arrows point to cells expressing either BEST4 or CFTR. Scale bar = 20 μm.

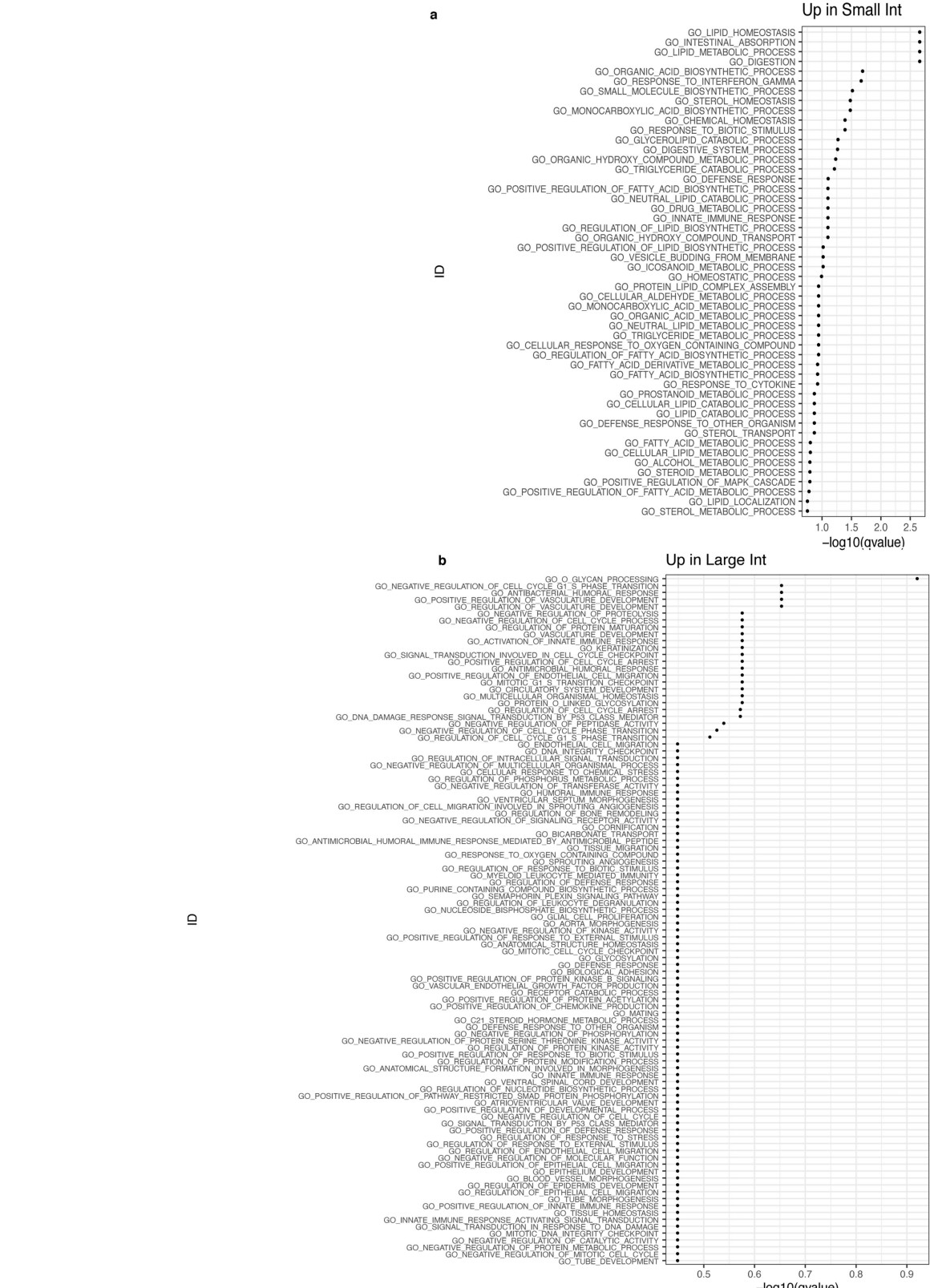

**Extended Data Fig. 4 | Function of BEST4 epithelial cells. a, b,** Gene ontology terms from genes upregulated in BEST4 enterocytes from adult small (**a**) versus large (**b**) intestines as determined from Milo analysis[13].

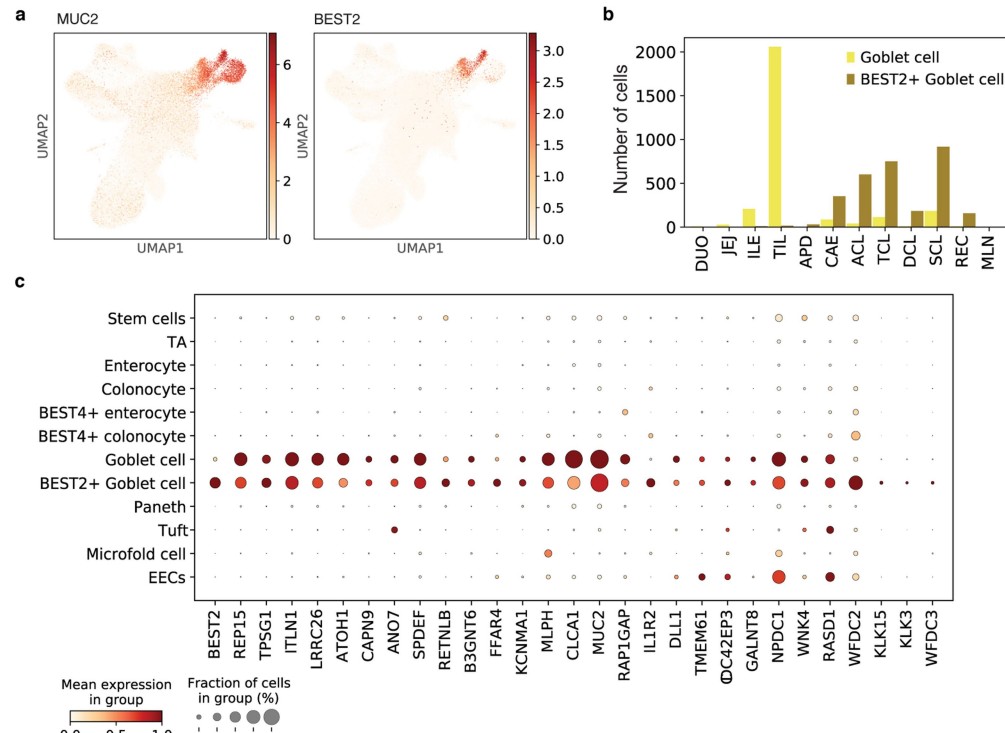

**Extended Data Fig. 5 | BEST2⁺ goblet cells. a**, UMAP visualization of expression of *MUC2* (indicating goblet cells) and *BEST2* in paediatric/adult epithelial cells from scRNA-seq dataset. **b**, Bar plot of the number of goblet cells captured across paediatric/adult intestinal tissues. **c**, Dot plot of gene expression correlating with *BEST2* expression across epithelial cell types from scRNA-seq dataset calculated using Jaccard Similarity measure.

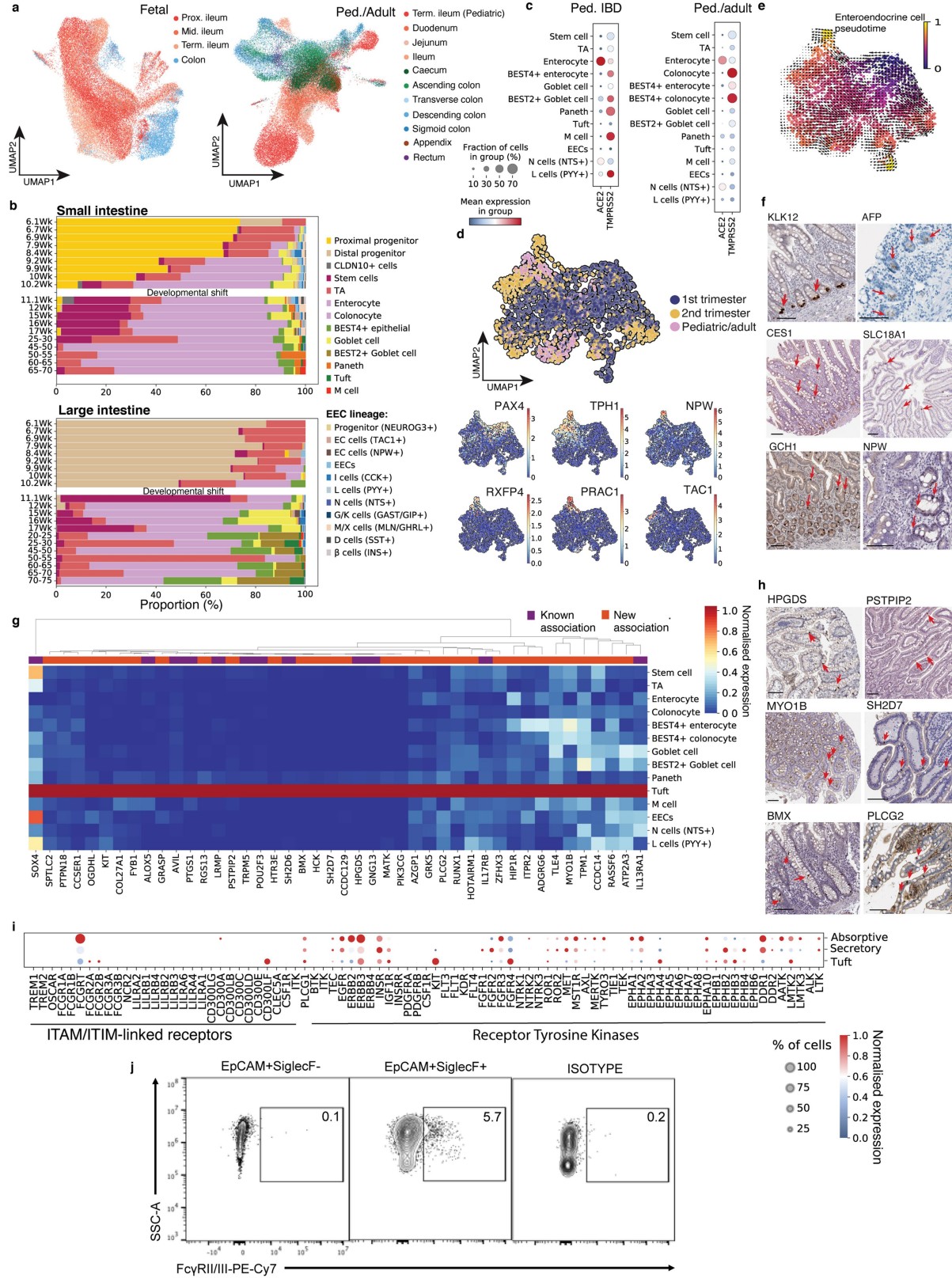

**Extended Data Fig. 6** | See next page for caption.

**Extended Data Fig. 6 | Epithelial cell types throughout intestinal life.**
**a**, UMAP of fetal (top) and pooled paediatric and adult (bottom) epithelial cells as in Fig. 2a, b coloured by gut region. **b**, Relative proportions of cell subtypes within total epithelial lineage as in Fig. 2a, b separated by donor age (row). Unit of age is years unless specified as weeks. **c**, Dot plot of *TMPRSS2* and *ACE2* expression by epithelial cells of the paediatric (left) and adult (right) intestine. **d**, **e**, UMAP of enteroendocrine cells (subsetted from Fig. 2a, b) coloured by **d**, (top) developmental age of donor and (bottom) normalised expression of key genes of NPW+ enterochromaffin cells and **e**, overlaid with calculated RNA velocity (arrows) and pseudotime (colour). **f**, Immunohistochemical staining of KLK12 (antibody: CAB025473, *n* = 3), AFP (antibody; HPA010607, *n* = 4), CES1 (antibody; HPA012023, *n* = 10), SLC18A1 (antibody; HPA063797, *n* = 12), GCH1 (antibody; HPA028612, *n* = 8), NPW (antibody; HPA064874, *n* = 8) in intestinal sections from Human Protein Atlas (proteinatlas.org). Red arrows highlight positive staining and n represents biological replicates across intestinal regions. **g**, Heat map of top differentially expressed genes in tuft cells across epithelial cell types in scRNA-seq dataset. The legend indicates whether the gene has a known association with tuft cells (purple) or are novel (orange). **h**, Immunohistochemical staining of HPGDS (antibody: HPA024035, *n* = 8), PSTPIP2 (antibody: HPA040944, *n* = 8), BMX (antibody: CAB032495, *n* = 7), MYO1B (antibody: HPA060144, *n* = 6), FYB1 (antibody: CAB025336, *n* = 7), SH2D7 (antibody: HPA076728, *n* = 7), PLCG2 (antibody: HPA020099, *n* = 8) protein expression in small intestine from Human Protein Atlas (proteinatlas. org). *n* represents biological replicates across intestinal regions. **i**, Dot plot showing expression of ITAM- and ITIM-linked receptors and receptor tyrosine kinases across tuft cells and pooled absorptive (TA and enterocytes) and secretory (Paneth, goblet and EECs) epithelial cells. **j**, Representative flow cytometry plots of FcγRII/III staining on EpCAM$^+$SiglecF$^-$ (non-tuft epithelial cells) and EpCAM$^+$SiglecF$^+$ (tuft cells) cells and isotype staining of EpCAM$^+$SiglecF$^+$ cells. Numbers show the percentage of cells within the gate out of the total population.

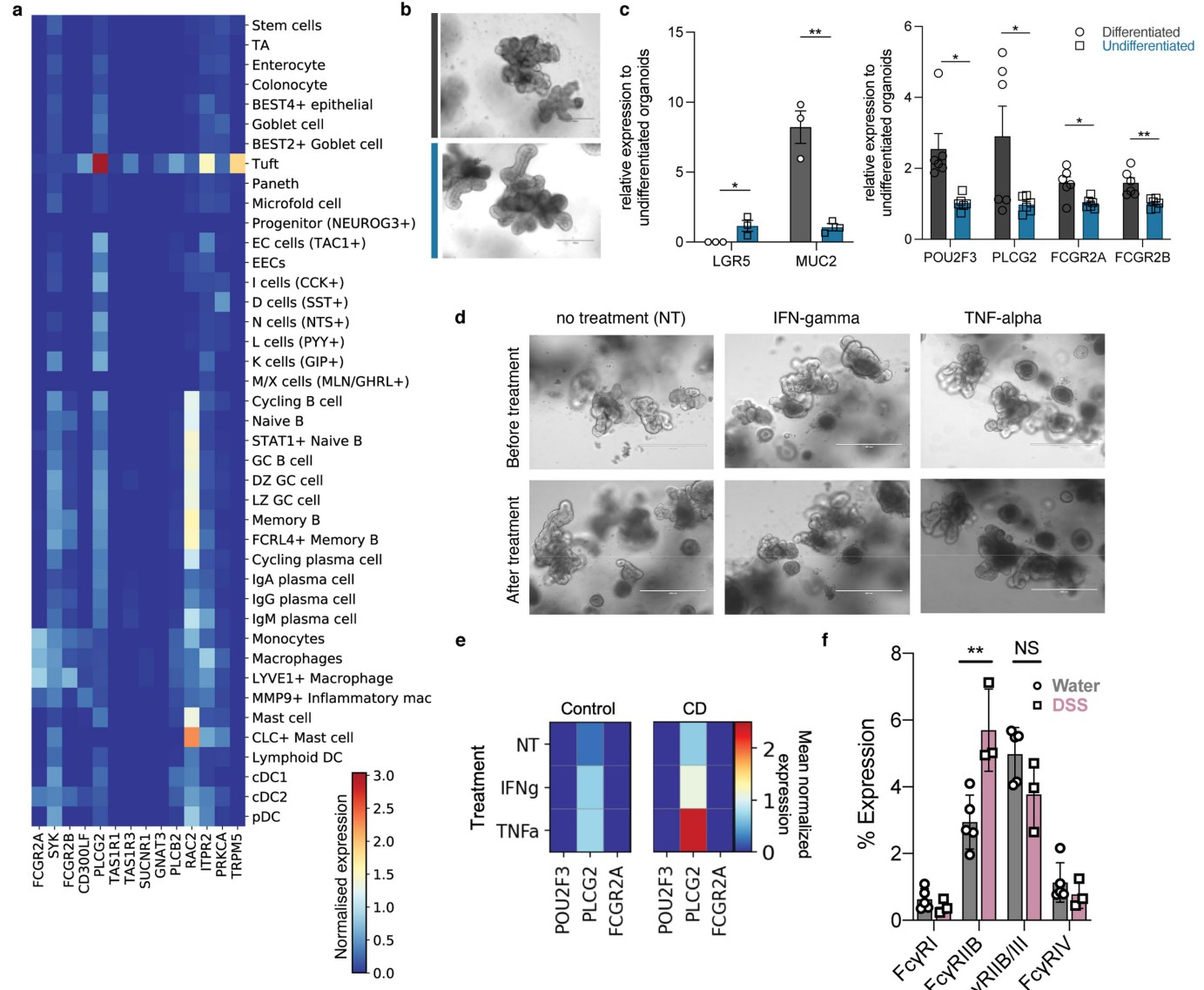

**Extended Data Fig. 7 | Tuft cells and PLCG2 activation. a**, Heat map of expression of FCGR2A and downstream signalling molecules for epithelial cells with B and myeloid cell types included for reference. **b**, Representative brightfield images of paediatric intestinal organoid line (derived from healthy donors) in culture medium (undifferentiated) or differentiation medium (differentiated). Scale bar = 400 μm. **c**, Bar plot of relative expression of *LGR5* and *MUC2* (left) (*n* = 3 from one patient) and other key mRNA (right) by intestinal organoids as in **b** (*n* = 6 from two patients). Mean with standard error of the mean (s.e.m.) is shown in bar plots, and statistics are calculated by multiple *t*-test analysis: *$P$ < 0.05 and **$P$ < 0.005. **d**, Representative bright-field images of paediatric intestinal organoid line (derived from healthy donors) without (NT) or with stimulation with inflammatory recombinant human proteins IFNγ or TNF. **e**, Heat map of normalised gene expression in scRNA-seq data of organoids from Crohn's disease (*n* = 1) and control (*n* = 3) paediatric biopsies stimulated with inflammatory cytokines as in **d. f**, Per cent expression of indicated Fcγ receptors by SiglecF⁺EpCAM⁺ small intestinal tuft cells in wild type (*n* = 5) and DSS-treated (*n* = 3) mice from a single experiment determined by flow cytometry. Mean with standard deviation is shown and statistics are calculated by multiple t-test analysis **$P$ < 0.01; NS = not significant.

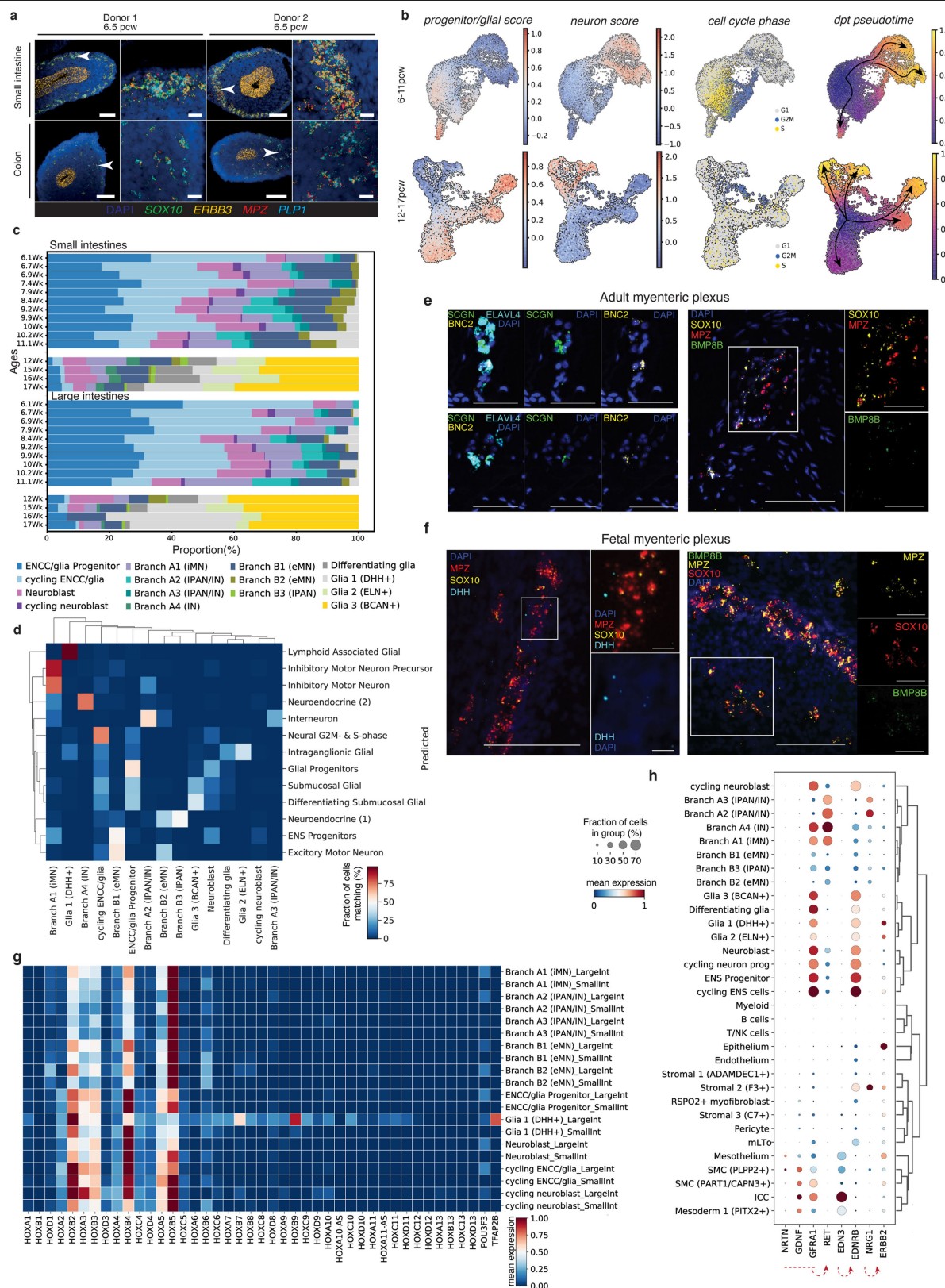

**Extended Data Fig. 8** | See next page for caption.

**Extended Data Fig. 8 | Cell types in the developing enteric nervous system.**
**a**, Multiplex smFISH images of *SOX10, ERBB3, MPZ, PLP1* expressing ENCCs in the human small and large intestine at 6.5 PCW (Scale bar panels: main = 100 µm, zoom = 20 µm, *n* = 6). *n* here and below are biological replicates across regions. **b**, UMAP visualization of neural lineage cells in 6–11 and 12–17 PCW timepoints coloured by glial or neuronal score (left), cell cycle phase (middle) or pseudotime (right). Arrows show differentiation trajectory inferred from scVelo arrows as in Fig. 3a, b. **c**, Bar plot with relative abundance of cell types among ENCC-lineage populations as described in (Figure 3a, b) across intestinal regions and developmental timepoints. **d**, Heat map showing percentage of neural cells (6–17 PCW) described in this study (columns) matching with cells described in ref. [3] (rows). **e**, Multiplex smFISH imaging of *SCGN/BNC2*-expressing enteric neurons (left, scale bar = 50 µm) and *BMP8B*-expressing Glia 2 subtype (right, scale bar panels main = 100 µm, zoom = 50 µm) in the adult sample (55-60 years, terminal ileum, n=1) and **f**, Multiplex smFISH of *DHH*-expressing Glia 1 cells (n=2, left, scale bar panels main = 100 µm, zoom = 10 µm) and *BMP8B*-expressing Glia 2 subtype (n=2, right, scale bar panels main = 100 µm, zoom = 50 µm) in the fetal myenteric plexus from 15 PCW small intestines. The boxed area is shown at higher magnification below, and n represents biological replicates across regions. **g**, HOX gene expression across neural subsets in 6–11 PCW samples. In the red box are Glia 1 (*DHH*) cells from all regions. Genes highlighted in red are colon-specific. **h**, Dot plot of key HSCR-associated ligand-receptor genes across the entire fetal scRNA-seq dataset. FPIL, fetal proximal ileum; FMIL, fetal middle ileum; FTIL, fetal terminal ileum; FLI, fetal large intestine.

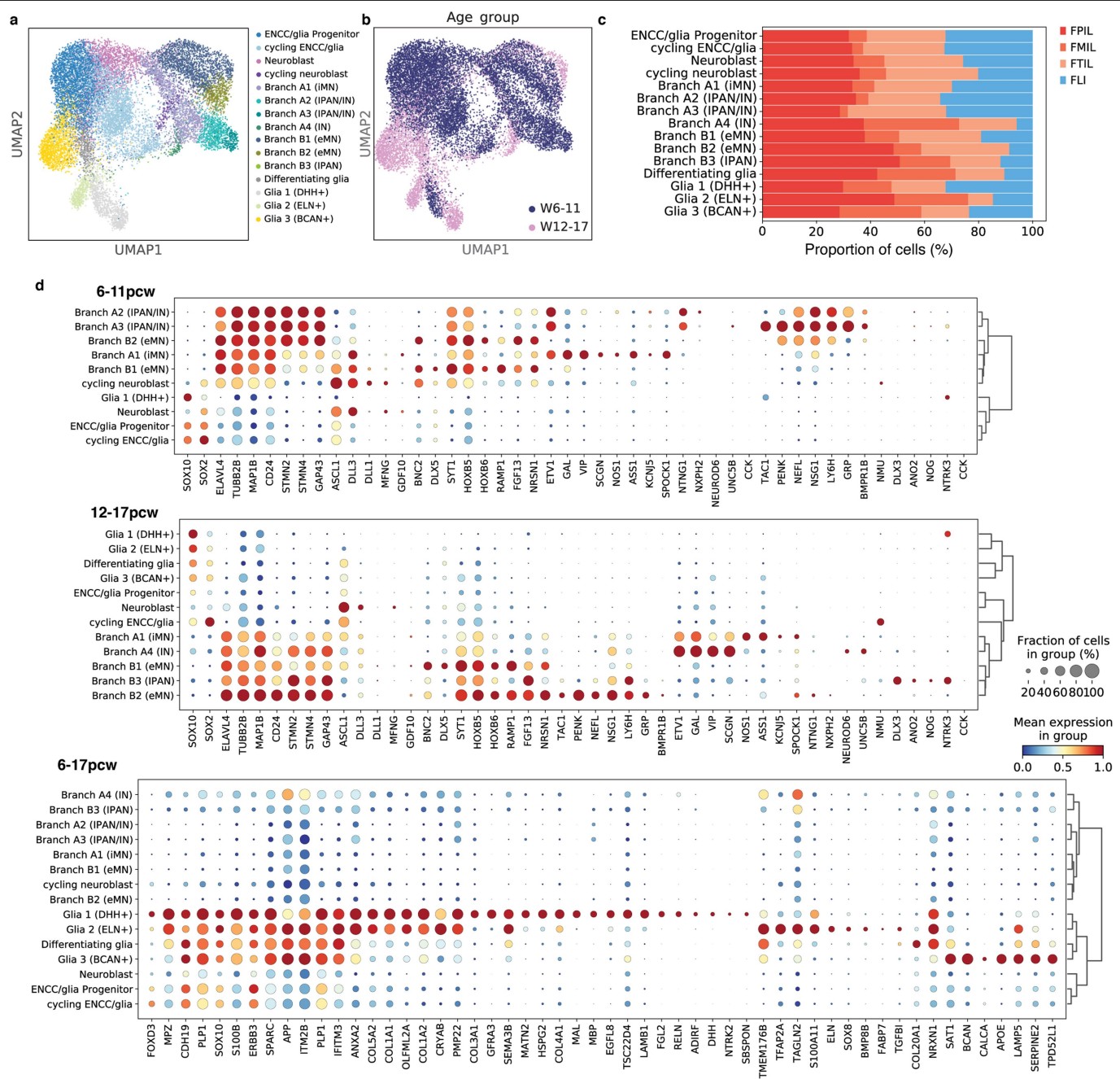

**Extended Data Fig. 9 | Annotation of developing enteric neural cells.**
**a**, **b**, UMAP visualization of neural subsets combined from 6–17 PCW coloured
by cell type annotation (**a**) or developmental stage (**b**). **c**, Bar plot showing
regional distribution of neural subsets at 6–17 PCW. **d**, Dot plots with
expression of key genes used to define enteric neuron subsets found at
6–11 PCW (above) and 12–17 PCW (middle) and glial cells across 6–17 PCW.

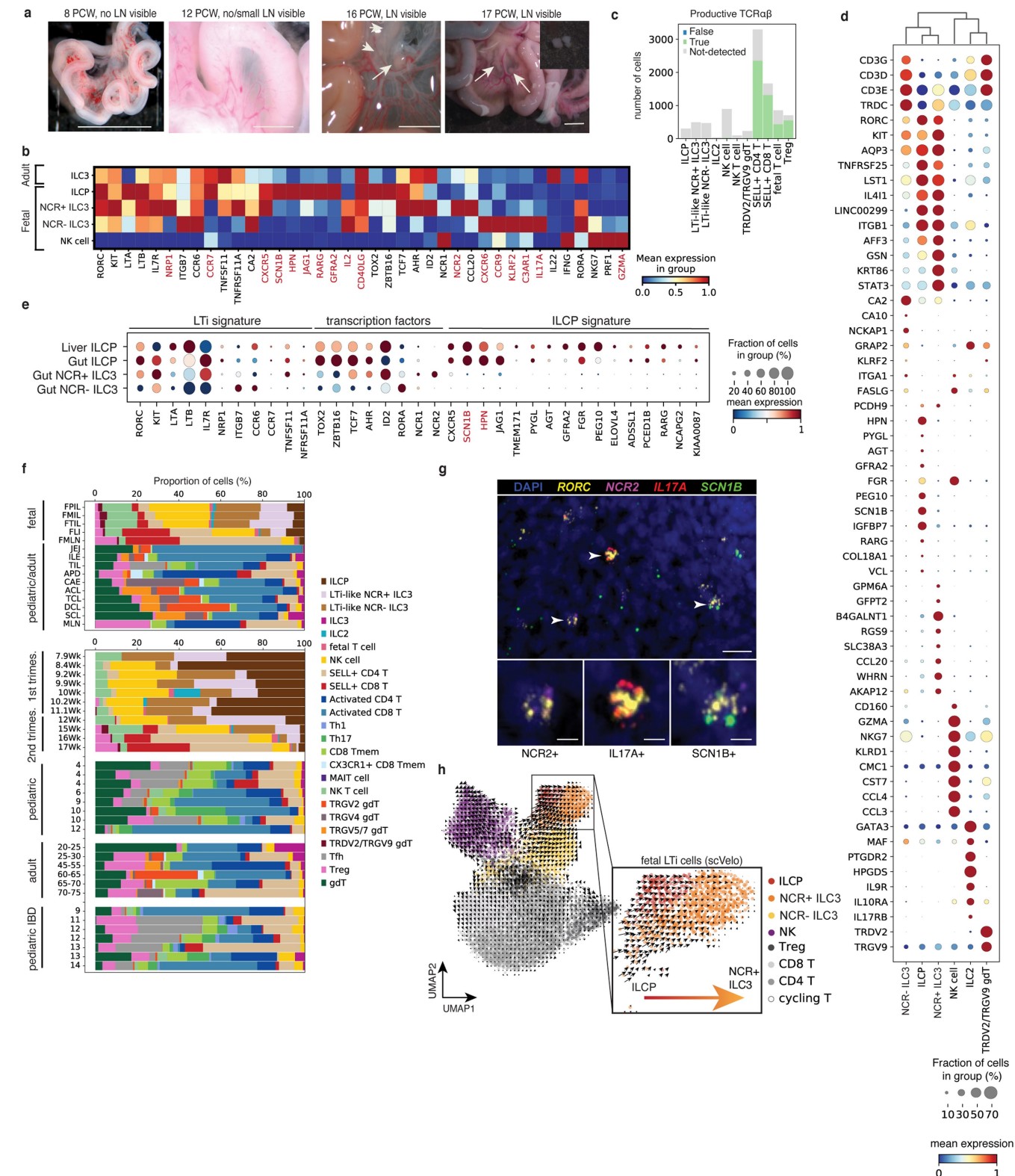

**Extended Data Fig. 10** | See next page for caption.

**Extended Data Fig. 10 | Identification of LTi-cell-like subset. a**, Photo images of human intestinal gut and developing lymph nodes (arrows) at 8–17 PCW. Scale bar = 1 cm. **b**, Heat map of relative expression of key LTi defining and marker genes expressed by LTi-like cell types, NK cells and adult ILC3 as in Fig. 4a. Genes in red are highlighted in the schematic in Fig. 4b. **c**, Bar plot with productive TCRαβ chain in fetal T and innate lymphoid cell types as determined by V(D)J sequencing paired with scRNA-seq data. **d**, Dot plot of scaled expression of selected differentially expressed genes in fetal immune subsets from scRNA-seq dataset. **e**, Dot plot of expression of selected LTi-like genes in fetal liver ILCPs compared to LTi-like subsets in the gut. **f**, Bar graph showing the relative proportion of cell types among total T and innate lymphocyte population across developmental and adult gut regions and ages. FPIL, fetal proximal ileum; FMIL, fetal middle ileum; FTIL, fetal terminal ileum; FLI, fetal large intestine; FMLN, fetal mesenteric lymph node; DUO, duodenum; JEJ, jejunum; ILE, ileum; APD, appendix; CAE, caecum; ACL, ascending colon; TCL, transverse colon; DCL, descending colon; SCL, sigmoid colon; REC, rectum; MLN, mesenteric lymph node. **g**, Representative multiplex smFISH staining of fetal ileum tissue at 15 PCW showing three LTi-like subsets: NCR2[+] ILC3, *IL17A*-expressing NCR[-] ILC3 and *SCN1B*-expressing ILCP cells (Scale bar panels: main = 20 μm, zoom = 5 μm, $n$ = 2, biological replicates across regions). **h**, UMAP visualization of fetal T and innate lymphoid cells (subsetted from Fig. 4a) coloured by cell type and overlaid with RNA velocity arrows. Inset panel shows ILCP and LTi-like NCR[+] ILC3 cells.

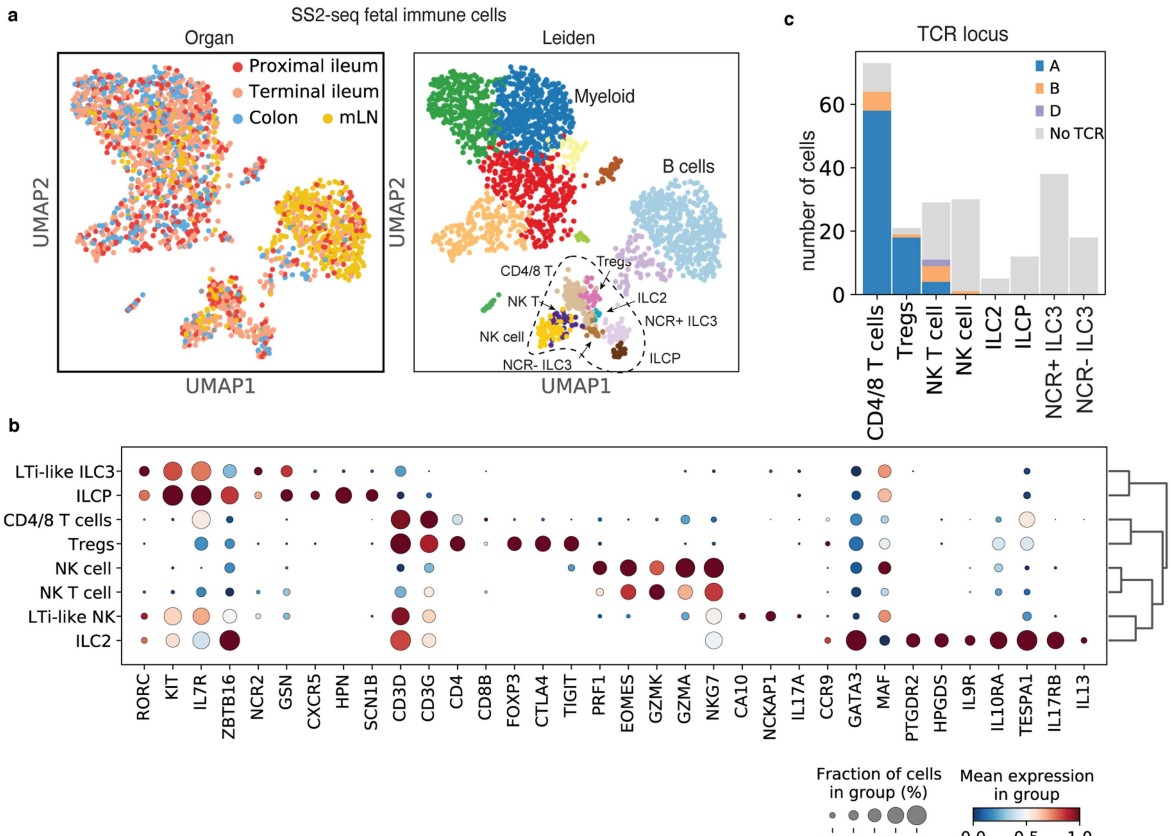

**Extended Data Fig. 11 | LTi-like cells in plate based single-cell sequencing data of sorted cells from the human fetal intestine. a**, UMAP visualization and feature plots of full-length Smart-seq2 data of flow cytometry-sorted CD45[+] cells from second-trimester fetal tissue coloured by intestinal region or Leiden clustering. **b**, Dot plot of key marker gene expression in T and innate lymphoid cell subsets captured in Smart-seq2 experiment as in **a**. **c**, Bar plot of productive TCRαβ and TCRγδ chain in fetal T and innate lymphoid cell types as in **a**.

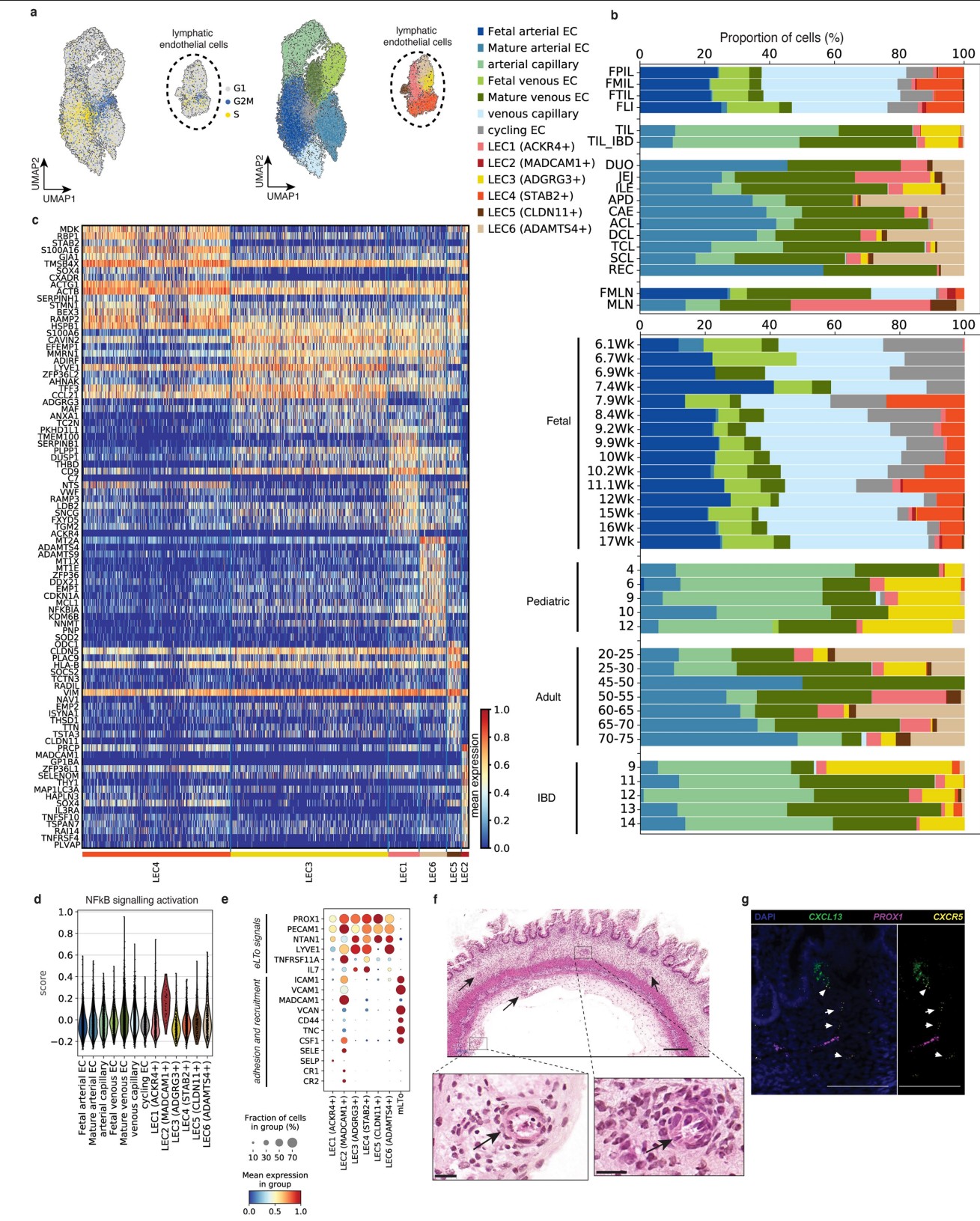

**Extended Data Fig. 12** | See next page for caption.

**Extended Data Fig. 12 | Endothelial populations in the intestinal tract.**
**a**, UMAP visualization of endothelial cell populations in fetal, paediatric and adult scRNA-seq data coloured by cell-cycle score (left) or annotation (right). Dashed line outlines lymphatic endothelial cell (LEC) subsets. **b**, Relative proportions of subtypes within total endothelial lineage separated by intestinal region (above) and intestinal region (below). Unit of age is years unless specified as weeks. Region names are: FPIL, fetal proximal ileum; FMIL, fetal middle ileum; FTIL, fetal terminal ileum; FLI, fetal large intestine; FMLN, fetal mesenteric lymph node; DUO, duodenum; JEJ, jejunum; ILE, ileum; APD, appendix; CAE, caecum; ACL, ascending colon; TCL, transverse colon; DCL, descending colon; SCL, sigmoid colon; REC, rectum; MLN, mesenteric lymph node. **c**, Heat map with top differentially expressed genes in the LEC subsets. **d**, Violin plot of NF-kB signalling activation score across endothelial subpopulations. **e**, Dot plot with scaled expression of selected genes involved in lymphoid tissue organization and immune cell recruitment amongst lymphatic endothelial cell subsets and mLTo cells. **f**, H&E staining of cross-section of fetal colon (15 PCW). Magnified panels show developing vessels (Scale bar panels: main = 200 μm, zoom = 20 μm, $n$ = 9). **g**, Representative multiplex smFISH of *PROX1* lymphatic vessels, *CXCR5* ILC3 subsets and *CXCL13*-expressing mLTo cells in the human fetal intestine at 15 PCW (scale bar = 100 μm, $n$ = 1). For **f**, **g**, $n$ represents biological replicates across regions.

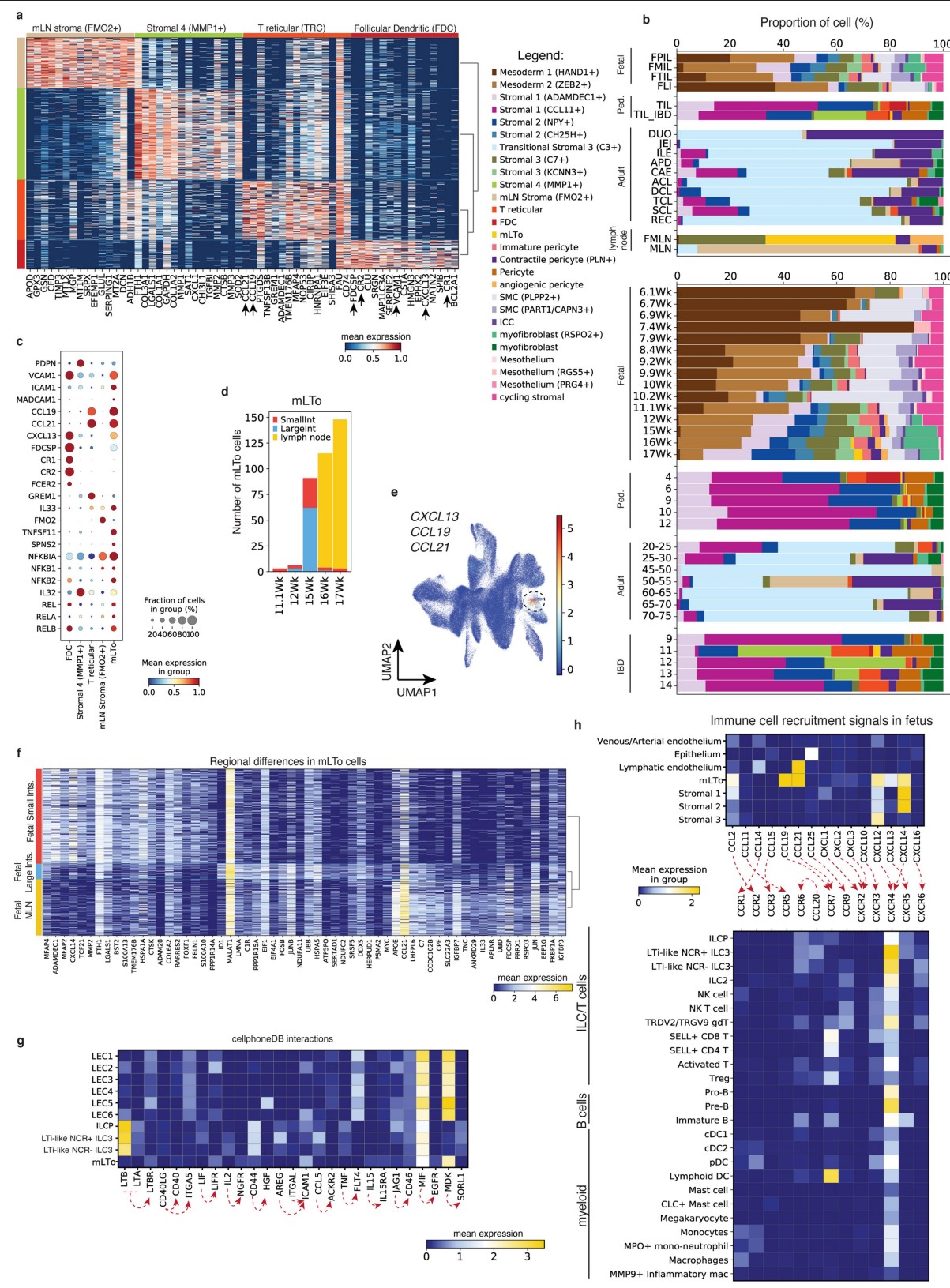

**Extended Data Fig. 13** | See next page for caption.

**Extended Data Fig. 13 | Stromal in the intestinal tract. a**, Heat map of top differentially expressed genes between follicular dendritic cells (FDCs) and T reticular cells (TRCs) and related stromal subsets. Each row is a cell. Arrows highlight key genes discussed in the text. **b**, Bar graph showing the relative proportion of cell types among the total stromal lineage across fetal and adult gut regions (top) and developmental ages (bottom). Unit of age is years unless specified as weeks. Region names are: FPIL, fetal proximal ileum; FMIL, fetal middle ileum; FTIL, fetal terminal ileum; FLI, fetal large intestine; FMLN, fetal mesenteric lymph node; DUO, duodenum; JEJ, jejunum; ILE, ileum; APD, appendix; CAE, caecum; ACL, ascending colon; TCL, transverse colon; DCL, descending colon; SCL, sigmoid colon; REC, rectum; MLN, mesenteric lymph node. **c**, Dot plot comparing key defining genes expressed across populations in **a** and fetal mesenchymal lymphoid tissue organiser (mLTo) cells. **d**, Bar plot showing number of mLTo in scRNA-seq dataset and coloured by gut region. **e**, UMAP visualization of stromal cells as in Fig. 4d showing co-expression of *CXCL13, CCL19*, and *CCL21*. **f**, Heat map of top differentially expressed genes of mLTo cells between different intestinal regions. Each row is a cell. **g**, Heat map of mean expression of ligand-receptor pairs in mLTo, LTi-like and LEC subset from scRNA-seq dataset as identified using CellphoneDB. **h**, Heat maps showing mean expression of curated immune recruitment signal genes by selected fetal stromal, epithelial and endothelial cell types (top) and their receptor expression in the immune cell types of the fetal gut and mLNs. Red arrows in **f** and **h** link cognate ligand receptor pairs.

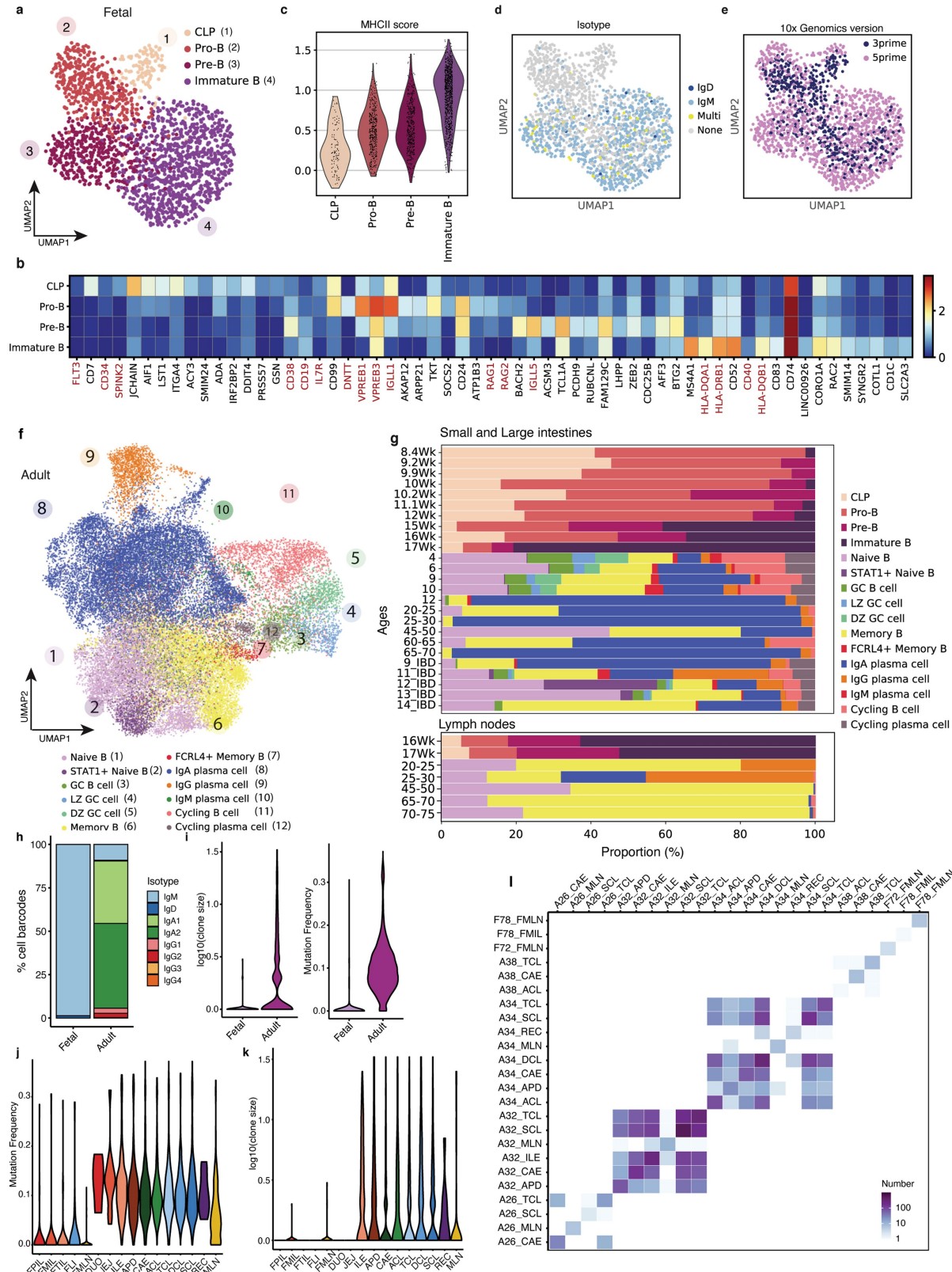

**Extended Data Fig. 14 |** See next page for caption.

**Extended Data Fig. 14 | Intestinal B cells and BCR analysis. a**, UMAP visualizations of scRNA-seq of subsetted B lineage cells from fetal samples. CLP, common lymphoid progenitor. **b**, Heat map with mean expression of differentially expressed gene in fetal B cell populations as in **a**. **c**, Violin plot of MHCII expression score of fetal B cell subsets as in **a**. **c**–**e**, UMAP visualization of fetal B lineage cells as in **a** coloured by (**d**) BCR isotype retrieved from 10x Genomics Chromium V(D)J sequencing and (**e**) 10x Genomics technology. **f**, UMAP visualizations of subsetted B lineage cells from paediatric and adult scRNA-seq samples. LZ, light zone; DZ, dark zone; GC, germinal centre. **g**, Relative proportions of subtypes within total B cell factions in the gut (above) and lymph nodes (below) separated by donor age (row) as in **a** and **f**. Unit of age is years unless specified as weeks. **h**, Estimated clonal abundances per donor for members of expanded B lineage cell clones in fetal and adult scRNA-seq datasets. **i**, Quantification of somatic hypermutation frequencies of IgH sequences from B lineage cells in fetal and adult scRNA-seq datasets as in **h**. **j**, **k**, Quantification of somatic hypermutation frequencies of IgH sequences (**j**) and estimated clonal abundances per donor for members of expanded B lineage cell clones (**k**) across fetal and adult gut regions. FPIL, fetal proximal ileum; FMIL, fetal middle ileum; FTIL, fetal terminal ileum; FLI, fetal large intestine; FMLN, fetal mesenteric lymph node; DUO, duodenum; JEJ, jejunum; ILE, ileum; APD, appendix; CAE, caecum; ACL, ascending colon; TCL, transverse colon; DCL, descending colon; SCL, sigmoid colon; REC, rectum; MLN, mesenteric lymph node. **l**, Binary count of co-occurrence of expanded B cell clones identified by single-cell V(D)J analysis shared across gut regions and donors.

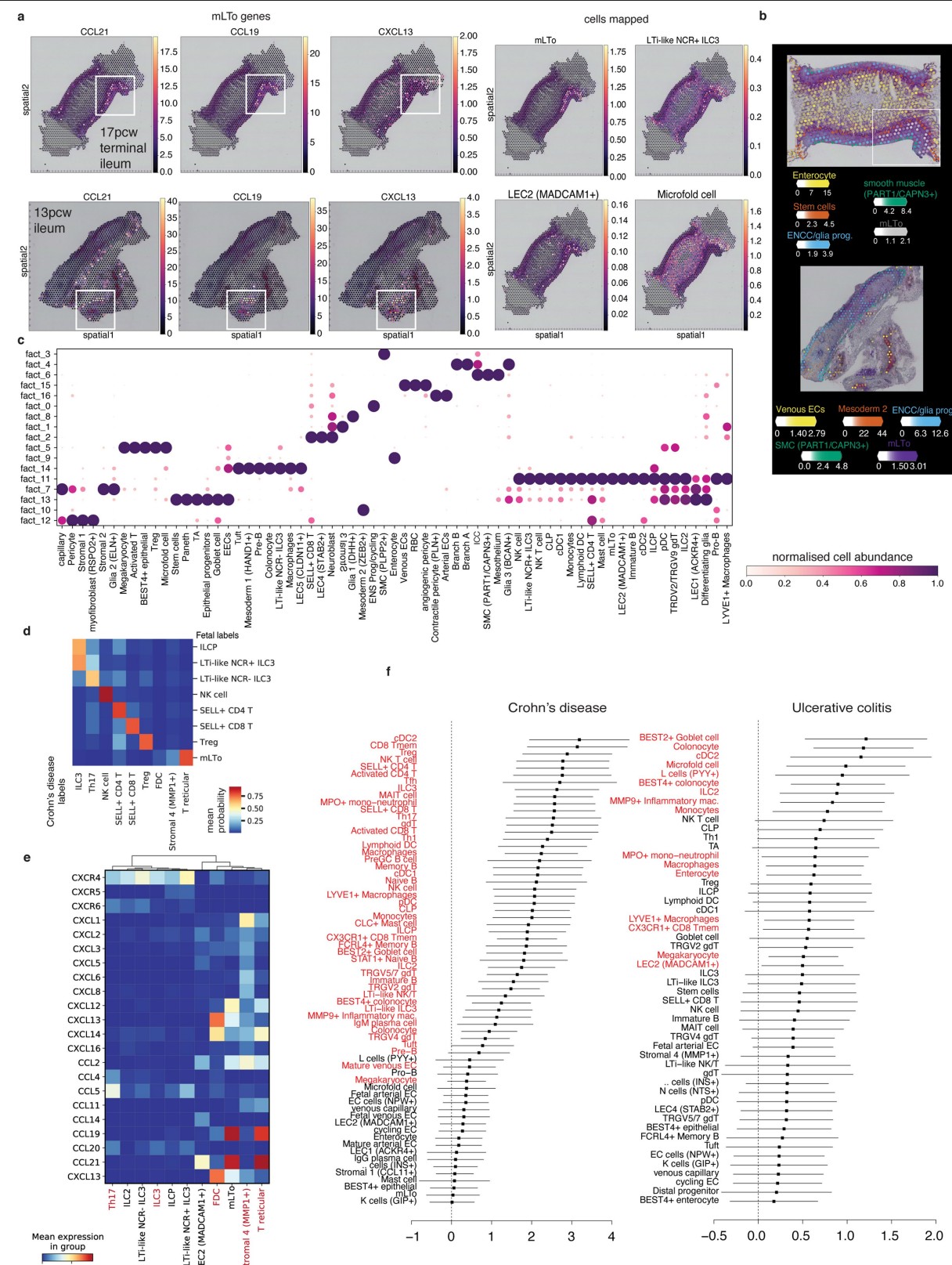

**Extended Data Fig. 15 |** See next page for caption.

**Extended Data Fig. 15 | Ectopic lymphoid tissue formation in paediatric Crohn's disease. a**, Expression of mLTo gene markers by spatial coordinates in 10x Genomics Visium data (left) and abundance of mLTo and ILC3 cells as estimated by cell2location[34] (right) across 17 PCW (top) and 13 PCW fetal ileum (bottom). White boxes highlight predicted developing SLO tissue zones. **b**, Spatial mapping of scRNA-seq data to 10x Genomics Visium data showing estimated abundance (colour intensity) of cell subsets (colour) in fetal terminal ileum from 17 PCW (top), ileum from 13 PCW (bottom). **c**, Abundances of cell types as identified using non-negative matrix factorization (NMF) in tissue zones from Visium data as in **b**. Dot plot shows NMF weights of cell types (columns) across NMF factors (rows), which correspond to tissue zones (normalized across factors per cell type by dividing by maximum values).

**d**, Heat map showing mean probability of immune and stromal cell types matching between fetal and Crohn's disease scRNA-seq datasets. **e**, Heat map with expression of cytokines and chemokines in cells involved in tertiary lymphoid organ development of fetal (black) and functionally related cell types in paediatric Crohn's disease (red). **f**, Forest plot of top cell types across fetal, paediatric (healthy and IBD), and adult data enriched for expression of genes associated with either Crohn's disease or ulcerative colitis. All cell types in red have FDR < 10%. The number of cells for each sample ($n$ = 159 samples in total with complete metadata) and coarse-grain cell type (9 different cell types in total) combination was modelled with a generalised linear mixed model with a Poisson outcome. Error bars show standard error for each factor as estimated using the numDeriv package.

# nature research

# Reporting Summary

Nature Research wishes to improve the reproducibility of the work that we publish. This form provides structure for consistency and transparency in reporting. For further information on Nature Research policies, see our Editorial Policies and the Editorial Policy Checklist.

## Statistics

For all statistical analyses, confirm that the following items are present in the figure legend, table legend, main text, or Methods section.

| n/a | Confirmed | |
|---|---|---|
| ☐ | ☒ | The exact sample size (*n*) for each experimental group/condition, given as a discrete number and unit of measurement |
| ☐ | ☒ | A statement on whether measurements were taken from distinct samples or whether the same sample was measured repeatedly |
| ☐ | ☒ | The statistical test(s) used AND whether they are one- or two-sided *Only common tests should be described solely by name; describe more complex techniques in the Methods section.* |
| ☐ | ☒ | A description of all covariates tested |
| ☐ | ☒ | A description of any assumptions or corrections, such as tests of normality and adjustment for multiple comparisons |
| ☐ | ☒ | A full description of the statistical parameters including central tendency (e.g. means) or other basic estimates (e.g. regression coefficient) AND variation (e.g. standard deviation) or associated estimates of uncertainty (e.g. confidence intervals) |
| ☐ | ☒ | For null hypothesis testing, the test statistic (e.g. $F$, $t$, $r$) with confidence intervals, effect sizes, degrees of freedom and $P$ value noted *Give P values as exact values whenever suitable.* |
| ☒ | ☐ | For Bayesian analysis, information on the choice of priors and Markov chain Monte Carlo settings |
| ☐ | ☒ | For hierarchical and complex designs, identification of the appropriate level for tests and full reporting of outcomes |
| ☐ | ☒ | Estimates of effect sizes (e.g. Cohen's *d*, Pearson's *r*), indicating how they were calculated |

*Our web collection on statistics for biologists contains articles on many of the points above.*

## Software and code

Policy information about availability of computer code

| | |
|---|---|
| Data collection | Software used include: 10x Genomics CellRanger (v3.0.0, v3.0.2) , 10x Genomics vdj (v.3.1.0 ), 10x Genomics SpaceRanger (v1.2.1 ), STAR aligner (Smart-seq2 data; version 2.5.1b). Visium spatial transcriptomics samples were aligned using 10x Genomics SpaceRanger (v1.2.1 ). 10X TCR sequences were aligned using 10x Genomics vdj (v.3.1.0 ) and Smartseq2 T cell receptor sequences were determined using TraCeR software (https://hub.docker.com/r/teichlab/tracer/). |
| Data analysis | Single cell data analysis was performed using Python (version 3), R (3.5.3), Pandas (version 0.24.2), limma (v3.46.0), NumPy (version 0.25.2), Anndata (version 0.6.19), Seaborn (version 0.11.0), and ScanPy (version 1.4 and 1.5.1). Ambient mRNA was removed using SoupX_1.4.8. Doublet removal using Scrublet (version 0.2.1). Batch correction- bbknn (version 1.3.9). TIgGER (v.03.1) , ChangeO (v.0.4.5), Shazam (v.0.1.11) were used for BCR analysis. Pseudotime calculated using scVelo (0.21) and Scanpy (1.5.1). CellPhoneDB (v2.0) was used for ligand-receptor analysis. Cell type enrichment analysis performed using miloR (https://github.com/MarioniLab/miloR). Flow cytometry data was visualised using FlowJo software (Version 10.7.0, Tree Star Inc.).GraphPadPrism 7 software was used for analysing FACS data. Additional custom codes used in this manuscript are available at Github: https://github.com/Teichlab/SpaceTimeGut, https://github.com/vitkl/fetal_gut_mapping/, https://github.com/natsuhiko/PHM. |

For manuscripts utilizing custom algorithms or software that are central to the research but not yet described in published literature, software must be made available to editors and reviewers. We strongly encourage code deposition in a community repository (e.g. GitHub). See the Nature Research guidelines for submitting code & software for further information.

## Data

Policy information about availability of data

All manuscripts must include a data availability statement. This statement should provide the following information, where applicable:
- Accession codes, unique identifiers, or web links for publicly available datasets
- A list of figures that have associated raw data
- A description of any restrictions on data availability

The expression data for fetal and adult regions is available in an interactive browsing website: www.gutcellatlas.org. Raw sequencing data are available at ArrayExpress (www.ebi.ac.uk/arrayexpress/; accession numbers E-MTAB-9543, E-MTAB-9536, E-MTAB-9532, E-MTAB-9533 and E-MTAB-10386).

# Field-specific reporting

Please select the one below that is the best fit for your research. If you are not sure, read the appropriate sections before making your selection.

☒ Life sciences   ☐ Behavioural & social sciences   ☐ Ecological, evolutionary & environmental sciences

For a reference copy of the document with all sections, see nature.com/documents/nr-reporting-summary-flat.pdf

# Life sciences study design

All studies must disclose on these points even when the disclosure is negative.

| | |
|---|---|
| Sample size | Sample size were determined by availability of donors within the sampling time-frame. No statistical methods were used to calculate appropriate sample size. We followed standards in the field and Human Cell Atlas criteria. |
| Data exclusions | No exclusion was applied to the uploaded raw data in ArrayExpress. For the final count matrix, we excluded cells based on pre-established criteria for single cells. We excluded doublets and ow quality cells. This criteria is further summarised in the Methods section. |
| Replication | Single-cell RNA sequencing was carried at on gut and lymph node tissue from 7 fetal donors and 4 adult diseased organ donors. All technical and biological replications of experiments were successful. |
| Randomization | Sample collection was based on availability of fetal and warm-autopsy donors. Since we were following developmental stage of the intestinal tract, we allocated donor samples into developmental groups based on age. |
| Blinding | This study made no comparison between discreet groups for human participants, thus binding of investigators was not necessary. For mouse studies, Tuft and non-tuft epithelial cells from the same mice were analysed, thus binding was not relevant. |

# Reporting for specific materials, systems and methods

We require information from authors about some types of materials, experimental systems and methods used in many studies. Here, indicate whether each material, system or method listed is relevant to your study. If you are not sure if a list item applies to your research, read the appropriate section before selecting a response.

## Materials & experimental systems

| n/a | Involved in the study |
|---|---|
| ☐ | ☒ Antibodies |
| ☒ | ☐ Eukaryotic cell lines |
| ☒ | ☐ Palaeontology and archaeology |
| ☐ | ☒ Animals and other organisms |
| ☐ | ☒ Human research participants |
| ☒ | ☐ Clinical data |
| ☒ | ☐ Dual use research of concern |

## Methods

| n/a | Involved in the study |
|---|---|
| ☒ | ☐ ChIP-seq |
| ☐ | ☒ Flow cytometry |
| ☒ | ☐ MRI-based neuroimaging |

## Antibodies

| | |
|---|---|
| Antibodies used | EpCAM-FITC (1:400, G8.8, Invitrogen, cat: 11-5791-82), anti-mouse CD45-Bv650 (1:200, 30-F11, BioLegend, cat: 103151), CD11b-Bv421 (1:300, M1/70, BD Biosciences, cat: 562632), Siglec- F-APC (1:200, 1RNM44N, Invitrogen, cat: 12-1702-82), FcγRI-PE (1:200, X54-5/7.1, BioLegend, cat: 139303), FcγRIIB-PE (1:200, AT130-2, Invitrogen, cat: 12-0321-82), FcγRII/RIII-PE (1:200, 2.4G2, BD Biosciences, cat: 553145), FcγRIV-PE (1:200, 9E9, BioLegend, cat: 149503) and Rat IgG2b, κ isotype-PE-Cy7 (1:200, LOU/C, BD Biosciences, cat: 552849), Brilliant Violet 650 mouse anti-human CD45 (dilution: 1:200, Biolegend, cat: 304043), Alexa Fluor 700 mouse anti-human CD4 (dilution: 1:200, Biolegend, cat: 300526), and APC-H7 mouse anti-human CD19 (dilution: 1:200, BD biosciences, cat: 560727). |

| Validation | EpCAM-FITC: Validated against TE-71 cell line compared to isotype control (Rat IgG2a K Isotype Control FITC). FACS plot shown on website.<br>anti-mouse CD45-Bv650: Staining of C57BL/6 mouse splenocytes. FACS plot shown on website.<br>CD11b-Bv421: Provided positive staining of C57BL/6 mouse bone marrow cells. FACS plot shown on website.<br>Siglec- F-APC: Validated as staining of mouse thioglycolate-elicited peritoneal exudate cells. FACS plot shown on website.<br>FcγRIIB-PE: positive staining of mouse splenocytes and compared to an isotype control. FACS plot shown on website.<br>FcγRI-PE: Validated on C57BL/6 bone marrow cells and compared to isotype control (Armenian hamster IgG PE). FACS plot shown on website.<br>FcγRII/RIII-PE: Validated by flow cytometric analysis of CD16/CD32 expression on mouse splenocytes. FACS plot shown on website.<br>FcγRIV-PE: Positive staining of C57BL/6 bone marrow cells compared to isotype control (Armenian hamster IgG PE). FACS plot shown on website.<br>Rat IgG2b, κ isotype-PE-Cy7: isotype control.<br>Brilliant Violet 650 mouse anti-human CD45: Validated in human peripheral blood lymphocytes. FACS plot shown on website.<br>Alexa Fluor 700 mouse anti-human CD4: Routinely tested and used for human peripheral blood lymphocytes. FACS plot shown on website.<br>APC-H7 mouse anti-human CD19: Validated for CD19 expression on human lysed whole blood. FACS plot shown on website. |
|---|---|

## Animals and other organisms

Policy information about studies involving animals; ARRIVE guidelines recommended for reporting animal research

| Laboratory animals | C57BL/6 mice were obtained from Jackson Laboratories (Margate, UK) and housed in specific pathogen-free conditions at a Home Office-approved facility at the University of Cambridge. Mice were maintained with a 12 hour light/ 12 hour dark cycle, with temperature ranging from 20-24°C and humidity of 45-65%. Female mice aged 10-14 weeks were used in experiments. |
|---|---|
| Wild animals | No wild animals were used in the study. |
| Field-collected samples | No field collected samples were used in the study. |
| Ethics oversight | All procedures were carried out in accordance with ethical guidelines with the United Kingdom Animals (Scientific Procedures) Act of 1986 and approved by The University of Cambridge Animal Welfare and Ethical Review Body. |

Note that full information on the approval of the study protocol must also be provided in the manuscript.

## Human research participants

Policy information about studies involving human research participants

| Population characteristics | Human fetal gut samples were obtained from the Human Developmental Biology Resource (HDBR, www.hdbr.org) and age ranged between 7-17 post-conception weeks.<br>Organ donors were either male or female and age ranged from 20-75 years. Ethnicity was not recorded, but expected to be primary Caucasian.<br>Pediatric (5-12 years; either male or female) patient biopsy material used in intestinal organoid culture. |
|---|---|
| Recruitment | Pediatric patient material used in intestinal organoid culture was obtained with informed consent from either parents and/or patients using age appropriate consent and assent forms as part of the ethically approved research study (REC-96/085).<br><br>Human adult tissue was obtained by the Cambridge Biorepository of Translational Medicine from deceased transplant organ donors after ethical approval (reference 15/EE/0152, East of England—Cambridge South Research Ethics Committee) and informed consent from the donor families.<br><br>Human fetal gut samples were obtained from the Human Developmental Biology Resource (HDBR, www.hdbr.org). The maternal consent was obtained through Newcastle hospital or through Cambridge Addenbrooke's hospital in collaboration with Roger A. Barker. |
| Ethics oversight | Procurement and study of fetal samples were approved by REC18/NE/0290- IRAS project ID: 250012, North East - Newcastle & North Tyneside 1 Research Ethics Committee and REC-96/085, East of England - Cambridge Central Research Ethics Committee.<br><br>Pediatric patient material used in intestinal organoid culture was obtained with informed consent from either parents and/or patients using age appropriate consent and assent forms as part of the ethically approved research study (REC-12/EE/0482, NRES Committee East of England, Hertfordshire and REC-17/EE/0265- IRAS project ID: 222907, East of England - Cambridge South Research Ethics Committee).<br><br>Human adult tissue was obtained by the Cambridge Biorepository of Translational Medicine from deceased transplant organ donors after ethical approval (reference 15/EE/0152, East of England—Cambridge South Research Ethics Committee) and informed consent from the donor families. |

Note that full information on the approval of the study protocol must also be provided in the manuscript.

# Flow Cytometry

## Plots

Confirm that:

☒ The axis labels state the marker and fluorochrome used (e.g. CD4-FITC).

☒ The axis scales are clearly visible. Include numbers along axes only for bottom left plot of group (a 'group' is an analysis of identical markers).

☒ All plots are contour plots with outliers or pseudocolor plots.

☒ A numerical value for number of cells or percentage (with statistics) is provided.

## Methodology

| | |
|---|---|
| Sample preparation | C57BL/6 mice received either normal drinking water or 2% (w/v) 36,000-50,000MW dextran sodium sulfate (DSS) (MP Biomedicals) to induce DSS colitis. For DSS treatment, mice received DSS water for 5 days followed by 14 days of normal drinking water, and then a final 5 days of 2% (w/v) DSS prior to being culled.<br><br>Small intestines of mice were flushed of faecal content with ice-cold PBS, opened longitudinally, cut into 0.5 cm pieces, and washed by vortexing three times with PBS with 10mM HEPES. Tissue was then incubated with an epithelial stripping solution (RPMI-1640 with 2% (v/v) FCS, 10mM HEPES, 1mM DTT, and 5 mM EDTA) at 37°C for two intervals of 20 minutes to remove epithelial cells. The epithelial fraction was subsequently incubated at 37°C for 10 minutes with dispase (0.3 U/mL, Sigma-Aldrich) and passed through a 100μm filter to obtain a single-cell suspension. Cells were blocked for 20 minutes at 4°C with 0.5% (v/v) heat-inactivated mouse/rat serum followed by extracellular staining in PBS at 4°C for 45 minutes with the following antibodies; EpCAM-FITC (1:400, G8.8, Invitrogen), CD45-Bv650 (1:200, 30-F11, BioLegend), CD11b-Bv421 (1:300, M1/70, BD Biosciences), Siglec-F-APC (1:200, 1RNM44N, Invitrogen), FcγRI-PE (1:200, X54-5/7.1, BioLegend), FcγRIIB-PE (1:200, AT130-2, Invitrogen), FcγRII/RIII-PE-Cy7 (1:200, 2.4G2, BD Biosciences), FcγRIV-PE (1:200, 9E9, BioLegend) and Rat IgG2b, κ isotype-PECy7 (1:200, LOU/C, BD Biosciences). Cells were then stained with LIVE/DEAD Fixable Aqua Dead Cell Stain Kit (Thermo Fisher Scientific) for 20 minutes at room temperature, fixed with 2% PFA, and analysed on a CytoFLEX LX (Beckman Coulter) flow cytometer. |
| Instrument | CytoFLEX LX (Beckman Coulter) flow cytometer. |
| Software | Flow cytometry data was analyzed using FlowJo software (Version 10.7.0, Tree Star Inc.) |
| Cell population abundance | EpCAM+SiglecF- cells that were defined as non-tuft epithelial cells were approximately 85% of live cells as analysed by flow cytometry. EpCAM+SiglecF+ cells made up 4%. This latter population was further assessed for CD11b cells in order to remove possible myeloid cells. Tuft cells from this gating were defined as EpCAM+SiglecF+CD11b- cells and represented approximately 94%. The purity of these populations was not determined as these cells were not sorted after analysis.<br><br>These proportions are included in the gating strategy for these populations in Extended data 4j. |
| Gating strategy | Mouse tuft cells were defined as EpCAM+SiglecF+CD11b- cells in epithelial fractions and non-tuft epithelial cells were defined as EpCAM+SiglecF-CD11b- cells in epithelial fractions. |

☒ Tick this box to confirm that a figure exemplifying the gating strategy is provided in the Supplementary Information.

