## [Peer Review File · Nature]

Manuscript Title: Cells of the human intestinal tract mapped across space and time

Editorial Notes:

Reviewer Comments & Author Rebuttals

Reviewer Reports on the Initial Version:

Referee #1 (Remarks to the Author):

Elmentaite et al. here introduce a cell type atlas of the human gut covering different anatomical regions across fetal, pediatric, and adult stages. This atlas comprises gut epithelial cells, mesenchymal and endothelial cells, secondary lymphoid organ tissue cells, as well as cells of the enteric nervous system. The authors first focus on the analysis of the epithelial compartment and describe a BEST4+ CFTR+ enterocyte population with a potential role in cystic fibrosis. Based on the analysis of IBD-GWAS genes, they further suggest a potential role of PLCG2-dependent inflammatory activation of intestinal tuft cells via FCGR2A during IBD progression. They utilize their rich single-cell RNA-seq data for an in-depth characterization of differentiation pathways of the enteric nervous system and investigate potential pathways involved in Hirschsprung's disease with impaired differentiation of the immune system. Based on the expression of disease-associated ligand and receptors they infer interactions of the neuronal lineage with stromal cells such as interstitial Cajal and smooth muscle cells, which could be perturbed in the disease context. Finally, they chart cell types driving the establishment of secondary lymphoid organ tissue during development, focusing on interactions between lymphoid-tissue-inducer and -organizer cells from the non-immune compartment. This analysis highlights distinct sub-types of ILC-lineage cells involved in this process. Furthermore, with the help of V(D)J sequencing analysis they investigate clonal expansion and somatic mutations across different regions of the gut. They utilize their developmental data to infer SLO developmental pathways that are potentially reactivated for aberrant lymphoid aggregate formation in the context of Crohn's disease.

The atlas introduced in this study represents an incredibly rich resource for investigating the dynamic cell type composition of the human gut during fetal, pediatric, and adult periods of life. The authors demonstrate with a number of examples that the atlas can be integrated with prior knowledge on disease-causing mutations and/or candidate disease-driver genes, to infer novel hypotheses on aberrant pathways and intercellular interactions involved in disease progression. This underscores the critical relevance of this resource as a basis for identifying novel therapeutic approaches to treat common intestinal malignancies such as IBD and Crohn's disease.

The quality of the data and the analysis is very high and the manuscript is well-structured and concisely written. The figures focus on the relevant information and present data in a clear and intuitive way. Crucially, the authors provide a rich web interface, which makes the data easily accessible also to non-experts and permits overlaying multiple layers of information in UMAPs such as gene expression, developmental stage, organ of origin, donor ID etc.

This resource represents a significance advance for the research and medical community. Nonetheless, I do have a number of concerns that need to be addressed by the authors:

1. Among the annotated epithelial clusters I did not see sub-clusters of SST and SCT expressing enteroendocrine cells. I would expect to observe these cells at least in the adult samples (not sure about the presence of these cells in fetal gut). In general, it would be good to annotate for which stages and which anatomical regions a given cell type can be observed. The authors should also explain in more detail how they arrived at the 103 cell types obtained by clustering. How did they determine which parameters to use for clustering (in particular, how were the resolution parameter and number of principal components selected)? Is the list of 103 cell types likely to be complete or do the authors anticipate that further resolution-increase is likely to reveal additional distinct sub-types?
2. The data shown in Figure 1F actually indicate expression of SCT in fetal enterocytes. I would expect to find expression of secretin only in hormone producing secretory cells. It is also not clear to me from the legend, if the dot plot highlights expression only for enterocytes or for all cells at the respective stage and region.
3. When the authors describe their finding of BEST4+ enterocytes across human small and large intestine (line 133), they should also mention that these cells have been described across these regions before and cite the corresponding paper (Ito et al., ref. 54), which they mention in the discussion.
4. In the discussion, the authors mention that BEST4+ enterocytes have only been described for the colon

(line 554), and they cite ref. 54 (Ito et al.). However, the study clearly describes the presence of BEST4+ absorptive enterocytes in the small intestine and colon as well, in contrast to BEST2+ goblet cells, which were only found in the colon. Did the authors also look for differential presence of these BEST2+ goblet cells across life stages and tissues?

5. It would be helpful to include a pathways enrichment or GSEA analysis in BEST4+ versus BEST4- enterocytes, and between BEST4+ cells found in small intestine versus colon. What is the function of these cells during homeostasis?

6. CFTR expression in epithelial cells could be relevant for mucus production by nearby goblet cells, for instance, by hydrating the mucus layer, and avoiding organ obstruction by a thick mucus layer, which could represent a first step of inflammatory disease. It would be interesting to investigate co-localization of BEST4+ cells with goblet cells. Could these cells have a specific role in aiding mucus production by goblet cells?

7. Is it possible to derive EEC differentiation trajectories from NEUROG3+ enriched in samples of the first trimester towards the different mature sub-types observed starting from the second trimester? Could the authors predict which pathways trigger maturation of this lineage between the two developmental stages?

8. The authors hypothesize that IgG sensing of tuft cells via FCGR2A activates PLCG2 and may affect the inflammatory response of the gut. Is there any evidence for a physiological role of PLCG2 in the context of inflammation. To complement the organoid experiment it would be informative to test whether this gene (and also Fcgr3) is up-regulated upon treatment of mice with inflammatory cytokines or in IBD mouse models (using data from published studies). With regard to the organoid experiments the authors should address whether up-regulation of PLCG2 is due to increased expression in tuft cells or other cell types, and whether FCGR2A also increases in expression upon treatment with inflammatory cytokines in tuft cells specifically.

9. Regarding the characterization of gut ILCs, I'm not convinced about the classification into ILC3s and ILCPs. I would expect up-regulation of KIT and ITGB7 in ILCPs versus mature ILC3s. I would also expect reduced levels of RORC in ILCPs versus mature ILC3s. The cells classified as ILC3s also have increased levels of ID2 compared to ILCPs and do not express IL17A which argues against their classification as mature cells. Since the cells classified as ILCPs up-regulate ZBTB16 and TCF7 I do believe that they could correspond to progenitors, however, I doubt the classification as mature ILC3s is correct for the respective cluster. Please also show additional progenitor markers from Lim et al. (2017) Cell, such as Cd7 and Il1r1. Do the ILCPs also express markers of ILC1s and ILC2, i.e., T-BET, CRTTh2, and GATA3, at low levels?

10. What is the meaning of the arrows in the UMAP of Figure 2? I could not find a description in the legend.

11. The description of the Milo method is too superficial. It is unclear to me how the authors control for overall differences in numbers and relative abundances (which might be of technical nature) between samples in integrated datasets analysed with Milo and how exactly they test for differential abundance statistically.

Referee #2 (Remarks to the Author):

Elmentaite and colleagues report here a vastly comprehensive catalogue of cells and single cell gene expression patterns, throughout different intestinal tissues and human age. This is an impressive catalogue (a word used by the authors), which the authors use to map cells involved in pathologies, such as IBD and Hirschsprung's disease. They also use this database to identify pathways that regulate Tuft cells and type 2 immunity, and the LT_i-LT_o crosstalk required for the development of lymphoid tissues during ontogeny and chronic inflammatory pathology.

In my particular field of expertise, immunology and the development of lymphoid tissues, the biology developed in the manuscript is precise and correct, and the interpretation of the data largely confirms previous knowledge. I have to confess that it is however extremely difficult, for me, to have a critical appreciation of the data, which consists mainly of scRNAseq data treatment. Therefore, given the correctness of the interpretation, and my inability to judge the quality of the data, I cannot manage to propose sensible revisions.

In sum, this manuscript is impressive in its scope and quantity of data, interpretation pertaining to my area of expertise is correct, but net advancement in knowledge is limited. This study will nevertheless provide the field with an invaluable resource.

Referee #3 (Remarks to the Author):

The manuscript by Elmentaite et al provides a new human gut cell atlas encompassing single cell RNA-sequencing analysis of fetal, pediatric and adult cell populations. 350 000 high-quality intestinal cells are included representing various gut cell types at distinct anatomical positions. The presentation focuses on 1) the epithelial compartment, highlighting distinct BEST4+ in small intestine versus colon and proposing tuft cells in the pathogenesis of IBD by IgG sensing 2) the gradual differentiation of ENS from ENCCs, highlighting presence of many glial populations and interactions with claimed relevance for Hirschsprung's disease 3) genetic programs controlling lymphoid structure formation, highlighting its reactivation during pediatric Crohn's disease.

The study forms a valuable resource for a vast range of intestinal research and gives an impressive description of the gradual development and maturation of the human intestine including associated mesenteric lymph nodes. However, as a whole, the manuscript represents a patchwork of topics, each alone with limited depth sometimes suffering from far-fetched conclusions. The manuscript would be much improved if the data-sets would be presented as a resource and speculative aspects moved to the discussion. It will important to justify how your atlas stands out and compares to the many recent publications describing the developing and adult gut by RNA-seq (see also major issue 4; e.g. Fawcner-Corbett et al., Cell 2021; Han et al., Nature 2020; Holloway et al., Cell Stem Cells, 2020; Cao et al. Science 2020). Alternatively the manuscript could be split - several impactful studies (up to 3) could potentially be made by following up each part with functional experiments.

Major Issues:

1) Presence of small intestine BEST4+ cells is claimed as a novel finding in the abstract and introduction, although BEST4+ cells have previously been reported across the human intestine including both small intestine and colon with regular immunohistochemistry (Ito et al., 2013 PLoS One). The present manuscript describes however a very interesting regional difference between the BEST4+ cells that are appropriate to highlight. Please rephrase the major finding in the abstract and introduction (perhaps focusing instead of the transcriptional difference between BEST4+ cells in colon and small intestine?).

2) PLCG2 expressing Tuft-cells:

a) The presumed link between IgG sensing Tuft cells and IBD are built on several assumptions forming circumstantial reasoning. For many IBD risk genes, the cells in which the gene acts has not been established. In the case of Plcg2, an obvious role within the immune system is shown. It is possible that the small proportion of Tuft cells are IgG sensing via Plcg2 and could contribute to disease, but convincing evidence are lacking herein. What is the evidence for that Fcgr2a can upregulate Plcg2? Is Fcgr2a expression increased in CD patients material? The suggested link is to this reviewer too weak to be highlighted in the results section. It would be more appropriate to more generally suggest signaling pathways involved in the immunological responses of Tuft cells, where Plcg2 could be one plausible gene. The possible role in IBD belongs to the discussion.

b) The presumed receptor activating Plcg2 is expressed in a very small proportion of Tuft cells (2,5%). Despite this, organoid cells upregulate PLCG2 upon TNF α or IFN γ treatment. Do you also observe an upregulation of Fcgr3 (or other ITAM receptor) following the treatment that could help boost the presumed signaling pathway? Similarly, is the receptor increased in CD patients?

The lead is interesting, consider to continue this project to substantiate the findings and separate this part into an own manuscript.

3) Analysis of the developing ENS has several issues that need to be addressed.

a) Overall Analysis: The ENS develops through a protracted time period and neuron subtype specification appears to depend on time of birth (Pham et al J, Comp Neurol. 1991, Bergner et al. J Comp Neurol 2014) as well as post-mitotic neuronal conversions in mouse (Morarach et al, 2021 Nat.Neuroscience). Additionally, it is likely that the enteric progenitor population develops with time in analogy to the gradual differentiation of neural tube progenitors to "radial glia" phenotype in the CNS, upon which Blbp amongst others markers are increase in expression. Moreover, submucosal plexus only starts to be generated by W12-14 and colonic ENS will partly consist of sacral neural crest, likely having a different expression profile compared to the vagal neural crest progenitors. Thus, to be able to investigate discrete differentiation processes, regional and temporally different data-sets will need to be analysed in separation from one another. It appears as if the vast majority of the analysis is performed with cells merged from the whole neurogenic period at all regions (thus a span of 77 days!). While an overall view with all stages should be shown and is informative, a detailed analysis of gliogenesis, neurogenesis and neuronal specification is likely hampered by inclusion of all stages/regions. A separation of for instance W6-10; W10-14 and W14-17 by the region would likely give better basis to clearly follow differentiation and branching events. This suggestion might resolve some of the specific issues identified below:

b) Committed glial precursors: It is not trivial to discern glial committed precursors from ENCCs or differentiated enteric glia. Higher levels of markers associated with adult glia would not necessary indicate glial-committed precursors (could also indicate a more mature progenitor population that is still bi-potent). It is not clear which genes are used to define committed enteric glia. If it is those specified in methods part (Sox10, Mpz, S100b, Erbb3, Plp1, Gas7, Col18a1), please refer to appropriate confirming paper or

propose a reasoning to picking these genes. The mentioned markers for this population (AGRM and SGO2) are also un-suggestive to mark glial committed precursors. AGRM is a known marker of radial glia that are multipotent (Marinero et al., 2011 PLoS One) and SGO2 has been shown to regulate mitosis (Gomez et al., EMBO Rep 2007). The expression of CCNA2 and CCNB1 also points to that this population corresponds to a specific stage during the cell cycle. A committed glial precursor population would show all phases of the cell cycle – is that the case? Moreover, your data indicate a higher proportion of committed glia precursors at early stages (Figure 4B), while it is known that neuron differentiation proceeds glial differentiation in the developing ENS (although it is much more simultaneous than in the developing CNS). As suggested above, separation of data by stage (and perhaps region) would make the interpretation easier.

c) Committed neuronal precursor: From the Velocity analysis these cells are probably committed to becoming neurons, going through their last cell-cycle, ie neurogenic. Would it be better to call them neuroblasts or neurogenic cells? These cells are expected to transiently express proneural genes such as higher ASCL1 while downregulating progenitor genes (Sox10, Sox2) and upregulating early neuronal genes (Tubb2, Pgp9.5, Elavl4). The neurogenic population has been defined in murine ENS (Morarach et al., Nat. Neurosci 2021) and it could be useful to utilize this information to annotate the human neurogenic phase (although of course some gene are likely to differ, the majority of genes are probably conserved). For instance, Mfng, Dll1, Dll3 are also markers of murine neuroblasts.

d) Annotation of Enteric Neuron Subtypes: The branches and logic of enteric neuron diversification in the mouse have recently been defined (Morarach et al Nat. Neurosci 2021). At neurogenesis, a binary event separates Etv1+ precursors (Branch A) from Bnc2+ precursors (Branch B), thus the same transcription factors observed in your human dataset. Branch A differentiate into Enteric Neuron Classes (ENC) 8-12 – corresponding to inhibitory motoneurons (Nos1/Gal/Vip), Neurod6+ putative interneurons as well as Ntng1/Nxph2 neurons that possibly could be a mix between atypical IPANs and interneurons. Branch B differentiate into excitatory motoneurons (Penk+/Tac1+), IPANs (Nmu+, Dix3+, Ano2+, Nog+, NTrk3+...), Interneuron/Intestinofugal Unc3/Cck cells and Sst+ putative interneurons. UMAP structure and Velocity analysis suggested that most neuron classes diverge at postmitotic stages by gradual conversions within the branches. Basic developmental processes are often conserved between species and authors could be helped to analyse their branching events and neuron types by consulting this framework in mouse.

The current attempts to annotate neuronal subtype is partly questionable, relying solely on the recent un-validated annotation of enteric snRNA-seq clusters (Drokhlyansky, Cell 2020) ignoring a large body of literature already present used to identify enteric neuron subtypes. The Drokhlyansky atlas refer to all CGRP+ cells as sensory, although several other neuron populations express this marker, including the cholinergic secreto/vasodilatory neurons that were expected within that atlas but not annotated. The Drokhlyansky PSN1 is likely corresponding to IPANs, while PSN2-4 would need to be confirmed (these correspond to the murine ENC7, parts of ENC12 and likely cholinergic secretomotor/vasodilatory neurons). Consider to also consult May-Zhang et al., Gastroenterology for human IPAN characterization (both with RNA-seq and tissue staining). Please note that CGRP appears mainly transcribed from CALCB rather than CALCA in the ENS.

For you annotation of possible enteric submucosal neurons, please note that this layer start to arise at around W12-14 (Fu et al., 2004. Anat. Embryol 208:33-41.) -thus your stages W14-W17 should primarily be used to annotate these neuron types. The number of neurons at these older stages appear however limited, which might make such identification difficult. The markers you indicate (Vip, Etv1, Scgn and Unc5b) are also expressed in myenteric subpopulations, why caution should be taken to annotating submucosa based on these genes. Note that murine IPANs (ENC6) express Unc5b, while Scgn is expressed in murine ENC7 (mature markers CCK/Ucn3) and ENC10 (Neurod6+). Thus, with further investigation of murine datasets you may be able to link Branch 1 to ENC6/7 and Branch 2b to ENC10. An alternative to annotate cells by function is to refer to the Class number (Enteric Neuron Classes 1-12 in the mouse myenteric plexus). Future functional experiments will be needed to faithfully annotate the neuron clusters into functional classes. If you find enough similarity between murine and human neuronal differentiation consider to use the same terminology as used in Morarach et al (Branch A, B; myenteric ENC1-12) for future clarity in the field.

e) Annotation of Glial classes: It is a hard to follow the annotation and reasoning for enteric glia and schwann cell-like clusters. All three schwann-cell like types are first defined as Schwann Cell Precursors (Row 268), while this term later in the text only denotes a subtype. It is also unclear why the markers PLP1, PMP22, CDH19 and ITGA4, CDH2 are used to define SCPs. No reference markers for human schwann cell precursors/cells that distinguish these from developing enteric glia has been published (if there is, please cite). All these markers are however robustly expressed within murine enteric glia (see: mousebrain.org – enteric glia, Zeisel et al., Cell 2018). Dhh is the only marker known to this reviewer that would faithfully distinguish schwann cell precursors from ENCCs (again this is from mouse studies including Uesaka et al., 2015).

It would be advisable to re-think the nomenclature for the glial subtypes considering the following: Myelin SCP: Are you sure that myelinating Schwann cells are present in the mesentery and/or gut wall?

Expression of Mpz is not a valid reason to call this a myelinating type as this is an early marker of schwann cells in general (Jessen and Mirsky, 2019, *Frontiers in Mol Neuroscience*). This glial type could correspond to the (unvalidated) putative SCP found in mouse ENS (enriched Mpz, Mal, Wnt6, Gfra3 and Dhh; Morarach et al, 2021). Your validation of Dhh+ cells in mesentery (fig 3) does not rule out the presence of these cells within the gut wall, please re-analyse this possibility (given that Dhh+ cell in mouse infiltrate the gut wall and give rise to neurons at postnatal stages, Uesaka et al, 2015)

Non-myelinating Schwann-like: What is the reasoning to denote this type schwann-like? The murine enteric glia consist of many subclasses (see Zeisel et al., 2018; mousebrain.org – enteric glia). These express variable levels of Eln, Sox8, Apoe, Bcan. Consider if your “non-myelin schwann-like cell” instead could correspond to a subdivision within the enteric glia

SCP: Please validate the presence of this cell type in tissue (as was performed with non-myelin Schwann cells with BMP8B). It is misleading to state enrichment in first trimester. These cells peak at W10 and are present with similar proportions at W9 as w17. Stating the age of first detection would be more informative.

In conclusion, considering no validation and functional analysis of glial populations, could they be annotated as enteric glia 1-4 in this atlas?

Does the velocity and UMAP look more crisp if only relevant stages (W9-17) are included in the analysis of glial clusters and differentiation? Are the different glial populations regionally distributed (colon versus small intestine)?

f) Acknowledgement of present literature of the developing human ENS: Several of the identified signaling molecules have previously been investigated in the developing human ENS in an immunohistochemical atlas comparing murine and human ENS transcriptional and signaling regulators (Memic et al., *Gastroenterology* 2018). It would be appropriate to use this resource as validation for expression scRNA-seq. For instance, the mentioned GDF10 as well as other TGFB-receptors Tgfr1, Tgfr2 and Tgfb2 protein expression have been detected at W6-10. The Branch 1 specific transcription factor Etv1 is also validated in human ENS within the same atlas.

A very recent paper (Fawcner-Corbett et al., *Cell* 2021) also characterizes the developing human ENS (week 12-19), please refer to this paper and compare with your own analysis.

g) Hirschsprung disease risk genes in dataset: Risk genes for HSCR are extensively investigated and have resulted in the identification of many genes that can explain the aganglionic phenotype (replicated in murine models). RET, GDNF, NRTN, SOX10, EDNRB, EDN3, ECE1, ZFH1B, PHOX2B, KIAA1279, TCF4 are major HSCR risk genes. Studies have also linked SEMA3A, MAPK10, NRG1, SOX2, BACE2 and GFRA1 (Tang et al., *Gastroenterology* 2018). Note that genes involved in ENS development have been investigated as candidates in many studies, where some have been linked (HOXB5) - while others have been disproven to contribute to HSCR, including ASCL1 (Carter et al., 2012 *J Hum Genet.*) Please limit your annotation to confirmed HSCR-disease linked genes and do not include own candidate genes (remove ASCL1, TLX2 and possibly others). Carefully review your HSCR risk genes. The illustration in major Figure 4G is questionable as it fails to illustrate the fine validation provided by this paper of already suspected transcription factors, cell-interactions and receptor/ligand interactions and instead high-lights new interactions that are not substantiated by experiments nor warranted from unexplained phenotypes in HSCR. Instead, for the HSCR field to gain the most from your resource, please make a Table similar to Fig 4f or Sup 4l with all significant HSCR risk genes. This could identify new leads to be followed up by experiments. It is for example very interesting that Ret is higher expressed in the colon than in the small intestine. Note that ERBB2 and NRG1 variants are only found in very few HSCR patients and that there are other sources of NRG1, including sub-epithelial mesenchyme (Holloway et al., *Cell Stem Cell* In Press) and smooth muscle cells and fibroblasts (Ementaite et al., *Developmental Cell* 2020). It is also fine to include genes previously implicated in ENS development either in the same or separate table, but they could be denoted “critical for ENS development” or something similar.

It is however true that patients do experience life-long complications including enterocolitis (Gosain et al 2016 *Pediatr Surg Int*) and an increased risk to develop Crohn's disease (Granström et al., 2017 *JPGN*). Hence, there are aspects about HSCR that we still do not understand that possibly extends beyond ENS differentiation and colonization. However, the “new” cell-cell interactions indicated by receptor-ligand matching do not seem to reach these gaps in knowledge, are highly speculative, not backed up with functional evidence or HSCR tissue stainings. Your conclusions that the analysis implicate fetal branch 2b and ICC/SCM interactions in pathogenesis of Hirschsprung disease (results) and that your high resolution genomic definition uncovers new insights into both rare and common diseases (in the introduction), represents two statements that are exaggerated. Please refrain from too much speculation. It is enough to show the gene expression of HSCR genes in your different cell populations and conclude that they are expressed more wide-spread than previously appreciated (or in novel cell populations). Please amend figures, introduction and results accordingly.

h) Unnecessary speculation: Page 7 row 326: Authors identify NTrk3 expression in an glial population (denoted pro-myelinating schwann cells) and try to link this expression to the previous observation of reduced IPANs in NTrk3 or Nt3 mutant mice. However, Ntrk3 is expressed in IPANs (Morarach et al., 2021), so a much more straight-forward explanation would be the intrinsic expression of this receptor. Please remove this speculation.

4) The authors admit that extensive single cell sequencing of human gut tissue at different stages has collectively been carried out in recent time and indicate that they intend to provide a holistic analysis. As such it would be very helpful for the field if the authors could put their new atlas into perspective of recently published studies. For instance, how does the human gut RNA-seq dataset and analysis of Han et al., 2020 Nature; May-Zhang et al., Gastroenterology 2020; Drokhlyansky et al., 2020; Holloway et al., Cell Stem Cell; 2020; Cao et al., Science 2020 and Fawkner-Corbett et al., Cell 2021 compare to the current manuscript?

Minor issues:

1) At many location, you refer to the ENS as "Neuronal" (Figure 1, Supplementary Figure 2, Results, Introduction). It is more appropriate to use the term "Neural" as this encompasses both neuronal and glial types/fates. Please change Neuronal to Neural when you refer to the whole ENS.

2) Figure 1A. Labelling of sampled areas on fetal tissue only states ileum. The proximal and mid ileum appears to be placed in the fetal duodenum and jejunum. Please correct sampling area or labelling of area.

3) P52; row 168. 3 is missing in NEUROG+

4) Supplementary Figure 1b. The label indicates presence of Glia in red, however no such cells are discernable in the figure

5) Page 6; Row 265: It is questionable to refer to terminal glia markers Plp1, Sox10, S100b and so as "neural crest precursor genes". Perhaps it is enough to define these markers as terminal glial markers without referring to neural crest.

6) Page 6 Row 242: What do you mean by "a shift" in ENS development? Why did such shift enable ENCC investigation?

7) Bnc2 is misspelled in results and figure legend 4.

8) Abstract: It is enough to say that you define glial and neuronal cell populations in the developing enteric nervous system (no need to say novel – most are already defined in the mouse). If you want to use novel, only the glial subsets appear truly novel and previously not described.

9) Page 3; Row135: Reference should be Figure 1E,F.

10) Page 5; Row 190: Rephrase "IBD pathogenesis is due to", you probably mean "IBD pathology is due to", or "IBD pathogenesis involves impaired intestinal...."

11) There are duplicated papers in the reference list (for instance Drokhlyansky)

12) It would be helpful with a table indicating the number of cells in each sample type and the 103 cell types/states (also indicating no of cells from each stage/region).

Good Luck!

Referee #4 (Remarks to the Author):

This very broad-ranging, highly ambitious study examines fetal, pediatric and adult gut and lymph node tissues at single cell resolution, including 11 distinct intestinal/secondary lymph node locations. Strengths of the study include its "time and space" analyses, providing invaluable single cell insight into gut development. The sharing of data and code through the Human Cell Atlas and will provide an enormously useful resource for the community. The time-series analyses provide an important re-definition of gut development at the single cell level. This is effectively leveraged for advancing Hirschsprung's disease (aganglionic megacolon) pathogenesis, a developmental disorder, but somewhat less so for inflammatory bowel disease (IBD).

Major

1. Consistency and developmental differences in sampling capture. Supplementary Table 1 should include cellular yields for each of the samples (EPCAM+/CD45-, CD45+EPCAM-) to provide insight into the consistency, replicability and robustness of findings. This is particularly important as many of their figures with respect to time-series changes may reflect only one sample (would be important to clarify). The description of the pediatric ileal tissues is not described beyond enteroid recovery; however, additional pediatric tissue results are reported beyond epithelial cells. The description of the fetal and adult tissue cell isolations implies that full-thickness digestion was performed; given their sorting strategy, how do the authors interpret the marked relative mesenchymal enrichment observed in fetal tissues, relative to post-

natal tissues (Figure 1C)? Full-thickness digestion and differences between fetal and post-natal processing and cell capture efficiencies will have marked differences in the interpretation of many items in Figure 1.

2. Figure 2A-B is a highly interesting time-series analyses of relative cell fractions, which effectively leverages the strength of this study, namely the time-series fetal datasets. As classified presently, there are marked small vs. large intestinal differences prior to the "developmental shift" defined in Fig 2B as between 10-12 weeks. For example, for the large intestine, the "crypt" clusters" arise quite abruptly after the "shift" which doesn't necessarily make sense, as large intestinal crypts include both enterocytes and goblet cells. Also, LGR5 and other stem cell markers, should anchor the crypt development descriptions. Ideally, the empiric serial fetal fractions (Fig 2B) could empirically justify the time-series predictions more explicitly (Fig 2A), especially with respect to the secretory vs. transit-amplifying/enterocyte dichotomy. Missing in the large intestinal analysis is a description of the Paneth cell equivalents, described by some groups as REG4+ cells. Finally it is well known that even within small or large intestine, marked cellular composition differences exist between regions (proximal to distal), so in text and/or more concise summaries in supplementary materials (e.g. UMAPs by regions with small or large intestine) should be provided.

3. Figure 2E-G: The findings of immune receptors, especially for IgG on Tuft cells is potentially quite important, but with the present datasets, the IBD connection is quite speculative; lines 190-198 and 226-239 are not particularly informative. Rather, I interpreted pre-natal presence in the context that IgG uniquely can traverse the placenta, thereby providing passive immunization generally and potential to activate Tuft cells. Post-natally, tuft cells mediate type 2 immunity, and Crohn's disease is clearly a type 1 response; the modest increase in PLCG2 (Supp Fig 3E) is not a particularly compelling/strong tie-in to IBD pathogenesis. I could not understand what points were trying to be made in Fig 2D with the Crohn's disease vs. ulcerative colitis comparison, and deleting these sections should be strongly considered, especially as this general tie-in is alluded to in their Developmental Cell 2020 manuscript.

4. The application of their datasets to Hirschsprung's disease genes (e.g. Ret, EDNRB) provides a fascinating glimpse into disease pathophysiology with interesting validation by RNAscope (Fig 3D) at 15 PCW (the failure of ganglion cell migration is believed to be prior to this timepoint). However missing in Figure 3 is a time and space summary of this migration defect (proximal to distal). At the earlier fetal timepoints, can RNAscope be applied to track proximal-to-distal migration of ganglion cells? From a translational perspective, short segment (vs. extensive) Hirschsprung's only affects the distal rectum, so it would be interesting to track a bit more precisely the timing of colonic innervation.

5. Figure 4. The focus on ILC populations is merited, especially given the connection to IBD GWAS genes in Figure 5G. However, no mention is made of ILC2 or ILC1 cells (Yudanin Immunity 2019). Were they not observed? There is a literature that ILC1 cells can derive from ILC3 cells over time, and certainly in IBD. Figure 4E is a lovely juxtaposition of CXCL13 (from follicular dendritic cells?), CXCR5 and RORC (LTi-like ILC3). Would be highly informative to provide a corresponding RNAscope co-segregation analysis for the LTi-like NK/T cells, especially given the anti-inflammatory nature of IL17A in Crohn's disease. Along these lines, it would be illuminating to also provide gene expression values for IL22 (epithelial protective) and IFNG (multiple cellular sources).

6. Fig 5. While it is logical to conclude the manuscript with gut to MLN connections, as presently formulated, compelling, novel and crisp conclusions are not provided. Regarding Fig 5G, not sure why they chose to test cellular enrichment for Crohn's disease as opposed to ulcerative colitis, where the roles/impact for leukocyte trafficking is stronger for the latter. While Figure 5G clearly shows substantial overlap in effects between different cell types (likely all cells contribute to some extent) the text concludes by emphasizing ILC3 cells (again logical), at the expense of Treg cells, with the latter having the highest enrichment. The role for differential expression in single cell datasets as a means of implicating cell subsets is logical, but added rigor would be attained by applying a complementary datasets (Encode, transcriptionally active regions, e.g Farh et al., Nature 2015). The Farh et al., manuscript implicates Th17 cells and not B/plasma cells particularly. Given the sparse transcriptome, it is quite possible that single cell approaches lack the sensitivity to identify key disease causing genetic variation.

7. More generally, the discussion should include caveats about the lack of sensitivity of the scRNASeq approaches. Even if scRNASeq co-localizes Best4 and CFTR, validation of an exclusive co-localization by protein staining (Supp Fig 2F only shows CFTR) is not provided.

Minor:

1. Figure 2A: did the authors mean to label the upper left-hand cluster "EEC" as opposed to "ECC"?
2. 4 pediatric samples listed in Figure 1A, yet multiple subsequent figures allude to 5 pediatric samples.
3. Figure 5F. Can labels more informative than "Stromal 1", "Stromal 2" be provided, without having to refer to supplementary materials.
4. Supp Figure H: were pediatric samples tested here for comparison?
5. The SARS-Cov19 data (ACE2, TRMPSS) is quite interesting and important, but sensitivities with respect to stem cell infections in utero should be considered.
6. Line 180: Given the absence of third trimester data, it is a bit misleading to say that Paneth cells

appear post-natally.

7. Line 257: DLL3 (vs. DDL3) typographical error?

8. Lines 578-9. The enteric nervous system has substantial effects beyond peristalsis.

Author Rebuttals to Initial Comments:

We thank the editor and reviewers for their time in considering our manuscript. In response to the reviewers' comments, we have performed additional analyses, which we feel have substantially improved the manuscript.

Additional work includes:

1. Integration of additional fetal (donor n=2 and tissue samples n=5) and adult (donor n=2 and tissue samples n=22) single cell RNA sequencing data.
2. 10x Genomics Visium spatial analysis of fetal intestinal tissues.
3. Additional smFISH staining to validate ILC and neural cell populations and clarification of nomenclature of enteric nervous system cells.

A point-by-point response to reviewers' comments is detailed below. Additional text is highlighted in blue and revised changes are highlighted in red.

Reviewer #1:

1.1	Among the annotated epithelial clusters I did not see sub-clusters of SST and SCT expressing enteroendocrine cells. I would expect to observe these cells at least in the adult samples (not sure about the presence of these cells in fetal gut). In general, it would be good to annotate for which stages and which anatomical regions a given cell type can be observed. The authors should also explain in more detail how they arrived at the 103 cell types obtained by clustering. How did they determine which parameters to use for clustering (in particular, how were the resolution parameter and number of principal components selected)? Is the list of 103 cell types likely to be complete or do the authors anticipate that further resolution-increase is likely to reveal additional distinct sub-types?	We thank the reviewer for their constructive comments on the manuscript. After further inspection of enteroendocrine cells, we observe and annotate a SST+ enteroendocrine cluster corresponding to D cells (Figure R1) and include these results in the revised Figure 2e. SCT expression was observed across mature enteroendocrine clusters as well as in all fetal epithelial cells (Figure R2).
-----	---	---

Figure R1: UMAP visualisation of EECs from fetal, pediatric and adult datasets with overlaid expression of key defining genes.

Figure R2: Dotplots showing expression of EPCAM and SCT in the fetal and ped./adult epithelial subsets.

As requested, we have also included a table with the summary of the number of cells in each stage and anatomical region for each cell type in the revised manuscript (Table 2).

Identifying cell clusters in a single-cell experiment is challenging because there is no one correct clustering for any dataset, and no gold-standard way to select a single best clustering. Algorithms for inspecting and selecting resolution of clustering exist (e.g. Clustree), however with a large number of clusters, interpretation becomes difficult.

We have chosen to annotate each sample group separately (fetal, pediatric and adult), choosing the best parameters within each group. In order to determine clustering parameters, we inspected the total variance of each cell lineage (epithelial, mesenchymal ect) by looking at top 50 PCs. We included up to 50 PCs depending on whether the resulting clustering revealed meaningful cell types. After annotation of each dataset separately, we then integrated the datasets together and inspected the clustering of annotated cells. The annotations were

		manually adjusted based on clustering between datasets. We observed that some clusters, especially in the fetal dataset, represented differentiating subsets (e.g. pericytes) and avoided over-annotation in such cases. In addition, given the size of our dataset we are able to distinguish rare subsets of cells not detected in other single cell RNAseq datasets (e.g. enteroendocrine subsets, Microfold cells). However, it is also possible that not all cell types were captured in this data due to technical reasons (e.g. we did not capture adult enteric neurons due to difficult dissociation of neurons). We anticipate that further integration with new data modalities, such as scATAC-seq and spatial methods will reveal additional cell types and states beyond those currently described, especially in lineages with less well described cell types/states (e.g. mesenchymal subsets).
1.2	The data shown in Figure 1F actually indicate expression of SCT in fetal enterocytes. I would expect to find expression of secretin only in hormone producing secretory cells. It is also not clear to me from the legend, if the dot plot highlights expression only for enterocytes or for all cells at the respective stage and region.	Figure 1f shows the expression of selected marker genes across BEST4+ cells in different anatomical regions. We have now clarified this in the figure legend: “f) Dotplot with relative expression of selected marker genes within BEST4+ epithelial cells from small and large intestines and different ages.” Thank you for highlighting the SCT expression in fetal enterocytes. Secretin, encoded by the SCT gene, has been reported to be widely expressed in the mouse developing enterocytes ¹. Similarly, we observe SCT gene expression across all epithelial subtypes of the developing human intestine (Figure R2). In adults, the SCT expression becomes restricted to the enteroendocrine lineage.

1.3	When the authors describe their finding of BEST4+ enterocytes across human small and large intestine (line 133), they should also mention that these cells have been described across these regions before and cite the corresponding paper (Ito et al., ref. 54), which they mention in the discussion.	Thank you, we have now included this reference (as ref. 11) in the results section as follows: “Notably, BEST4+ epithelial cells, previously observed in human small and large intestines ² and predicted to transport metal ions and salts^{3,4}, varied in abundance between intestinal regions <..>”
1.4	In the discussion, the authors mention that BEST4+ enterocytes have only been described for the colon (line 554), and they cite ref. 54 (Ito et al.). However, the study clearly describes the presence of BEST4+ absorptive enterocytes in the small intestine and colon as well, in contrast to BEST2+ goblet cells, which were only found in the colon. Did the authors also look for differential presence of these BEST2+ goblet cells across life stages and tissues?	We indeed find a subset of goblet cells that is specifically marked by BEST2 expression and is present in the large intestinal tissues (Figure R3a-b). We have now included these results in the revised Figure 2b and Supplementary 3g-i. Pediatric/adult BEST2+ cells clustered apart from BEST2- goblet cells (Figure R3c), but shared expression of goblet cell characteristic genes, including MUC2, CLCA1 and SPDEF (Figure R3d). Amongst the genes enriched in BEST2+ goblet cells were Kallikreins KLK15 and KLK3, as well as protease inhibitors WFDC2 and WFDC3 (Figure R3d). However, the majority of the differentially expressed genes reflected their colonocyte or enterocyte identity (e.g. CD177 and LYPD8 were expressed in colonic goblet cells, while APOA1 and RBP2 in small intestinal goblets; Figure R3e). Amongst fetal epithelial cells we found 43 large intestinal cells that expressed BEST2 (Figure R3f-g). Fetal BEST2-expressing cells clustered together with small intestinal cells (possibly due to the small number of these cells present in the data), but showed expression of genes defined using adult BEST2+ Goblet cells, including KLK15 (Figure R3h).

Figure R3: BEST2+ goblet cells in fetal and adult epithelial cells. a) Feature plots with expression of *MUC2* (indicating goblet cells) and *BEST2* in pediatric/adult epithelial cells. b) Barplot with number of goblet cells captured across intestinal tissues. c) UMAP visualisation of pediatric/adult epithelial subsets and their regional distribution. d) Dotplot with genes that correlate with *BEST2* expression calculated using Jaccard Similarity measure. e) UMAP visualisation of regional distribution of fetal epithelial cells. f) Feature plots with expression of *MUC2* (indicating goblet cells) and *BEST2*. Arrow points to 43 BEST2-expressing goblet cells. g) Dotplot of genes as in d) and additionally *DEFA5* and *DEFA6* specific to Paneth cells.

1.5	It would be helpful to include a pathways enrichment or GSEA analysis in BEST4+ versus BEST4- enterocytes, and between BEST4+ cells found in small intestine versus colon. What is the function of these cells during homeostasis?	We thank the reviewer for this suggestion. We ran GO term enrichment analysis on genes from Milo⁵ analysis of BEST4+ enterocytes from adult small versus large intestines and vice versa. Among GO terms for BEST4+ enterocytes of the adult small intestine are “Cotranslational protein targeting to membrane”, “Translation initiation”, “Viral gene expression” and biosynthesis of organic acid, monocarboxylic acid and fatty acid (Figure R4; top). In the large intestines these cells upregulated genes associated with “Response to oxygen levels”, “Cellular carbohydrate metabolism process”, “ATP metabolic process”, “Purine containing compound metabolic process” and transport of inorganic anion, chloride and sodium ion (Figure R4; bottom). These terms suggest a major functional difference of these cells between intestinal regions is biosynthesis of acids in the small intestines versus metabolism of small molecules in the large intestine. We include this analysis in Supplementary Fig. 3f and highlight these findings in the text of the revised manuscript and here for your convenience: “Gene ontology term enrichment analysis of genes differentially expressed by BEST4+ cells between adult small and large intestines suggests that further major distinction in the function of these cells is biosynthesis of acids and metabolism of small molecules, respectively (Supplementary Fig. 3f).”
------------	--	---

		Figure R4 displays two dot plots showing Gene Ontology (GO) terms from genes upregulated in BEST4+ enterocytes from adult small (top) versus large (bottom) intestines as determined by Milo analysis. The top plot, titled "Up in Smallint", shows GO terms on the y-axis and $-\log_{10}(qvalue)$ on the x-axis (ranging from 10 to 40). The bottom plot, titled "Up in Largeint", shows GO terms on the y-axis and $-\log_{10}(qvalue)$ on the x-axis (ranging from 2.5 to 3.5).
1.6	CFTR expression in epithelial cells could be relevant for mucus production by nearby goblet cells, for instance, by hydrating the mucus layer, and avoiding organ obstruction by a thick mucus layer, which could represent a first step of inflammatory disease. It would be interesting to investigate co-localization of BEST4+ cells with goblet cells. Could these cells have a specific role in aiding mucus production by goblet cells?	We appreciate this suggested link between BEST4+ enterocytes and goblet cells for hydrating the mucus layer of the intestines. We would like to highlight findings by Ito and colleagues in PLOS 2013 figure 1C, in which Best4+ cells are visualised next to cells with a goblet cell morphology. In addition, we show examples of such co-occurrence in images from Protein Cell Atlas (Figure R5). We have now referenced this in the revised manuscript and below: “Interestingly, small intestinal BEST4+ cells were marked by high expression of chloride channel and cystic fibrosis gene CFTR (Figure 1f), which we also observed at the protein level (Supplementary Fig. 3d-e).

		Previous work has placed BEST4+ cells in close proximity to cells resembling goblet cells². This highlights a possible role of BEST4+ enterocytes in aiding mucus production by goblet cells specifically in the small intestines.”  Figure R5: Proximity of goblet and BEST4+ cells in the human small intestine. IHC staining of BEST4 and CFTR in small intestinal sections (proteincellatlas.org). Black arrows point to cells with goblet cell morphology and red arrows point to cells expressing either BEST4 or CFTR.
1.7	Is it possible to derive EEC differentiation trajectories from NEUROG3+ enriched in samples of the first trimester towards the different mature sub-types observed starting from the second trimester? Could the authors predict which pathways trigger maturation of this lineage between the two developmental stages?	Subclustering of EECs revealed NEUROG3+ precursor cells and multiple mature subsets resembling populations recently described in intestinal organoid experiments, including M/X cells (MLN/GHRL+), D cells (SST+), L cells (GCG+), N cells (NTS+), K cells (GIP+), I cells (CCK+) and enterochromaffin (EC) cells (TPH1+) either expressing Neuropeptide W (NPW+) or TAC1+ (Figure R1 above and R6). NEUROG3+ progenitors were enriched in the first trimester, while hormone expressing subsets were instead enriched in the second trimester samples (Figure R1). In addition, we capture a small subset of insulin expressing β cells (INS+) in the first trimester proximal intestinal gut, possibly contaminating from developing pancreatic bud (Figure R1).

While *NPW* expression is known to stimulate food intake ⁶ and observed to be broadly expressed by EECs ^{7,8}, here we observe a specific subpopulation of NPW+ ECs marked also by expression of *PRAC1* and *FXYD2* (Figure R6a-b).

We further delineate genes involved in the differentiation of EC cells from *NEUROG3*+ precursors (Figure R6c). These genes included known transcription factors such as *NEUROD1*, *CHGB*, *CHGB*, *PAX4* and *PAX6*, as well as recently described genes such as *FEV* ^{9,10}. Finally, we use proteincellatlas.org images to visualise some of these genes, including *KLK12* and *AFP*, that were found in *NEUROG3*+ progenitors and were located at the bottom of the crypts. Other identified genes marked specific subsets of intestinal cells (Figure R6d).

We have now summarised these results in the revised Figure 2e-g (Supplementary Fig. 4c-e) and in the main text.

Figure R6: a) UMAP visualisation of EECs from fetal, pediatric and adult datasets, where arrows depict differentiation paths

		inferred from scVelo analysis. b) Feature plots showing key genes of NPW+ enterochromaffin (EC) cells. c) Heatmap with genes that change along the differentiation trajectory from NEROG3+ progenitors to EC cells as highlighted by red arrow in a). Arrows indicate known genes associated with EC differentiation in mice and organoids. d) Immunohistochemical staining of selected genes from c) in the small intestinal sections from Human Protein Atlas (proteinatlas.org). Arrows point to positive cells.
1.8	The authors hypothesize that IgG sensing of tuft cells via FCGR2A activates PLCG2 and may affect the inflammatory response of the gut. Is there any evidence for a physiological role of PLCG2 in the context of inflammation. To complement the organoid experiment it would be informative to test whether this gene (and also Fcgr3) is up-regulated upon treatment of mice with inflammatory cytokines or in IBD mouse models (using data from published studies). With regard to the organoid experiments the authors should address whether up-regulation of PLCG2 is due to increased expression in tuft cells or other cell types, and whether FCGR2A also increases in expression upon treatment with inflammatory cytokines in tuft cells specifically.	Thank you for insightful comments regarding the PLCG2 role in inflammation. Activated PLCG2 is involved in inflammatory responses in immune cells triggered by receptors such as FCGR2A. In neutrophils PLCG2 downstream of Fc receptor-mediated and integrin signalling is required for degranulation and adhesion/spreading in inflamed capillary venules ¹¹. Furthermore, gain-of-function mutation in murine PLCG2 has been shown to result in hyperreactive external calcium entry in B cells and expansion of innate inflammatory cells ¹². Other receptor types, via cross-talks, are also implicated, but the links are less-well documented. In addition, we now show a significant increase in inhibitory FcγRIIb-expressing tuft cells in DSS colitis mice (Figure R7).

Figure R7: Percent expression of indicated Fcγ receptors by SiglecF+EpCAM+ small intestinal tuft cells in wild type (n=5) and DSS treated (n=3) mice determined by flow cytometry. Mean with standard deviation is shown in bar plots, and statistics are calculated by multiple t-test analysis **= $p < 0.01$; NS= non-significant.

Based on feedback from each reviewer, we have decided to move mention of a role for tuft cells in IBD via PLCG2 to the discussion section and have included this section here for your convenience:

“Since IgG can uniquely transverse the placenta, this provides a potential route for passive immunization and tuft cell activation in utero. Additionally, IgG in maternal breast milk may activate tuft cells in early life. Two missense variants of PLCG2 have been linked to aberrant B cell responses in early onset IBD (de Lange et al. 2017; Uhlig 2013) and primary immune deficiency (Martín-Nalda et al. 2020). It is possible that tuft cells, via PLCG2 signalling, could also contribute to IBD pathology and immune-related disease, a link that requires further investigation.”

Regarding the organoid experiments in the

original manuscript, PLCG2 expression is increased in all epithelial cells. During this experiment, the organoids were stimulated with cytokines in maintenance condition and we did not observe organoid-derived Tuft cells in this data (no expression of Tuft cell specific gene POU2F3) (Figure R8). We were unable to detect FCGR2A or FCGR2B expression by these organoids derived from control or Crohn's pediatric samples and treated with inflammatory cytokines (Figure R8). These results are confounded again by the absence or low number of tuft cells detected in these samples, but suggest that epithelial PLCG2 can also be activated in the absence of Fcy receptors.

We clarify this in the text and here:

“While there was no morphological difference in stimulated organoid lines and limited expression of FCGR2A (Supplementary Fig. 3F), we confirm an increase in PLCG2 expression across stimulated organoid epithelial cells (Supplementary Fig. 3G, n=3).”

Figure R8: Heatmap of normalised gene expression in single cell RNAseq data of organoids from Crohn's (n=1) and control (n=3) pediatric biopsies stimulated with inflammatory cytokines.

In addition, we have performed *in vitro* experiments in which we differentiate human intestinal organoids and see an increase in

		Tuft cells as marked by POU2F3 expression as well as increased FCGR2A/B and PLCG2 expression in differentiated cultures (Figure R9). These results support the presence of Fcγ receptor-expressing tuft cells in humans. These results are now summarised in the revised manuscript, Supplementary Fig 3j-l and below.  Figure R9: Morphology of undifferentiated (back) and differentiated (blue) intestinal organoids. Expression of LGR5 and MUC2, and other key genes by human intestinal organoids as determined by qPCR.  <caption>Relative expression of key genes in differentiated vs undifferentiated organoids</caption>   Gene Relative expression to undifferentiated organoids     LGR5 ~1.5   MUC2 ~8.0   POU2F3 ~2.5   PLCG2 ~3.0   FCGR2A ~1.5   FCGR2B ~1.5   	Gene	Relative expression to undifferentiated organoids	LGR5	~1.5	MUC2	~8.0	POU2F3	~2.5	PLCG2	~3.0	FCGR2A	~1.5	FCGR2B	~1.5
Gene	Relative expression to undifferentiated organoids															
LGR5	~1.5															
MUC2	~8.0															
POU2F3	~2.5															
PLCG2	~3.0															
FCGR2A	~1.5															
FCGR2B	~1.5															
1.9	Regarding the characterization of gut ILCs, I'm not convinced about the classification into ILC3s and ILCPs. I would expect up-regulation of KIT and ITGB7 in ILCPs versus mature ILC3s. I would also expect reduced levels of RORC in ILCPs versus mature ILC3s. The cells classified as ILC3s also have increased levels of ID2 compared to ILCPs and do not express IL17A which argues against their classification as mature cells. Since the cells classified as ILCPs up-regulate ZBTB16 and TCF7 I do believe that they could correspond to progenitors, however, I doubt the classification as mature ILC3s is correct for the respective cluster. Please also show additional progenitor markers from Lim et al. (2017) Cell, such as Cd7 and Il1r1. Do the ILCPs also express markers of ILC1s and ILC2, i.e., T-BET, CRTh2, and GATA3, at low levels?	In response to the reviewer's comment, we provide human ILCP marker gene expression described by Lim et al. (Cell, 2017) (Figure R10a). However, we note that this study describes ILCPs in peripheral blood (PB) and suggests that PB ILCPs are dramatically different from the equivalent cells in the adult gut. The majority of marker genes described in human PB ILCPs were expressed across gut ILC subsets (Figure R10a). Lim et al. further point to differences in ILCPs of tissue versus PB and classify fetal liver ILCPs as LinCD127+ (IL7R), CD117+ (KIT), RORC+ cells that further express CCR6, neuropilin-1 (NRP1), but not NKp44 (NCR2). Fetal liver ILCPs also did not produce significant amounts of IL-17A or IL-22. Our ILCP cells best fit the above description of fetal liver ILCPs as described in Lim et al., Cell (2017) (Figure R10a). In addition, we directly compare transcriptional signatures of our gut ILC subsets and the human fetal liver ILC														

progenitors described in Popescu et al., Nature 2019 (Figure R10b). Along with other genes characteristic of intestinal ILCPs, we note high expression of *SCN1B* and *HPN* that we also observe in liver ILCPs. These cells might be the first migrants from the fetal liver to gut. To visualise *SCN1B*+ ILCPs in the fetal gut, we use smFISH imaging and show the presence of all three LTi-like populations described in our study (Figure R10c).

Regarding ILC3 maturity, we observed the expression of *IL22* in the adult compared to fetal *NRC2*+ ILC3 (Figure R10a), suggesting that fetal gut ILC3 represents an immature ILC3 counterpart. We use “mature ILC3” nomenclature when describing adult ILC3s in the revised manuscript:

“The second cluster labelled ‘LTi-like NCR+ ILC3’ clustered closely with mature ILC3 cells from adult samples..”.

Figure R10: a) Dotplot with expression of genes described in peripheral ILCPs and fetal liver CD177+ ILCPs by Lim et al. (Cell, 2017). In addition, we show the expression of ILC1 and ILC2 markers and selected cytokines. b) Dotplot with scaled expression of ILC3 and ILCP markers in fetal liver ILCPs from Popescu et al., Nature 2019 and LTi-like subsets described in this study. c) smFISH imaging of ILCP (*SCN1B*+), LTi-like NCR- ILC3 (*IL17A*+) and LTi-like NCR+ ILC3 (*NCR2*+) cells in the human fetal intestine (15PCW).

We would like to further note that after inspection of G D chain expression in the smart-seq2 data, we did not find TCR α or TCR β chains in the cells previously named “LTi-like NK/T” (Figure R11b). Based on RORC and KIT expression and similarity of these cells to the adult Th17 subset, we have decided to revise their annotation to “NCR- ILC3”. The expression of IL17 by this subset is consistent with ILC3 cells described in mouse¹³.

These results are now included in the main Figures 4 and 5 as well as Supplementary Fig. 7.

Figure R11: a) UMAP visualization of T and innate-like cells in fetal, pediatric and adult samples. Dotted line denotes lymphoid tissue inducer (LTi)-like cell types. “ILC3” refers to the adult subset. b) Barplot showing TRACER analysis of TCR chains in cells captured using full-length Smart-seq2 single cell sequencing. c) Heatmap showing mean probability of cell types matching between

		fetal and Crohn's disease stromal and immune cell populations.
1.10	What is the meaning of the arrows in the UMAP of Figure 2? I could not find a description in the legend.	The arrows in Figure 2a depict the inferred paths of differentiation from epithelial progenitor and crypt cells towards secretory and absorptive enterocytes. We have now removed these arrows in the revised figure and replaced these with arrows determined from RNA velocity analysis to show the developmental relationship between epithelial cell types. This figure is included as Figure 2a-b with a clarified legend in the revised manuscript and below:  Figure R12: UMAP of epithelial cell types detected in single cell RNAseq dataset of (a) fetal small and large intestine and (b) pooled pediatric and adult intestinal samples. RNA velocity analysis is overlaid on UMAP.
1.11	The description of the Milo method is too superficial. It is unclear to me how the authors control for overall differences in numbers and relative abundances (which might be of technical nature) between samples in integrated datasets analysed with Milo and how exactly they test for differential abundance statistically.	In short, Milo method performs differential abundance testing by assigning cells to partially overlapping neighbourhoods on a k-nearest neighbour graph. The variability in the number of cells between samples is modelled using a generalized linear model, and the total number of cells in each sample is used as an offset. Further detail about the Milo method can also be found in a recent preprint (Dann et al., 2020: https://doi.org/10.1101/2020.11.23.393769)
Reviewer #2:		

2.1	Elmentaite and colleagues report here a vastly comprehensive catalogue of cells and single cell gene expression patterns, throughout different intestinal tissues and human age. This is an impressive catalogue (a word used by the authors), which the authors use to map cells involved in pathologies, such as IBD and Hirschsprung's disease. They also use this database to identify pathways that regulate Tuft cells and type 2 immunity, and the LTi-LTo crosstalk required for the development of lymphoid tissues during ontogeny and chronic inflammatory pathology. In my particular field of expertise, immunology and the development of lymphoid tissues, the biology developed in the manuscript is precise and correct, and the interpretation of the data largely confirms previous knowledge. I have to confess that it is however extremely difficult, for me, to have a critical appreciation of the data, which consists mainly of scRNAseq data treatment. Therefore, given the correctness of the interpretation, and my inability to judge the quality of the data, I cannot manage to propose sensible revisions. In sum, this manuscript is impressive in its scope and quantity of data, interpretation pertaining to my area of expertise is correct, but net advancement in knowledge is limited. This study will nevertheless provide the field with an invaluable resource.	We thank the reviewer for their time in reviewing our manuscript and for appreciating its scope and accuracy in interpretation. We also believe that this manuscript and dataset will be of great value to the field. In particular, we are excited by the following key features of our manuscript:  ● The most comprehensive single-cell census of healthy human intestines covering in utero development, childhood and adulthood, and encompassing approximately 350,000 cells from 27 donors and up to 10 anatomical regions. ● A systems integrated analysis approach leading to characterisation and first human description of three key cell players driving gut-associated lymphoid tissue formation in development: cells of mesenchymal, endothelial and lymphoid origin, and adaptation of this cellular program for immune cell recruitment in pediatric Crohn's disease ● Novel identification of IgG sensing by tuft cells. ● Novel enteric neuronal progenitors, and cell type-specific expression of Hirschsprung's disease- risk genes in a neuromuscular cell circuitry.
-----	---	---

Referee #3:

3.1	The manuscript by Elmentaite et al provides a new human gut cell atlas encompassing single cell RNA-sequencing analysis of fetal, pediatric and adult cell populations. 350 000 high-	We thank the reviewer for their positive outlook of our manuscript and valuable suggestions. With regards to how our manuscript
-----	--	---

	quality intestinal cells are included representing various gut cell types at distinct anatomical positions. The presentation focuses on 1) the epithelial compartment, highlighting distinct BEST4+ in small intestine versus colon and proposing tuft cells in the pathogenesis of IBD by IgG sensing 2) the gradual differentiation of ENS from ENCCs, highlighting presence of many glial populations and interactions with claimed relevance for Hirschsprung's disease 3) genetic programs controlling lymphoid structure formation, highlighting its reactivation during pediatric Crohn's disease. The study forms a valuable resource for a vast range of intestinal research and gives an impressive description of the gradual development and maturation of the human intestine including associated mesenteric lymph nodes. However, as a whole, the manuscript represents a patchwork of topics, each alone with limited depth sometimes suffering from far-fetched conclusions. The manuscript would be much improved if the data-sets would be presented as a resource and speculative aspects moved to the discussion. It will important to justify how your atlas stands out and compares to the many recent publications describing the developing and adult gut by RNA-seq (see also major issue 4; e.g. Fawkner-Corbett et al., Cell 2021; Han et al., Nature 2020; Holloway et al., Cell Stem Cells, 2020; Cao et al. Science 2020). Alternatively the manuscript could be split - several impactful studies (up to 3) could potentially be made by following up each part with functional experiments.	compares to the studies published to date, we include Table R1, summarising the samples and cell types described previously and in the present study. In summary, we present the most in-depth sampling of the adult human gut, with single-cell transcriptomes from up to 12 distinct regions along the intestinal tract from disease transplant donors, in contrast to other published datasets where biopsies were used. In addition, we include transcriptomes from the developing mesenteric lymph nodes as well as their counterparts from the adult, and in doing so provide an in-depth catalogue of immune cells that survey the adult and developing gut. Through careful scholarship, analysis and interpretation of this data, we have provided unprecedented insights (i.e. Tuft immune sensing, enteroendocrine differentiation, lymphoid tissue formation, and timepoints and locations of ENS differentiation) and contextualised these with other single cell studies (i.e. BEST4+ enterocytes/colonocytes, comparable branching of ENS development in mice and humans).
3.2	Major Issues: 1) Presence of small intestine BEST4+ cells is claimed as a novel finding in the abstract and introduction, although	We clarify this in abstract: "This reveals the presence of BEST4+ absorptive cells throughout the human

	BEST4+ cells have previously been reported across the human intestine including both small intestine and colon with regular immunohistochemistry (Ito et al., 2013 PLoS One). The present manuscript describes however a very interesting regional difference between the BEST4+ cells that are appropriate to high-light. Please rephrase the major finding in the abstract and introduction (perhaps focusing instead of the transcriptional difference between BEST4+ cells in colon and small intestine?).	intestinal tract, demonstrating the existence of transcriptionally distinct BEST4+ cells beyond the colon.” and introduction: “In doing so, we reveal new insights into the epithelial compartment: we identify transcriptional differences between the BEST4+ cells in the small and large intestine <..>”.
3.3	2) PLCG2 expressing Tuft-cells: a) The presumed link between IgG sensing Tuft cells and IBD are built on several assumptions forming circumstantial reasoning. For many IBD risk genes, the cells in which the gene acts has not been established. In the case of Plcg2, an obvious role within the immune system is shown. It is possible that the small proportion of Tuft cells are IgG sensing via Plcg2 and could contribute to disease, but convincing evidence are lacking herein. What is the evidence for that Fcgr2a can upregulate Plcg2? Is Fcgr2a expression increased in CD patients material? The suggested link is to this reviewer too weak to be high-lighted in the results section. It would be more appropriate to more generally suggest signaling pathways involved in the immunological responses of Tuft cells, where Plcg2 could be one plausible gene. The possible role in IBD belongs to the discussion.	Thank you, we agree that the link between IgG sensing by tuft cells and IBD is not substantially supported and we have moved comments about this to the discussion as suggested (please also refer to comment 1.8 for further information). We have also used the finding of FCGR2A expression by tuft cells as an example more generally of the ability of tuft cells to engage in immune signalling. The relevant text from the manuscript is included below in blue. With regards to whether FCGR2A can upregulate PLCG2, there is considerable background data that FCGR2A is a receptor that activates PLCG2 and that one receptor is able to activate many PLC molecules. However, to our knowledge there is no data about the upregulation of PLCG2 in response to FCGR2a signalling. Furthermore, we cannot see expression of FCGR2A in our organoid cultures derived from pediatric Crohn’s and control samples (Figure R8 above and again below). The clear evidence presented in this manuscript that PLCg2 is expressed at higher levels in Turf cells compared to other cell types (the finding not previously appreciated) will have considerable impact on many studies in the field. In particular, IBD- linked variants of PLCG2 will need to be assessed in the context of this cell type.

“We discover that FCGR2A, a receptor activated by the Fc fragment of IgG upstream of PLCG2, and other downstream signaling molecules are expressed by tuft cells. Since IgG can uniquely transverse the placenta, this provides a potential route for passive immunization and tuft cell activation *in utero*. Additionally, IgG in maternal breast milk may activate tuft cells in early life. Two missense variants of PLCG2 have been linked to aberrant B cell responses in early onset IBD^{14,15} and primary immune deficiency¹⁶. It is possible that tuft cells, via PLCG2 signalling, could also contribute to IBD pathology and immune-related disease, a link that requires further investigation. Overall, this data suggests a novel and potentially very impactful immune-sensing role for intestinal tuft cells.”

Figure R8: Heatmap of normalised gene expression in single cell RNAseq data of organoids from Crohn’s (n=1) and control (n=3) pediatric biopsies stimulated with inflammatory cytokines.

3.4 b) The presumed receptor activating Plcg2 is expressed in a very small proportion of Tuft cells (2,5%). Despite this, organoid cells upregulate PLCG2 upon TNFa or IFNg treatment. Do you also observe an upregulation of Fcgr3 (or other ITAM receptor) following the treatment that could help boost the

There was no observable expression of FCGR2A, FCRGR3 and other ITAM receptors in the organoid (Figure R8 above) or in pediatric Crohn’s and control single cell data. Additionally, FCGR3 is not upregulated by intestinal organoids following cytokine treatment (Figure R13). These results may, however, be confounded by the few tuft cells captured in these datasets. Thank you for

presumed signaling pathway? Similarly, is the receptor increased in CD patients?

The lead is interesting, consider to continue this project to substantiate the findings and separate this part into an own manuscript.

suggesting pursuing in a dedicated follow-up manuscript, we will consider this.

Figure R13: Heatmap showing mean expression of selected receptors by intestinal organoids stimulated by TNF α , IFN γ or control culture medium and processed by single-cell RNAseq.

3.5	3) Analysis of the developing ENS has several issues that need to be addressed. a) Overall Analysis: The ENS develops through a protracted time period and neuron subtype specification appears to depend on time of birth (Pham et al J, Comp Neurol. 1991, Bergner et al. J Comp Neurol 2014) as well as post-mitotic neuronal conversions in mouse (Morarach et al, 2021 Nat.Neuroscience). Additionally, it is likely that the enteric progenitor population develops with time in analogy to the gradual differentiation of neural tube progenitors to “radial glia” phenotype in the CNS, upon which Blbp amongst others markers are increase in expression. Moreover, submucosal plexus only starts to be generated by W12-14 and colonic ENS will partly consist of sacral neural crest, likely having a different expression profile compared to the vagal neural crest progenitors. Thus, to be able to investigate discrete differentiation processes, regional and temporally different data-sets will need to be analysed in separation from one another. It appears as if the vast majority of the analysis is performed with cells merged from the whole neurogenic period at all regions (thus a span of 77 days!). While an overall view with all stages should be shown and is informative, a detailed analysis of gliogenesis, neurogenesis and neuronal specification is likely hampered by inclusion of all stages/regions. A separation of for instance W6-10; W10-14 and W14-17 by the region would likely give better basis to clearly follow differentiation and branching events. This suggestion might resolve some of the specific issues identified below:	Many thanks for the expert insights into ENS development. After separating the samples into multiple time point bins as suggested, we observed that cells from 6-11pcw and 12-17pcw displayed different differentiation trajectories, as predicted by the reviewer. We now include these results in the revised Figure 3a, b as well as Figure R14 below for your convenience.
------------	---	---

Figure R14: UMAP projection of enteric neural crest cell progenitors (ENCC) and their progeny at a) 6-11 pcw and b) 12-17pcw timepoints. The overlaid arrows show scVelo derived differentiation trajectory. Selected marker genes are listed for each

population. c) Barplot with relative abundance (percentage %) of cell types amongst ENCC-lineage populations as described in a) and b) across intestinal regions and developmental timepoints.

Overall, we did not observe major regional differences between neural cells- similar trajectories were observed when colonic ENS cells were subclustered (Figure R15).

Figure R15: UMAP visualisation of colonic neural cells colored by cell type annotation, sample, 10x Genomics technology version as well as key marker genes in the developing colon at a) 6-11pcw and b) 12-17pcw.

To delineate the sacral versus vagal origin of neural subsets, we examined HOX gene expression patterns in the neuronal subsets from 6-11pcw samples, as suggested by the

		reviewer (Figure R16). We observe low expression of anterior Hox genes in the Glia 1 (DHH+) subset of cells, along with expression of posterior Hox genes in colonic Glia 1 (DHH+) subset. This suggests that colonic Glia 1 (DHH+) subset may have sacral origin. In addition, colonic Glia 1 (DHH+) subset expressed POU3F3, specific to colonic neurons and uniquely expressed TFAP2B, which was not observed in equivalent cells from the small intestinal regions (Figure R16, genes in red). We now include this analysis in Supplementary Fig 5 and discuss this in text of the revised manuscript: “HOX gene expression further showed that colonic glia 1 cells expressed posterior HOX genes as well as TFAP2B, suggesting their sacral/trunk origin (Supplementary Fig. 5g-h).” Figure R16: HOX gene expression across developing neural subsets in 6-11pcw samples. FPIL= fetal proximal ileum, FTIL= fetal terminal ileum, FLI=fetal large intestine. Highlighted in red square are Glia 1 (DHH+) cells from all regions. Genes highlighted in red are colon specific.
3.6	b) Committed glial precursors: It is not trivial to discern glial committed precursors from ENCCs or differentiated enteric glia. Higher levels of markers associated with adult glia would not	To clarify, we score the progenitor/glial populations using literature curated gene list from Lasrado et al., Science (2017) (namely ERBB3 , PLP1 , COL18A1 , SOX10 , GAS7 , FABP7 , NID1 , QKI , SPARC , MEST ,

necessary indicate glial-committed precursors (could also indicate a more mature progenitor population that is still bi-potent). It is not clear which genes are used to define committed enteric glia. If it is those specified in methods part (Sox10, Mpz, S100b, Erbb3, Plp1, Gas7, Col18a1), please refer to appropriate confirming paper or propose a reasoning to picking these genes. The mentioned markers for this population (AGRM and SGO2) are also un-suggestive to mark glial committed precursors. AGRM is a known marker of radial glia that are multipotent (Marinaro et al., 2011 PLoS One) and SGO2 has been shown to regulate mitosis (Gomez et al., EMBO Rep 2007). The expression of CCNA2 and CCNB1 also points to that this population corresponds to a specific stage during the cell cycle. A committed glial precursor population would show all phases of the cell cycle – is that the case? Moreover, your data indicate a higher proportion of committed glia precursors at early stages (Figure 4B), while it is known that neuron differentiation proceeds glial differentiation in the developing ENS (although it is much more simultaneous than in the developing CNS). As suggested above, separation of data by stage (and perhaps region) would make the interpretation easier.

WWTR1, GPM6B, RASA3, FLRT1, ITPRIPL1, ITGA4, POSTN, PDPN, NRCAM, TSPAN18, RGCC, LAMA4, PTPRZ1, HMGA2, TGFB2, ITGA6, SOX5, MTAP, HEYL, GPR17, TTYH1). We have clarified this in the Methods section.

Regarding committed glial progenitors, we observed that this population is composed of G2M phase only and refer to it as “cycling glia/progenitor cells” in the revised manuscript and Figure R14 (above).

Separation of the timepoints also allowed us to further clarify the timing of glia versus neuron differentiation in the human gut. In 6-11pcw, the most abundant population was ENS progenitors and only a small cluster of “Glia 1 (DHH+)” cells were observed at this time point (Figure R14a, c). The trajectory analysis suggests that at this time point, most of the ENS progenitors differentiate into enteric neurons via neuroblasts (Figure R14a).

In 12-17pcw, we observed differentiation of the ENS progenitors to both glial and neuronal types (Figure R14b). At this timepoint, ENS progenitors appeared to give rise to three types of glial populations (Figure R14b). Firstly, via a branching event progenitors differentiate to DHH+ Glia 1 and ELN+ Glia 2 populations via differentiating glia. Secondly, progenitors differentiate to BCAN+ Glia 3 population (previously named Enteric Glia).

We comment on this differentiation in the text of the revised manuscript:

“While differentiated neuronal cells were abundant at 6-11PCW, glial cell types were enriched in the second time point (12-17PCW) suggesting that neuronal differentiation precedes gliogenesis. All terminal glial cells expressed high levels of ENCC progenitor/terminal glial marker

		genes including FOXD3, MPZ, CDH19, PLP1, SOX10, S100B and ERBB3, but lacked RET (Supplementary Fig. 5f). We further observe three types of enteric glia: glia 1 (DHH, RXRG, NTRK2, SBSPON, MBP), glia 2 (ELN, TFAP2A, SOX8, BMP8B), glia 3 (BCAN, APOE, CALCA, HES5, FRZB) and a subset of differentiating glia (COL20A1) present in 12-17PCW. HOX gene expression further showed that colonic glia 1 cells expressed posterior HOX genes as well as TFAP2B, suggesting their sacral/trunk origin (Supplementary Fig. 5g-h). We visualised BMP8B⁺ cells in the myenteric plexus, while DHH⁺ cells were found both in the mesentery and myenteric plexus (Figure 3c, Supplementary Fig. 5i). “
3.7	c) Committed neuronal precursor: From the Velocity analysis these cells are probably committed to becoming neurons, going through their last cell-cycle, ie neurogenic. Would it be better to call them neuroblasts or neurogenic cells? These cells are expected to transiently express proneural genes such as higher ASCL1 while downregulating progenitor genes (Sox10, Sox2) and upregulating early neuronal genes (Tubb2, Pgp9.5, Elavl4). The neurogenic population has been defined in murine ENS (Morarach et al., Nat. Neurosci 2021) and it could be useful to utilize this information to annotate the human neurogenic phase (although of course some gene are likely to differ, the majority of genes are probably conserved). For instance, Mfng, DII1, DII3 are also markers of murine neuroblasts.	We agree with the reviewer's suggestion, and have changed our classification of “committed neuronal precursors” to “Neuroblast”. In Figure R17 we shown that the neuroblast bridge upregulates expression of ASCL1, ELAVL4, TUBB2B (but not UCHL1 (Pgp9.5)), and downregulates SOX10 and SOX2, fitting with their identity as neuroblasts.

		 Figure R17: Feature plots with genes upregulated and downregulated in neuroblasts at 6-11 pcw.
3.8	d) Annotation of Enteric Neuron Subtypes: The branches and logic of enteric neuron diversification in the mouse have recently been defined (Morarach et al Nat. Neurosci 2021). At neurogenesis, a binary event separates Etv1+ precursors (Branch A) from Bnc2+ precursors (Branch B), thus the same transcription factors observed in your human dataset. Branch A differentiate into Enteric Neuron Classes (ENC) 8-12 – corresponding to inhibitory motoneurons (Nos1/Gal/Vip), Neurod6+ putative interneurons as well as Ntn1/Nxph2 neurons that possibly could be a mix between atypical IPANs and interneurons. Branch B differentiate into	We have updated our neuron annotations based on reviewer’s comments and mouse data described in (Morarach et al Nat. Neurosci 2021), paying particular attention to division of branches (Figure R14a,b). In particular, we have labelled Branch A and B based on the expression of ETV1 and BNC2/DLX5 respectively (Figure R18a). In the UMAP of samples from 6-11pcw, Branch A co-expressed NOS1/GAL/VIP (inhibitory motor neurons; iMN) and further differentiated into NTNG1/NXP2+ cells (IPAN). We did not observe the NEUROD6 expression at this stage (Figure R18a). We also show a population of cells co-expressing PENK/TAC1/GRP at the end of

excitatory motorneurons (Penk+/Tac1+), IPANs (Nmu+, Dlx3+, Ano2+, Nog+, NTrk3+...), Interneuron/Intestinofugal Unc3/Cck cells and Sst+ putative interneurons. UMAP structure and Velocity analysis suggested that most neuron classes diverge at postmitotic stages by gradual conversions within the branches. Basic developmental processes are often conserved between species and authors could be helped to analyse their branching events and neuron types by consulting this framework in mouse.

Branch A, suggestive of differentiation of excitatory motor (eMN) neurons from this lineage (Figure R18a). This observation is in contrast to differentiation of excitatory motor neurons only from Branch B as observed in mice. This population did not express *FUT9/GDA* as described in mature murine eMN (ENC1-4) by Morarach et al.,2021, suggesting these may be immature eMNs. In addition, we see a population of eMNs stemming from Branch B at 12-17pcw (Branch B (eMN)). The eMN populations differentiated from Branch A and B showed similar expression of marker genes (Figure R18a,b). This could suggest the possibility of human eMNs differentiating from both Branches A and B at different developmental timepoints.

Figure R18: Dotplots with expression of key genes used to define enteric neuron classes found at a) 6-11 pcw and b) 12-17pcw. Genes highlighted in red are discussed in text. Genes highlighted with black outline are expressed in eMN cells.

Neuron differentiation observed at 12-17pcw was otherwise mostly consistent with observations in mice by Morarach et al., (Nat. Neurosci 2021). At this time point, we note further branching of Branch A to iMN (*NOS1/GAL/VIP*, Branch A (iMN)) as well as *NEUROD6+* cells that are interneurons that reassemble murine ENC10. We did not observe *UNC3/CCK* expression suggestive of mature Interneuron/Intestinofugal cells.

In addition, cells from 12-17pcw showed two

		neuronal subclusters. Both subpopulations expressed BNC2, but not ETV1, suggesting they differentiated from Branch B. The first population as mentioned (Branch B (eMN)) above expressed genes observed in eMNs also seen at 6-11pcw including TAC1/PENK. The second population showed the expression consistent with IPANs including DLX3, ANO2, NOG and NTRK3 expression, and therefore labelled as “Branch B (IPAN)”.
3.9	The current attempts to annotate neuronal subtype is partly questionable, relying solely on the recent un-validated annotation of enteric snRNA-seq clusters (Drokhlyansky, Cell 2020) ignoring a large body of literature already present used to identify enteric neuron subtypes. The Drokhlyansky atlas refer to all CGRP+ cells as sensory, although several other neuron populations express this marker, including the cholinergic secreto/vasodilatory neurons that were expected within that atlas but not annotated. The Drokhlyansky PSN1 is likely corresponding to IPANs, while PSN2-4 would need to be confirmed (these correspond to the murine ENC7, parts of ENC12 and likely cholinergic secretomotor/vasodilatory neurons). Consider to also consult May-Zhang et al., Gastroenterology for human IPAN characterization (both with RNA-seq and tissue staining). Please note that CGRP appears mainly transcribed from CALCB rather than CALCA in the ENS.	To address this we have used the annotation consistent with Morarach et al., (Nat. Neurosci 2021) paper as described above. Based on the genes described in mature ENC1-12 classes, we could distinguish that Branch A iMN, resembled ENC8-9 (NOS1, GAL); Branch A NEUROD6+ INs resembled ENC10. Branch A IPAN/IN; resembled mature ENC12 (NTNG1, NXPH2) (Figure R18 a,b). Not all genes described in mature enteric neuron classes ENC1-12 were expressed in our populations (Figure R19). This could reflect species differences as well as differences in the sampled timepoints in our and mouse dataset (absence of differentiated neurons in our dataset).

Figure R19: Overlaid expression of genes described in ENC1-12 in Morarach et al., (Nat. Neurosci 2021).

In our revised annotation, Branch B (IPAN) also showed expression of CALB2 and SST (Figure R20), described in human IPAN A population (May-Zhang et al., Gastroenterology). This further supported our annotation of this population as IPANs rather than PSN subsets.

Figure R20: Dotplot with expression of IPAN A genes from (May-Zhang et al., Gastroenterology) in 12-17pcw neural populations.

		Finally, we visualise SCGN+ and BNC2+ enteric neurons in the adult tissue, supporting that these cells exist in post-natal gut tissue (Figure R21, bottom panels).  Figure R21: smFISH imaging of BMP8B+ Glia 2 subtype and SCGN+/BNC2+ enteric neurons in the fetal (small intestine, 15pcw) and adult (55-60 years, terminal ileum). We have now added these results in the Supplementary Fig 5.
3.10	For your annotation of possible enteric submucosal neurons, please note that this layer starts to arise at around W12-14 (Fu et al., 2004. Anat. Embryol 208:33-41.) -thus your stages W14-W17 should primarily be used to annotate these neuron types. The number of neurons at these older stages appear however limited, which might make such identification difficult. The markers you indicate (Vip, Etv1, Scgn and Unc5b) are also expressed in myenteric subpopulations, why caution should be taken to annotating submucosa based on these genes. Note that murine IPANs (ENC6) express Unc5b, while Scgn is expressed in murine ENC7 (mature markers CCK/Ucn3) and ENC10	We have taken on board the reviewer's suggestion to be cautious when discriminating between myenteric and submucosal enteric neurons in our data. To resolve the spatial distribution of neuronal and glial subsets in the developing human gut, we have generated and analysed 10x Visium spatial transcriptomics data on developing small intestinal regions from 13 and 17pcw gut. We use Cell2Location algorithm¹⁷ to map the single-cell transcriptomes to the tissue zones (Methods). In this data, we observe the expected mapping of hallmark cell types, including enterocytes, intestinal stem cells as well as smooth muscle cells (Figure R 22a).

(Neurod6+). Thus, with further investigation of murine datasets you may be able to link Branch 1 to ENC6/7 and Branch 2b to ENC10. An alternative to annotate cells by function is to refer to the Class number (Enteric Neuron Classes 1-12 in the mouse myenteric plexus). Future functional experiments will be needed to faithfully annotate the neuron clusters into functional classes. If you find enough similarity

between murine and human neuronal differentiation consider to use the same

terminology as used in Morarach et al (Branch A, B; myenteric ENC1-12) for future clarity in the field.

Regarding neuronal populations, we observed mapping of both neuronal branches to myenteric plexus (Figure R22b). However, the current 10x Genomics Visium chips at 50 micron resolution may not be able to discriminate between rare cell subsets. Further studies will be required to confidently map these populations in myenteric versus submucosa.

Figure R22: a) Spatial mapping of scRNAseq to Visium spatial transcriptomics sections of fetal tissue from 17pcw terminal ileum (top), 17pcw proximal ileum (middle) and 13pcw ileum (bottom). Cell abundance (colour intensity) of cell subsets (colour) is shown; b) Spatial cell abundance (colour intensity) of neuronal, smooth muscle and glial cell subsets (rows) across tissue sections (columns).

Regarding the reviewer's suggestion to use consistent nomenclature, we use Branch

		A/B nomenclature in our data and suggest neuron functional class by considering the gene expression suggested by the reviewer in comment 3.8. Since we do not observe convincing similarity between developing human enteric neurons and ENC classes described in adult mice, we refrain from using ENC classes in our annotations, but rather point to similar populations in text. Further direct comparisons between human and mouse datasets will be required to address similarities between mouse and human cells.
3.11	e) Annotation of Glial classes: It is a hard to follow the annotation and reasoning for enteric glia and schwann cell-like clusters. All three schwann-cell like types are first defined as Schwann Cell Precursors (Row 268), while this term later in the text only denotes a subtype. It is also unclear why the markers PLP1, PMP22, CDH19 and ITGA4, CDH2 are used to define SCPs. No reference markers for human schwann cell precursors/cells that distinguish these from developing enteric glia has been published (if there is, please cite). All these markers are however robustly expressed within murine enteric glia (see: mousebrain.org – enteric glia, Zeisel et al., Cell 2018). Dhh is the only marker known to this reviewer that would faithfully distinguish schwann cell precursors from ENCCs (again this is from mouse studies including Uesaka et al., 2015). It would be advisable to re-think the nomenclature for the glial subtypes considering the following: Myelin SCP: Are you sure that myelinating Schwann cells are present in the mesentery and/or gut wall? Expression of Mpz is not a valid reason to call this a myelinating type as this is an early marker of schwann cells in general	Thank you, we have now achieved a clearer differentiation trajectory by subsetting of the ENS data as described in point 3.6 above. “Myelin Schwann-like” cluster was enriched for MPZ, MAL, GFRA3 and DHH expression suggesting similarity to “putative SCP” found in mice (Morarach et al, 2021). To avoid confusion in the field we name this cell “Glia 1 (DHH+)”. We also note that some DHH expression was observed within the gut wall in MPZ+ SOX10+ cells, but the expression of DHH was not as abundant as in the cells outside of the gut wall (Figure R23). In addition, DHH+ Glia 1 cells confidently mapped to tissue zones outside the gut wall in the spatial transcriptomics data (Figure R22b). Cells originally named as “SCP” clustered closely with DHH+ glial cells in 6-11pcw samples and we refer to this population as “Glia 1 (DHH+)”.  Figure R23: smFISH imaging of DHH+ glial population in the human fetal intestine (15pcw).

(Jessen and Mirsky, 2019, *Frontiers in Mol Neuroscience*). This glial type could correspond to the (unvalidated) putative SCP found in mouse ENS (enriched Mpz, Mal, Wnt6, Gfra3 and Dhh; Morarach et al, 2021). Your validation of Dhh+ cells in mesentery (fig 3) does not rule out the presence of these cells within the gut wall, please re-analyse this possibility (given that Dhh+ cell in mouse infiltrate the gut wall and give rise to neurons at postnatal stages, Uesaka et al, 2015)

Non-myelinating Schwann-like: What is the reasoning to denote this type schwann-like? The murine enteric glia consist of many subclasses (see Zeisel et al., 2018; *mousebrain.org* – enteric glia). These express variable levels of Eln, Sox8, Apoe, Bcan. Consider if your “non-myelin schwann-like cell” instead could correspond to a subdivision within the enteric glia.

SCP: Please validate the presence of this cell type in tissue (as was performed with non-myelin Schwann cells with BMP8B). It is misleading to state enrichment in first trimester. These cells peak at W10 and are present with similar proportions at W9 as w17. Stating the age of first detection would be more informative.

In conclusion, considering no validation and functional analysis of glial populations, could they be annotated as enteric glia 1-4 in this atlas?

Does the velocity and UMAP look more crisp if only relevant stages (W9-17) are included in the analysis of glial clusters and differentiation?

Are the different glial populations regionally distributed (colon versus small intestine)?

In the umap from combined timepoints, DHH+ glial cells from 6-11pcw and 12-17pcw samples clustered together (Figure R24a) and we further provide genes defining all three glia subsets (Figure R24b). As suggested by the reviewer, we annotate these subsets as Glia 1-3 and provide their defining genes in the brackets (Glia 1 (DHH+), Glia 2 (ELN+), Glia 3 (BCAN+) and a subset of Differentiating glia (Figure R14b).

Regarding regional distribution of glial populations, we observe all populations in both small and large fetal intestines (Figure R15 above and Figure R24c).

Figure R24: a) UMAP visualisation of neural subsets from 6-17pcw. b) Dotplot with genes defining each glial population. c) Barplot showing regional distribution of neural subsets in 6-17pcw timepoint. FPIL- fetal proximal ileum, FMIL- fetal middle ileum, FTIL- fetal terminal ileum, FLI- fetal large intestine.

These results are now summarised in Figure 3, Supplementary Fig. 5 as well as in section of the revised manuscript titled “Enteric nervous system differentiation during development”.

3.12	f) Acknowledgement of present literature of the developing human ENS: Several of the identified signaling molecules have previously been investigated in the developing human ENS in an immunohistochemical atlas comparing murine and human ENS transcriptional and signaling regulators (Memic et al., Gastroenterology 2018). It would be appropriate to use this resource as validation for expression scRNA-seq. For instance, the mentioned GDF10 as well as other TGFB-receptors Tgfr1, Tgfr2 and Tgfb2 protein expression have been detected at W6-10. The Branch 1 specific transcription factor Etv1 is also validated in human ENS within the same atlas. A very recent paper (Fawkner-Corbett et al., Cell 2021) also characterizes the developing human ENS (week 12-19), please refer to this paper and compare with your own analysis.	We acknowledge work by Memic et al. in text as follows: “.. observed in developing human gut previously (Memic et al., Gastroenterology 2018).“ We summarise general similarities and differences between the recently published Fawkner-Corbett et al., (Cell, 2021) study in Table R1 at the end of this document. In addition, we perform logistic regression analysis to directly compare the developing ENS cells between two studies. Overall, we observe a good correspondence between cells in the present study and those described in Fawkner-Corbett et al., (Cell, 2021) (Figure R25). In the present study we annotate cells based on mouse dataset (Memic et al., Gastroenterology 2018) and describe the branching events occurring at different developmental stages. In short, Glia 1 (DHH+) cells matched with “Lymphoid associated glia”, Glia 2 (ELN+) matched with intraganglionic glia and Glia 3 (BCAN+) with Submucosal glia (Figure R25). The cell names in (Fawkner-Corbett et al., Cell 2021) were assigned based on gene expression and validation for the location of these subtypes was not provided, hence we decided to keep the Glia1-3 nomenclature. Branch B cells matched with Neuroendocrine (1), while Branch A IPAN/IN with Interneurons and Neuroendocrine (2) subsets described in this study (Figure R25). Branch A (eMN) matched with interneurons that were annotated based on expression of TAC1 and PENK. As the reviewer suggested, these genes were also indicative of ENC1-4 and excitatory motor neurons. Based on this and genes shared between Branch A and B eMN cells as in Figure R18, we decided to keep their annotation and
-------------	---	--

		clarify this in text as follows: “...and a population resembling excitatory motor neurons (eMN) marked by expression of PENK, TAC1 and SLC18A3 (Supplementary Fig. 5c-d). Similar population was recently described in the human fetal gut¹⁸ (Supplementary Fig. 5e). ”  Figure R25: Heatmap showing percentage of cells matching between ENS cells described in this study and cells from Fawcner-Corbett et al., Cell 2021 dataset (Predicted). Timepoints 6-17pcw These results are included in the Supplementary Fig 5e of the revised manuscript.
3.13	g) Hirschsprung disease risk genes in dataset: Risk genes for HSCR are extensively investigated and have resulted in the identification of many genes that can explain the aganglionic phenotype (replicated in murine models). RET, GDNF, NRTN, SOX10, EDNRB, EDN3, ECE1, ZFHX1B, PHOX2B, KIAA1279, TCF4 are major HSCR risk genes. Studies have also linked SEMA3A, MAPK10, NRG1, SOX2, BACE2 and GFRA1 (Tang et al., Gastroenterology 2018). Note that genes involved in ENS development have been investigated as candidates in many	We agree with the reviewer's comment and as recommended, we limit the analysis to the major risk genes suggested by the reviewer (RET, GDNF, NRTN, SOX10, EDNRB, EDN3, ECE1, ZEB2 (ZFHX1B), PHOX2B, KIF1BP (KIAA1279), TCF4) as well as those published in Tang et al., Gastroenterology 2018. As suggested by the reviewer, we have amended the text and updated the Figure 3 to show the expression of HSCR-disease genes across neural subsets (Figure R26a). Reassuringly, we did not observe the

studies, where some have been linked (HOXB5) - while others have been disproven to contribute to HSCR, including ASCL1 (Carter et al., 2012 J Hum Genet.) Please limit your annotation to confirmed HSCR-disease linked genes

and do not include own candidate genes (remove ASCL1, TLX2 and possibly others). Carefully review your HSCR risk genes. The illustration in major Figure 4G is questionable as it fails to illustrate the fine validation provided by this paper of already

suspected transcription factors, cell-interactions and receptor/ligand interactions and instead high-lights new interactions that are not substantiated by experiments nor warranted from unexplained phenotypes in HSCR. Instead, for the HSCR field to gain the most from your resource , please make a Table similar to Fig 4f or Sup 4l with all

significant HSCR risk genes. This could identify new leads to be followed up by experiments. It is for example very interesting that Ret is higher expressed in the colon than in the small intestine. Note that ERBB2 and NRG1 variants are only found in very few HSCR patients and that there are other sources of NRG1, including sub-epithelial mesenchyme (Holloway et al., Cell Stem Cell In Press) and smooth muscle cells and fibroblasts (Elmentaite et al., Developmental Cell 2020). It is also fine to include genes previously implicated in ENS development either in the same or separate table, but they could be denoted “critical for ENS development” or something similar.

It is however true that patients do experience life-long complications including enterocolitis (Gosain et al 2016 Pediatr Surg Int) and an increased risk to develop Crohn’s disease (Granström et al., 2017 JPGN). Hence, there are

expression of GDNF, NRTN or EDN3 in the neuronal lineages. Instead these genes were expressed by stromal and smooth muscle cells (Supplementary 5j and Figure R26b).

Figure R26: a) Heatmap with mean expression of Hirschsprung's- disease associated genes across developing neural subsets separated by region of origin (small-pink vs large- blue intestine) and developmental time point (6-11pcw- purple and 12-17pcw- yellow). b) Dotplot with key Hirschsprung's disease-associated ligand-receptor genes in the whole fetal dataset. Dashed arrows indicate ligand-receptor relationship.

We discuss these findings in the revised manuscript as follows:

“To identify neural cells involved in Hirschsprung’s disease, we screened for expression of known Hirschsprung’s disease-associated genes^{19,20}. To control for variability in the bowel region in which patients experience Hirschsprung's disease, we assessed expression in small and large intestines separately.

	aspects about HSCR that we still do not understand that possibly extends beyond ENS differentiation and colonization. However, the “new” cell-cell interactions indicated by receptor-ligand matching do not seem to reach these gaps in knowledge, are highly speculative, not backed up with functional evidence or HSCR tissue stainings. Your conclusions that the analysis implicate fetal branch 2b and ICC/SCM interactions in pathogenesis of Hirschsprung disease (results) and that your high resolution genomic definition uncovers new insights into both rare and common diseases (in the introduction), represents two statements that are exaggerated. Please refrain from too much speculation. It is enough to show the gene expression of HSCR genes in your different cell populations and conclude that they are expressed more widespread than previously appreciated (or in novel cell populations). Please amend figures, introduction and results accordingly.	The majority of Hirschsprung’s disease-associated genes were expressed across multiple differentiating populations with varying intensity (Figure 3e). For example, TCF4 and ZEB2 was expressed across neural subsets, while MAPK10, PHOX2B and RET were highly expressed by neuronal branches and SOX10, SOX2 and BACE2 enriched in the glial compartment (Figure 3d). Expression levels of disease genes also varied between neuron Branches A and B. For example, RET was highly expressed in Branch A, but not in Branch B neurons. Interestingly, RET was more highly expressed by colonic neuroblasts and Branch A (iMN) neurons, while ZEB2 and EDNRB were more highly expressed across colonic glia and neuroblast subsets (Figure 3e). In addition, key ligands implicated in Hirschsprung’s disease, including GDNF, NRTN and EDN3 were primarily expressed by mesothelium, smooth muscle cells (SMC) and interstitial cells of Cajal (ICC) (Supplementary Fig. 5j), as described previously. Together, our analysis shows broad expression of Hirschsprung’s disease-associated genes across neural and ICC/SMC cells, implicating these neural subsets in this disease pathogenesis. “
3.14	h) Unnecessary speculation: Page 7 row 326: Authors identify NTrk3 expression in an glial population (denoted pro-myelinating schwann cells) and try to link this expression to the previous observation of reduced IPANs in NTrk3 or Nt3 mutant mice. However, Ntrk3 is	We have now removed this section from the revised manuscript. Thank you.

	expressed in IPANs (Morarch et al., 2021), so a much more straight-forward explanation would be the intrinsic expression of this receptor. Please remove this speculation.	
3.15	4) The authors admit that extensive single cell sequencing of human gut tissue at different stages has collectively been carried out in recent time and indicate that they intend to provide a holistic analysis. As such it would be very helpful for the field if the authors could put their new atlas into perspective of recently published studies. For instance, how does the human gut RNA-seq dataset and analysis of Han et al., 2020 Nature; May-Zhang et al., Gastroenterology 2020; Drokhyansky et al., 2020; Holloway et al., Cell Stem Cell; 2020; Cao et al., Science 2020 and Fawkner-Corbett et al., Cell 2021 compare to the current manuscript?	In response to the reviewer's comment, we provide Table R1 summarising the key features of the currently available datasets in comparison to our study. The studies published so far focused either exclusively on human developmental stages (Holloway et al., Cell Stem Cell; 2020; Cao et al., Science 2020 and Fawkner-Corbett et al., Cell 2021) or disease comparisons from a single adult anatomical location (Smillie et al., Cell 2019; Martin et al. Cell 2019). In contrast, here we focus on comparisons between fetal and healthy adult intestinal cells from multiple locations. A unique feature of our study is sampling of fetal as well as adult mesenteric lymph nodes draining the gut. Finally, we would like to emphasise that we hypersample the adult intestinal tract in detail across 11 intestinal locations and mesenteric lymph nodes, which has never been done before. We summarise the cell types described in studies published so far (Table R1) and would like to emphasize that the present study includes a more detailed annotation of fetal immune cells with many cell states described for the first time in the gut (e.g. presence of CLP, Pro-B cells, ILCPs, Megakaryocytes). In the epithelial compartment, we also capture Microfold, enteroendocrine subsets and Tuft cells for the first time in human development. We also describe transcriptomes of BEST4+ cells in the small

		intestine and BEST2+ Goblet cells in the colon- cell types that have not been captured thus far. In the stromal compartment, we uniquely identify cells that likely organise lymphoid organs, including FDCs and T reticular cells and define subsets of lymphatic endothelial cells.
3.16	Minor issues: 1) At many location, you refer to the ENS as “Neuronal” (Figure 1, Supplementary Figure 2, Results, Introduction). It is more appropriate to use the term “Neural” as this encompasses both neuronal and glial types/fates. Please change Neuronal to Neural when you refer to the whole ENS.	Thank you for highlighting this nomenclature. We have now changed “Neuronal” to “Neural” where appropriate.
3.17	2) Figure 1A. Labelling of sampled areas on fetal tissue only states ileum. The proximal and mid ileum appears to be placed in the fetal duodenum and jejunum. Please correct sampling area or labelling of area.	We clarify that “proximal small intestine” encompasses fetal region that will develop into adult duodenum and jejunum. We change the labelling of the area in Figure 1A.
3.18	3) P52; row 168. 3 is missing in NEUROG+	Thank you, we have corrected this error.
3.19	4) Supplementary Figure 1b. The label indicates presence of Glia in red, however no such cells are discernable in the figure	There is a small population of glial cells at the very top and centre of the UMAP of Figure 1b. We have added a circle around this population for clarity. Thank you.
3.20	5) Page 6; Row 265: It is questionable to refer to terminal glia markers Plp1, Sox10, S100b and so as “neural crest precursor genes”. Perhaps it is enough to define these markers as terminal glial markers without referring to neural crest.	We have redefined these genes as terminal glial markers as recommended. Thank you.
3.21	6) Page 6 Row 242: What do you mean by “a shift” in ENS development? Why did such shift enable ENCC investigation?	By ‘shift’ in ENS development we are referring to the dramatic change in the neural cell types present between 10pwc and 12 pwc. By capturing ENCCs and this diversification of cell types, we had the opportunity to study the differentiation of

		neuronal cells from the ENCC progenitor. We have now removed this statement from the revised manuscript to avoid confusion.
3.22	7) Bnc2 is misspelled in results and figure legend 4.	We thank the reviewer for highlighting this error that we have now corrected in the revised manuscript.
3.23	8) Abstract: It is enough to say that you define glial and neuronal cell populations in the developing enteric nervous system (no need to say novel – most are already defined in the mouse). If you want to use novel, only the glial subsets appear truly novel and previously not described.	We have restricted the use of “novel” when referring to neural subsets in the revised manuscript.
3.24	9) Page 3; Row135: Reference should be Figure 1E,F.	We have fixed this in text. Thank you.
3.25	10) Page 5; Row 190: Rephrase “IBD pathogenesis is due to”, you probably mean “IBD pathology is due to”, or “IBD pathogenesis involves impaired intestinal...”	We have fixed this in text. Thank you.
3.26	11) There are duplicated papers in the reference list (for instance Drokhlyansky)	We have fixed this in text. Thank you.
3.27	12) It would be helpful with a table indicating the number of cells in each sample type and the 103 cell types/states (also indicating no of cells from each stage/region). Good Luck!	We provide Supplementary Table 2 with the number of cell types/states by biological (region, age group, sample name) as well as technical (dissociation fraction) factors.
Referee #4:		
4.1	This very broad-ranging, highly ambitious study examines fetal, pediatric and adult gut and lymph node tissues at single cell resolution, including 11 distinct	We sincerely thank the reviewer for their time in reviewing our manuscript and for appreciating its scope and value of its insights into gut development. It is our hope

	intestinal/secondary lymph node locations. Strengths of the study include its "time and space" analyses, providing invaluable single cell insight into gut development. The sharing of data and code through the Human Cell Atlas and will provide an enormously useful resource for the community. The time-series analyses provide an important re-definition of gut development at the single cell level. This is effectively leveraged for advancing Hirschsprung's disease (aganglionic megacolon) pathogenesis, a developmental disorder, but somewhat less so for inflammatory bowel disease (IBD).	that this manuscript and dataset will be of substantial value to the scientific community. We have taken onboard the reviewer's helpful suggestions and made changes as detailed below.
4.2	Major 1. Consistency and developmental differences in sampling capture. Supplementary Table 1 should include cellular yields for each of the samples (EPCAM+/CD45-, CD45+EPCAM-) to provide insight into the consistency, replicability and robustness of findings. This is particularly important as many of their figures with respect to time-series changes may reflect only one sample (would be important to clarify). The description of the pediatric ileal tissues is not described beyond enteroid recovery; however, additional pediatric tissue results are reported beyond epithelial cells. The description of the fetal and adult tissue cell isolations implies that full-thickness digestion was performed; given their sorting strategy, how do the authors interpret the marked relative mesenchymal enrichment observed in fetal tissues, relative to post-natal tissues (Figure 1C)? Full-thickness digestion and differences between fetal and post-natal processing and cell capture efficiencies will have marked	Cell isolation from fetal samples was done on the full thickness of the gut, while adult samples were large sections of the mucosa and pediatric samples were collected as biopsies. All samples were enriched using magnetic-activated cell sorting (MACS) protocol, which leads to less pure enrichment than by FACS. Therefore, although pediatric samples were enriched for EPCAM+ cells, we still capture a large proportion of immune cells in these samples. To rigorously estimate the impact of biological factors on cell type composition while accounting for the effect of sorting strategy as well as other technical factors, we modeled the number of cells for each sample (n=160 samples in total) and cell lineage (9 different cell lineages in total) combination using a generalised linear mixed model with a Poisson outcome. The 5 clinical factors (Age group, Donor, Biopsy or not, Disease status and Gender) and 4 technical factors (Enrichment fraction, 10x kit version, Liberase enzyme and Sample) were fitted as random effects to overcome the collinearity among the factors. The effect of each clinical/technical factor on cell

differences in the interpretation of many items in Figure 1.

lineage composition was estimated by the interaction term with the cell lineage. The 'glmer' function in the lme4 package implemented on R was used to fit the model. The standard error of variance parameter for each factor was estimated using the numDeriv package. **The results of this analysis were provided in Supplementary Fig. 2c-d as well as Figure R27 below. The detailed statistical method is demonstrated in Supplementary Note Section 2.**

For example, the analysis shows that the standard scRNA-seq does not increase nor decrease the epithelial cells, while EPCAM+ MACS enrichment increases epithelial cells by ~5 times (Figure R27a). The mesenchymal cells were significantly enriched in the 1st and 2nd trimester of development compared to postnatal samples. Age group and enrichment fraction accounted for the most variation in the data (Figure R27b).

Figure R27: a) Dotplot where the fold change represents the enrichment (or depletion, low fold change) of cells compared with baseline. The local true sign rate (LTSR) value represents statistical significance of the fold change estimate ranging from 0 to 1, where 1 represents a confident estimate. b) The forest plot showing the relative importance (SD) of each technical/biological factor on the cell

		type proportion.
4.3	2. Figure 2A-B is a highly interesting time-series analyses of relative cell fractions, which effectively leverages the strength of this study, namely the time-series fetal datasets. As classified presently, there are marked small vs. large intestinal differences prior to the "developmental shift" defined in Fig 2B as between 10-12 weeks. For example, for the large intestine, the "crypt" clusters" arise quite abruptly after the "shift" which doesn't necessarily make sense, as large intestinal crypts include both enterocytes and goblet cells. Also, LGR5 and other stem cell markers, should anchor the crypt development descriptions. Ideally, the empiric serial fetal fractions (Fig 2B) could empirically justify the time-series predictions more explicitly (Fig 2A), especially with respect to the secretory vs. transit-amplifying/enterocyte dichotomy. Missing in the large intestinal analysis is a description of the Paneth cell equivalents, described by some groups as REG4+ cells. Finally it is well known that even within small or large intestine, marked cellular composition differences exist between regions (proximal to distal), so in text and/or more concise summaries in supplementary materials (e.g. UMAPs by regions with small or large intestine) should be provided.	We have now substantially revised Figures 2a-b taking into account the reviewers' comments (please also see point 1.10). We have maintained the developmental shift to draw attention to the abrupt shift in mature epithelial cell types at 10-12pcws particularly in the large intestines (also Figure 2c). We believe that both the colonic progenitors and crypt cells (now redefined as stem cells) may give rise to the small number of colonocytes and goblet cells detected during early development. We have taken the reviewer's advice and refined "crypt" cell annotation specifying stem cell (LGR5+) versus TA cells. To provide quantitative evidence for the developmental relationship between epithelial cell types in Figure 2a, we have performed RNA velocity analysis and projected these results onto the revised UMAP (also shown below as Figure R10). This supports the bifurcation of intestinal stem cells into secretory and absorptive enterocytes/colonocytes in both fetal development and adulthood.  Figure R12: UMAP of epithelial cell types detected in single cell RNAseq dataset of (a) fetal small and large intestine and (b) pooled pediatric and adult intestinal samples. RNA velocity analysis is overlaid on UMAP. We observe REG4-expressing Goblet and enteroendocrine cell subsets (Figure R28)

that could represent the Paneth cell equivalent mentioned by the reviewer.

Figure R28: Dotplot with expression of REG4 in pediatric and adult epithelium.

Additionally, we have now provided umap plots colored by intestinal region in Supplementary 4a and below (Figure R29)

Figure R29: UMAP of fetal (top) and pooled pediatric and adult (bottom) single epithelial cells coloured by intestinal region of origin.

4.4 3. Figure 2E-G: The findings of immune receptors, especially for IgG on Tuft cells is potentially quite important, but with the present datasets, the IBD connection is quite speculative; lines 190-198 and 226-239 are not particularly informative. Rather, I interpreted pre-natal presence in the context that IgG uniquely can traverse the placenta, thereby providing

Thank you for the suggestion and please also refer to our responses in 1.8 and 3.3 above.

We agree that the link between IgG sensing on tuft cells and IBD is speculative and have now moved this in the discussion. We have also removed Fig 2d- the comparison between Crohn's and UC. We agree with the

	passive immunization generally and potential to activate Tuft cells. Post-natally, tuft cells mediate type 2 immunity, and Crohn's disease is clearly a type 1 response; the modest increase in PLCG2 (Supp Fig 3E) is not a particularly compelling/strong tie-in to IBD pathogenesis. I could not understand what points were trying to be made in Fig 2D with the Crohn's disease vs. ulcerative colitis comparison, and deleting these sections should be strongly considered, especially as this general tie-in is alluded to in their Developmental Cell 2020 manuscript.	reviewers interpretation that FCGR2A expression by a subset of tuft cells may facilitate tuft cell activation via IgG traversing the placenta during late stages of in utero development (2nd/3rd trimester), particularly since tuft cells are only detectable in our data from second trimester and we see little evidence of fetal B cell activation. During infancy, IgG in maternal milk may similarly provide passive protection. We have added this discussion to the revised manuscript: “In the context of fetal development, IgG can uniquely traverse the placenta and provide a potential route for tuft cell activation in utero. Additionally, IgG in maternal breast milk may activate tuft cells and provide passive immunity more generally during infancy. Two missense variants of PLCG2 have been linked to aberrant B cell responses in early onset IBD^{14,15} and primary immune deficiency¹⁶. It is possible that tuft cells, via PLCG2 signalling, could also contribute to IBD pathology and immune-related disease, a link that requires further investigation.”
4.5	4. The application of their datasets to Hirschsprung's disease genes (e.g. Ret, EDNRB) provides a fascinating glimpse into disease pathophysiology with interesting validation by RNAscope (Fig 3D) at 15 PCW (the failure of ganglion cell migration is believed to be prior to this timepoint). However missing in Figure 3 is a time and space summary of this migration defect (proximal to distal). At the earlier fetal timepoints, can RNAscope be applied to track proximal-to-distal migration of ganglion cells? From a translational perspective, short segment (vs. extensive) Hirschsprung's only affects the distal rectum, so it would be interesting to track a bit more precisely the timing of colonic innervation.	Please also see our response to point 3.13. Using multiple smFISH, we consistently visualise ENCC progenitors (or ganglion cells) in the submucosal layer of the developing small intestine at 6.5pcw (Figure R30). These observations are in line with previous reports that ENS colonisation in humans starts prior to 4pcw and is complete by 7pcw²¹. We would like to note that early human fetal tissue is extremely rare and the access to tissue further depends on donor consent. Future studies that use model organisms may be able to trace ENS development in more detail^{22,23}.

Figure R30: smFISH images of SOX10+ ERBB3+ MPZ+ PLP1+ ENCCs in developing human small and large intestine at 6.5 pcw.

To address the reviewer's comment, we have included smFISH imaging and a more comprehensive summary of proximal-to-distal space of ENS development in the revised manuscript (Figure 3 and Supplementary 5).

Firstly, we separate the cell from 6-11pcw and 12-17pcw to inspect the changes in differentiation trajectories during early and late fetal development (Figure R14a-b).

While at 6-11pcw, ENCCs differentiate primarily to neurons via neuroblasts, giving rise to two distinct branches- Branch A (*ETV1+*) and Branch B (*BNC2+*) (Figure R14a). This dynamic changes at 12-17pcw and becomes biased towards glial differentiation (Figure R14b).

To capture proximal to distal differentiation, we separate the cell type composition by region (Figure R14c). In addition, when tracking Hirschsprung's disease-associated gene expression, we calculate mean gene expression for small and large intestine cell subtypes separately and observe higher expression of ZEB2 and ENDRB in subsets of colonic cells, compared to small intestinal cells (Figure R26a). In addition, we observe Hirschsprung's disease-associated ligands GDNF, NRTN and EDN3 expressed by

stromal and smooth muscle cells
(Supplementary 5j and Figure R26b).

Figure R14: UMAP projection of enteric neural crest cell progenitors (ENCC) and their progeny at a) 6-11 pcw and b) 12-17pcw timepoints. The overlaid arrows show scVelo derived differentiation trajectory.

Selected marker genes are listed for each population. c) Barplot with relative abundance (percentage %) of cell types amongst ENCC-lineage populations as described in a) and b) across intestinal regions and developmental timepoints.

Figure R26: a) Heatmap with mean expression of Hirschsprung's- disease associated genes across developing neural subsets separated by region of origin (small-pink vs large- blue intestine) and developmental time point (6-11pcw- purple and 12-17pcw- yellow). b) Dotplot with key Hirschsprung's disease-associated ligand-receptor genes in the whole fetal dataset.

We now summarise these results in the revised manuscript in the section entitled "Enteric nervous system differentiation during development".

4.6 5. Figure 4. The focus on ILC populations is merited, especially given the connection to IBD GWAS genes in Figure 5G. However, no mention is made of ILC2 or ILC1 cells (Yudanin Immunity 2019). Were they not observed? There is a literature that ILC1 cells can derive

Fetal ILC2 (total 47 cells) initially did not generate a distinct Leiden cluster, but we distinguish these cells based on high expression of *HPGDS* and *PTGDR2* (Figure R31 a). We did not detect productive abTCR in the ILC2 subset using VDJ information, further supporting their ILC2 identity (Figure

from ILC3 cells over time, and certainly in IBD. Figure 4E is a lovely juxtaposition of CXCL13 (from follicular dendritic cells?), CXCR5 and RORC (LTi-like ILC3). Would be highly informative to provide a corresponding RNAScope co-segregation analysis for the LTi-like NK/T cells, especially given the anti-inflammatory nature of IL17A in Crohn's disease. Along these lines, it would be illuminating to also provide gene expression values for IL22 (epithelial protective) and IFNG (multiple cellular sources).	R31 b). As expected IL13 and IL22 were expressed by ILC2 and adult ILC3, respectively. We now include ILC2 annotation in main Figure 4 and Supplementary Fig. 6. We could not confidently call ILC1 cells in our data as canonical markers for these (TBX21, IFNG, CXCR3, IKZF3) were expressed broadly by EOMES+ NK cells (also group 1 ILCs)(Figure R31 a). This has also been observed by other groups ²⁴. Mouse studies suggest that fetal gut and fetal mesentery harbor ILC2 and ILC3 cells, but few ILC1 or NK cells ²⁵. In addition, low frequencies of ILC1 and ILC2 have been observed in human non-inflamed colon ^{26,27}. ILC1 cells are either too rare to be detected or cannot be distinguished from NK cells at transcriptional level in our data. We further note that after inspection of G D chain expression in our full length smart-seq2 data, there are no TCRαb or TCR$\gamma$$\delta$ chains in the cells previously named "LTi-like NK/T" (Figure R11b). Based on RORC and KIT expression and similarity of these cells to the adult Th17 subset, we have decided to revise their annotation to "NCR- ILC3". Also please see point 1.9 and Figure R11. These changes are now included in the main Figures 4.
---	---

Figure R11: a) UMAP visualization of T and innate-like cells in fetal, pediatric and adult samples. Dotted line denotes lymphoid tissue inducer (LTi)-like cell types. "ILC3" refers to the adult subset. b) Barplot showing TRACER analysis of TCR chains in cells captured using full-length Smart-seq2 single cell sequencing. c) Heatmap showing mean probability of cell types matching between fetal and Crohn's disease stromal and immune cell populations.

We further use smFISH imaging to visualise IL17A+ LTi-like NCR- ILC3 cells in tissue (Figure R31 c). These cells were rare in tissue sections, but were observed near other RORC- expressing cells (Figure R31 c). As the reviewer suggests, CXCL13 is the key molecule expressed by mature follicular dendritic cells. In addition to CXCL13, mLTo cells also express CCL21 and CCL19 that in adults are produced by T reticular cells. mLTo cells therefore may represent a population of fibroblasts with the ability to recruit both B and T cells.

In our data, LTi-like NCR- ILC3 cells did not

		express CXCR5, but instead showed expression of other recruitment receptors including CCR9 and CXCR6 (Figure R29 d). CXCL16, the ligand for CXCR6, was also expressed by mLT₀ as well as myeloid cells, while CCL25 (ligand for CCR9) was expressed primarily by epithelium. Therefore, it is possible that LTi-like NCR- ILC3 cells are recruited to the tissue independently from mLT₀ cells. In addition, to resolve spatial distribution of mLT₀ and ILC3 subsets in the developing human gut, we have generated and analysed 10x Visium spatial transcriptomics data on developing small intestinal regions from 13 and 17pcw gut. We use the Cell2Location algorithm ¹⁷ to map the single-cell transcriptomes to the tissue zones (Methods). In two tissue sections we observe a zone marked by CCL21, CCL19 and CXCL13 expression (Figure R31 e). As expected mLT₀ cells confidently mapped to this tissue zone. In addition, we observe mapping of LTi-like NCR+ ILC3 cells to the same tissue zone (Figure R31 f). However, LTi-like NCR- ILC3 were mapped with less confidence possibly due to their low abundance. We now include these results in the revised manuscript (Figure 5d and Supplementary Fig. 9a-c) and in the section titled “Stromal cells in human mLN and GALT development”.
--	--	---

Figure R31: a) Dotplot with expression of selected genes in ILC and NK subsets identified in this study. b) Number of productive TCR $\alpha\beta$ chains detected in fetal T/ILC cell subsets. c) Multiplex smFISH of ILCP (*SCN1B*+), LTi-like NK/T (*IL17A*+) and LTi-like ILC3 (*NCR2*+) cells in the human fetal intestine (15PCW). d) Dotplot with expression of selected recruitment molecules in fetal dataset. e) mRNA counts of genes expressed in mLT₀ cells in spatial

4.7

6. Fig 5. While it is logical to conclude the manuscript with gut to MLN connections, as presently formulated, compelling, novel and crisp conclusions are not provided. Regarding Fig 5G, not sure why they chose to test cellular

coordinates (CCL21, CCL19 and CXCL13, top panel). f) Cell abundance of mLT0 and

ILC3 subsets as estimated by cell2Location method (bottom panel) across tissue sections from 17 pcw ileum (top row) and 13 pcw ileum (bottom row). White boxes highlight developing SLO tissue zones.

We have now revised the manuscript to sharpen the conclusions of this section as follows:

“Our integrated analysis of cell lineages has highlighted equivalent cell types and cellular

enrichment for Crohn's disease as opposed to ulcerative colitis, where the roles/impact for leukocyte trafficking is stronger for the latter. While Figure 5G clearly shows substantial overlap in effects between different cell types (likely all cells contribute to some extent) the text concludes by emphasizing ILC3 cells (again logical), at the expense of Treg cells, with the latter having the highest enrichment. The role for differential expression in single cell datasets as a means of implicating cell subsets is logical, but added rigor would be attained by applying a complementary datasets (Encode, transcriptionally active regions, e.g Farh et al., Nature 2015). The Farh et al., manuscript implicates Th17 cells and not B/plasma cells particularly. Given the sparse transcriptome, it is quite possible that single cell approaches lack the sensitivity to identify key disease causing genetic variation.

networking in mLN and GALT formation during second trimester fetal development. Furthermore, the presence of the same cell signatures in pediatric CD suggests a reactivation of these programs for lymphoid tissue formation and inflammatory cell recruitment during intestinal disease.”

In our additional analysis in the revised manuscript, we recapitulate results found by Farh et al. (2015). Specifically, we show enrichment of Crohn’s-associated GWAS genes by Th17, Th1, and ILC subtypes (Figure R32a).

The advantage of our study is that scRNAseq allows us to resolve rarer and less well defined cell subsets. Here, we point specifically to the implication of ILC3 cells (not described in Farh et al.) because their enrichment of the GWAS genes implicates the involvement in lymphoid tissue formation in IBD and links to our model of recapitulation of fetal programs in SLO formation. We now explain this more clearly in the revised manuscript.

Figure R32: Top cell types across fetal, pediatric healthy and Crohn’s disease, and adult data enriched in genes associated with a) Crohn’s disease or b) Ulcerative Colitis . Cell types with FDR >10% are colored in red.

4.8	More generally, the discussion should include caveats about the lack of sensitivity of the scRNASeq approaches. Even if scRNASeq co-localizes Best4 and CFTR, validation of an exclusive co-localization by protein staining (Supp Fig 2F only shows CFTR) is not provided.	We have now included the following statement in the discussion section, in response to the reviewer's comment: "Future studies using complementary approaches, such as protein staining and chromatin accessibility studies will shed further light on these novel human cell states and their tissue architecture."
4.9	Figure 2A: did the authors mean to label the upper left-hand cluster "EEC" as opposed to "ECC"?	We have corrected it to EEC. Thank you.
4.10	4 pediatric samples listed in Figure 1A, yet multiple subsequent figures allude to 5 pediatric samples.	In total, we include 8 healthy and 7 IBD pediatric samples. We have now updated the Figure 1a to reflect these numbers.
4.11	Figure 5F. Can labels more informative than "Stromal 1", "Stromal 2" be provided, without having to refer to supplementary materials.	For consistency in the field, the stromal cells were annotated using nomenclature of equivalent colonic stromal cells ²⁸ . In response to the reviewer's comment, we have included a defining gene in the legend of the Figure 5a (e.g. Stromal 1 (ADAMDEC1+)) that consistently marked previously published subsets and those described in the this study.
4.12	Supp Figure H: were pediatric samples tested here for comparison?	Supplementary Figure 6h shows the proportion of B cell isotypes present in fetal and adult samples. Pediatric samples were not included in this analysis as these samples were processed using 3' 10x Genomics technology, not compatible with VDJ capture.
4.13	The SARS-Cov19 data (ACE2, TRMPSS) is quite interesting and important, but sensitivities with respect to stem cell infections in utero should be considered.	We thank the reviewer for highlighting this very important consideration. We have now changed our description of the data to read: "This highlights for the first time the expression of these viral entry molecules in the gut at early life." We have also removed speculation that the

		gut epithelium could be a route for in utero infection from the discussion.
4.14	Line 180: Given the absence of third trimester data, it is a bit misleading to say that Paneth cells appear post-natally.	After inspecting the fetal data for the expression of defensin (DEFA5 and DEFA6) characteristic to paneth cells (Figure R33), we find a small population of cells (122 cells) that were clustered with differentiating Goblet cells. We have now included these cells in the fetal data annotation in main Figure 2a. These results are in keeping with recent publications where similar paneth cells were captured in human embryos¹⁸. These cells may represent immature paneth cells that have been reported to appear at around 20-28 post-conception weeks, while mature paneth cells are detectable 30 post-conception weeks onwards^{18,29}.  Figure R33: Dotplot showing expression of paneth cell marker genes in the fetal epithelial subpopulations.
4.15	Line 257: DLL3 (vs. DDL3) typographical error?	We have corrected it to DLL3. Thank you.
4.16	Lines 578-9. The enteric nervous system has substantial effects beyond peristalsis.	We agree with the reviewer's comment. We have now removed this statement from the revised manuscript.

Table R1: Summary of scRNAseq studies on human and mouse gut.

Study	Region	Age	Technology	Cell type summary
Han et al., 2020 Nature	Mouse foregut	E8.5 (5–10 somites; s), E9.0 (12–15 s), and E9.5 (25–30 s)	10x Genomics Chromium Single Cell 5' Library	Definitive endoderm (further 11 major clusters with 26 stage-specific sub-clusters), Splanchnic mesoderm (further 13 major clusters with 36 stage-specific sub-clusters), cardiac, other mesoderm (somatic and paraxial), endothelium, blood, ectoderm, neural, and extraembryonic clusters
May-Zhang et al., Gastroenterology 2020	Mouse myenteric neurons. Validation in human myenteric ganglia and adjacent smooth muscle from duodenum, ileum and colon.	Mice at 6 to 7.5 weeks and at two separate times in day. Validation on organ donors 18–35 years.	Single nuclei processed by 10x Genomics 3' Chromium Single Cell library and modified version of Cel-Seq. Validation by HCR FISH and bulk RNA seq in humans.	Find 13 populations of neuronal cells as previously described in mice and additional 2 main clusters. Further subclustering suggesting 22 distinct populations. 1 glial population filtered out.
Drokhlyansky et al., 2020;	Colon cancer and normal colon proximal to the cancer of patients. Small intestine and colon of mice.	Patients were 35-90 years and of both genders. Mice were between 11-14 weeks or 50-52 weeks old and included both genders.	RAISINseq (labelled transgenic mice) with 10x Genomics 3' Chromium Single Cell and Smartseq2. MIRACL (mining rare cells)-seq (human and mouse neurons) with 10x Genomics	Mouse: 21 neuronal and 3 glia clusters. Human: 14 neuronal subsets: 4 PEMN, 5 PIMN, 2 PIN, 1 PSN, and 2 PSVN subsets

			3' Chromium Single Cell.	
Holloway et al., Cell Stem Cell; 2020	Human fetal tissue at 13.5–19 weeks gestation. HPCS-derived organoids following hindgut patterning and specification into a CDX2+ intestinal lineage.	Fetal development: liver, intestine and kidney. HIO: Day 0 (spheroid formation), 3, 7 and 14.	10x Genomics Chromium Next GEM Single Cell 3' Library v2 (lung and intestine samples) or V3 (kidney samples).	Fetal tissues: epithelium, mesenchyme, EC (7), immune, neuronal. HIO: Mesenchyme (4 clusters), epithelium (9), endothelium (1), neuronal (2).
Cao et al., Science 2020	15 organs: Adrenal gland, cerebellum, cerebrum, eye, heart, intestine, kidney, liver, lung, muscle, pancreas, placenta. Spleen, stomach, thymus, sentinel tissue.	72 to 129 days post-conception	Three-level single-cell combinatorial indexing for gene expression (sci-ATAC-seq3).	77 main cell types across tissues. Intestine: stromal, SM, Vas. EC, Lym. EC, Myeloid, Lymphoid, epithelial, mesothelial, chromaffin, erythroblasts, ENS neurons, ENS glia. Highlights NEUROG+ pancreatic islet epsilon progenitors, TPH1+ enterochromaffin cells, G, L, K and I cells, ghrelin+ EEC progenitors. Also HSPCs.
Smillie et al., 2019	68 colonic biopsies. 12 healthy, 18 UC donors	20 – 77 years	10x Genomics Chromium Single Cell 3' Library (V2 and V3)	51 epithelial, stromal, and immune cell subsets, including BEST4+ enterocytes, microfold-like cells, and IL13RA2+IL11+ inflammatory fibroblasts
Martin et al., 2019	Inflamed and uninfamed ileums, and venous blood from Crohn's disease patients	3 - 40 years	10x Genomics Chromium Single Cell 3' Library (V2)	47 clusters: stromal (fibroblasts, glial and endothelial cells) and seven distinct immune cell lineages consisting of T cells, innate lymphoid cells (ILCs), B cells, plasma cells (PCs),

	22 ileal biopsies used for single-cell sequencing.			mononuclear phagocytes (MNPs), plasmacytoid dendritic cells (pDCs), and mast cells.
Elmentaite et al., 2020 Developmental Cell	Human pediatric (terminal ileum) and early fetal (Duodenum, ileum, colon) intestine.	6-10PCW 4-12 years	10x Genomics Chromium Single Cell 3' Library (V2)	Immune, erythroblast, endothelial, neural crest, smooth muscle, mesenchymal, and epithelial cell populations
Fawkner-Corbett et al., Cell 2021	77 human intestinal samples that were collected from 17 individual embryos. Tissues: Terminal ileum, hindgut, Proximal colon, Distal colon	8-19PCW	Sample hashing and 10x Genomics Chromium Single Cell 3' Library (V3). Sample hashing	Report 101 fetal cell states. 9 intestinal compartments, annotated by transcriptional signatures—epithelial, fibroblast, endothelial (EC), pericytes, neural (ENS), muscularis, mesothelium, myofibroblast, and immune
Present study	Fetal: Proximal, middle, terminal ileum, colon, mesenteric lymph node Adult: Duodenum, Jejunum, Ileum, Appendix Caecum, ascending colon, transverse colon, descending colon, sigmoid	Fetal (7-17PCW) Adult (27-64 years)	10x Genomics Chromium Single Cell 3' (V2) and 5' (V2)	Unique cell transcriptomes described in this study: Epithelial compartment: - Fetal Tuft cells. - Fetal Microfold cells. - small intestinal BEST4+ cells. - colonic BEST2+ Goblet cells. - in vivo NEUROG3+ progenitors. - NPW+ enterochromaffin cells. Neuronal: Comparable clusters observed in Fawkner-Corbett et al., Cell 2021. However, the annotations are improved and are now based on the mouse data ³⁰ .

	colon, rectum, mesenteric lymph node			Immune:  - ILCP - LTI-like NCR- ILC3 cells - fetal ILC2 - adult CX3CR1+ CD8 T - adult Tfh cells - adult MAIT cells - adult and fetal gd T cell subsets (e.g. fetal TRDV2+TRGV9+ gd T cells) - CLP (common lymphoid progenitor) - Distinction of Pro-B cells and Immature B cells - Germinal centre B cells (light and dark zone) and FCRL4+ Memory B cells in the gut (observed in human tonsils ³¹). Myeloid:  - Lymphoid DCs (in fetal and adult). -Megakaryocytes Stromal:  - FDCs - T reticular cells - lymphatic endothelial subtypes (LEC1-6). - Stromal 1 (CCL11+) in adults. - mLN stromal cells (FMO2+). - PRG4+ mature mesothelial cells.
--	---	--	--	--

1. Siu, F. K. Y., Sham, M. H. & Chow, B. K. C. Secretin, a known gastrointestinal peptide, is widely expressed during mouse embryonic development. *Gene Expr. Patterns* **5**, 445–451 (2005).
2. Ito, G. *et al.* Lineage-specific expression of bestrophin-2 and bestrophin-4 in human intestinal epithelial cells. *PLoS One* **8**, e79693 (2013).
3. Smillie, C. S. *et al.* Intra- and Inter-cellular Rewiring of the Human Colon during

- Ulcerative Colitis. *Cell* **178**, 714–730.e22 (2019).
4. Parikh, K. *et al.* Colonic epithelial cell diversity in health and inflammatory bowel disease. *Nature* **567**, 49–55 (2019).
 5. Dann, E., Henderson, N. C., Teichmann, S. A., Morgan, M. D. & Marioni, J. C. Milo: differential abundance testing on single-cell data using k-NN graphs. *Cold Spring Harbor Laboratory* 2020.11.23.393769 (2020) doi:10.1101/2020.11.23.393769.
 6. Levine, A. S., Winsky-Sommerer, R., Huitron-Resendiz, S., Grace, M. K. & de Lecea, L. Injection of neuropeptide W into paraventricular nucleus of hypothalamus increases food intake. *Am. J. Physiol. Regul. Integr. Comp. Physiol.* **288**, R1727–32 (2005).
 7. Roberts, G. P. *et al.* Comparison of Human and Murine Enteroendocrine Cells by Transcriptomic and Peptidomic Profiling. *Diabetes* vol. 68 1062–1072 (2019).
 8. Beumer, J. *et al.* High-Resolution mRNA and Secretome Atlas of Human Enteroendocrine Cells. *Cell* **181**, 1291–1306.e19 (2020).
 9. Gehart, H. *et al.* Identification of Enteroendocrine Regulators by Real-Time Single-Cell Differentiation Mapping. *Cell* **176**, 1158–1173.e16 (2019).
 10. Beumer, J., Gehart, H. & Clevers, H. Enteroendocrine Dynamics – New Tools Reveal Hormonal Plasticity in the Gut. *Endocr. Rev.* **41**, 695–706 (2020).
 11. Jakus, Z., Simon, E., Frommhold, D., Sperandio, M. & Mócsai, A. Critical role of phospholipase C γ 2 in integrin and Fc receptor-mediated neutrophil functions and the effector phase of autoimmune arthritis. *J. Exp. Med.* **206**, 577–593 (2009).
 12. Yu, P. *et al.* Autoimmunity and inflammation due to a gain-of-function mutation in phospholipase C gamma 2 that specifically increases external Ca²⁺ entry. *Immunity* **22**, 451–465 (2005).
 13. Zeng, B. *et al.* ILC3 function as a double-edged sword in inflammatory bowel diseases. *Cell Death Dis.* **10**, 315 (2019).
 14. de Lange, K. M. *et al.* Genome-wide association study implicates immune activation of multiple integrin genes in inflammatory bowel disease. *Nat. Genet.* **49**, 256–261 (2017).
 15. Uhlig, H. H. Monogenic diseases associated with intestinal inflammation: implications for

- the understanding of inflammatory bowel disease. *Gut* **62**, 1795–1805 (2013).
16. Martín-Nalda, A. *et al.* Severe Autoinflammatory Manifestations and Antibody Deficiency Due to Novel Hypermorphic PLCG2 Mutations. *J. Clin. Immunol.* **40**, 987–1000 (2020).
 17. Kleshchevnikov, V. *et al.* Comprehensive mapping of tissue cell architecture via integrated single cell and spatial transcriptomics. doi:10.1101/2020.11.15.378125.
 18. Fawkner-Corbett, D. *et al.* Spatiotemporal analysis of human intestinal development at single-cell resolution. *Cell* **184**, 810–826.e23 (2021).
 19. Tang, C. S.-M. *et al.* Identification of Genes Associated With Hirschsprung Disease, Based on Whole-Genome Sequence Analysis, and Potential Effects on Enteric Nervous System Development. *Gastroenterology* **155**, 1908–1922.e5 (2018).
 20. Zhang, Z. *et al.* Sporadic Hirschsprung Disease: Mutational Spectrum and Novel Candidate Genes Revealed by Next-generation Sequencing. *Sci. Rep.* **7**, 14796 (2017).
 21. Lake, J. I. & Heuckeroth, R. O. Enteric nervous system development: migration, differentiation, and disease. *Am. J. Physiol. Gastrointest. Liver Physiol.* **305**, G1–24 (2013).
 22. Nowotschin, S. *et al.* The emergent landscape of the mouse gut endoderm at single-cell resolution. *Nature* **569**, 361–367 (2019).
 23. Pijuan-Sala, B. *et al.* A single-cell molecular map of mouse gastrulation and early organogenesis. *Nature* **566**, 490–495 (2019).
 24. Martin, J. C. *et al.* Single-Cell Analysis of Crohn's Disease Lesions Identifies a Pathogenic Cellular Module Associated with Resistance to Anti-TNF Therapy. *Cell* **178**, 1493–1508.e20 (2019).
 25. Innate Lymphoid Cell Development: A T Cell Perspective. *Immunity* **48**, 1091–1103 (2018).
 26. Forkel, M. *et al.* Distinct Alterations in the Composition of Mucosal Innate Lymphoid Cells in Newly Diagnosed and Established Crohn's Disease and Ulcerative Colitis. *J. Crohns. Colitis* **13**, 67–78 (2019).
 27. Mazzurana, L. *et al.* Tissue-specific transcriptional imprinting and heterogeneity in

human innate lymphoid cells revealed by full-length single-cell RNA-sequencing. *Cell Res.* (2021) doi:10.1038/s41422-020-00445-x.

28. Kinchen, J. *et al.* Structural Remodeling of the Human Colonic Mesenchyme in Inflammatory Bowel Disease. *Cell* **175**, 372–386.e17 (2018).
29. Lueschow, S. R. & McElroy, S. J. The Paneth Cell: The Curator and Defender of the Immature Small Intestine. *Front. Immunol.* **11**, 587 (2020).
30. Morarach, K. *et al.* Diversification of molecularly defined myenteric neuron classes revealed by single-cell RNA sequencing. *Nat. Neurosci.* **24**, 34–46 (2021).
31. King, H. W. *et al.* Single-cell analysis of human B cell maturation predicts how antibody class switching shapes selection dynamics. *Sci Immunol* **6**, (2021).

Reviewer Reports on the First Revision:

Referee #1 (Remarks to the Author):

The authors have carefully addressed all my previous concerns and strengthened their highly valuable resource, which is now ready for publication.

Referee #3 (Remarks to the Author):

The authors have responded satisfactory to most concerns. However some issues still remain as summarized below. Overall I commend the authors for their extensive efforts in assembling, analyzing and presenting this extensive human gut cell atlas, which is likely to become a valuable resource.

Comment 3)

The authors have been responsive and presented their data with more sound interpretation. -It could be wise to revise the result section slightly to improve clarity. For instance, the detection of *plcg2* downstream mediators are only mentioned “matter of factly” and not exemplified with gene names in the text or with reference to picture (Page 4 175-177).

-In figure legend, please amend to: i) “Summary schematic of proposed signaling pathways in tuft cells”. No experiments have verified any of the suggested interactions in tuft-cells yet.

-For the experiment showing increased tuft cells, *Fcgr2a* and *Plcg2* in differentiated human organoid line, please show the individual data-points in the bar plot and indicate how many

replicates/experiments (n) were included in the calculation.

Comment 5 (ENS)

The separation of the datasets into two ranges of stages have led to clearer trajectories/clusters. The nomenclature into Branch A and B according to previous murine characterization is good for the future clarity in the ENS field. Some other neuron class nomenclature appear to have been solved nicely. Thank you for the extensive gene expression on UMAPs for validation of your reasoning/claims.

I would only like to make a few changes:

Comment 8/9

a) Branch A –two different clusters are specified (denoted IPAN/IN and eMN). Both these clusters are defined as ENC12 neurons in the murine dataset (number of cells was too few to subcluster). As in mouse, both your cell groups express *Ntng1*. It was demonstrated in Morarach et al., 2021 (Fig. 4o) that no *Ntng1* fibers are innervating the muscle - hence none of these clusters are likely excitatory motor neurons. If this is the case in human remains to be explored, but there is also no evidence to classify this group as excitatory MN. These genes are for example expressed in many types of interneurons in the CNS where they modify excitation of other neurons. In the mouse both these enteric neuron groups express *Piezo2* (partly) and could thus be IPANs/INs. The equivalent neuron types in the Drokhylyansky atlas are PIN3 and PSN3 - as you can see they form a similar UMAP pattern as in juvenile mouse ENS. Please amend this nomenclature to IPAN/IN for both Branch A “tip-clusters” if you want to assign function to these neurons. Perhaps you could use some numbering system to distinguish the clusters within the branches (for instance: Branch A1 (iMN), Branch A2 (IPAN/IN), Branch A3 (IPAN/IN)). Please also amend the result text, supplementary figures and tables accordingly. Finally – the high-lighted genes for these tip clusters do not correspond with the dotplots/differentially expressed gene lists (for instance *Grp* is stronger in IPAN/IN – see also R2: 3 below).

b) The most typical markers in mouse for excitatory motor neurons are Calretinin (in juvenile/adults), and Enkephalin (*Penk*), but its known that both enkephalin and Calretinin are expressed in subsets, not all eMN and may not be exclusively expressed in eMN (Reiche et al., 1998 *Dtsch Tierarztl Wochenschr* 105; and Qu et al., 2008 *Cell and Tissue Research*). The recent human ENS atlas also suggest that there are *Penk*/*Tac* high (e.g. PEMN2) and low (e.g. PEMN3) excitatory motoneurons. Hence, not only the strong *Penk*+ neurons are to be considered as excitatory motor neurons. Morarach et al suggest that *Ndufa4l2* is an early marker of excitatory motor neurons

(although not expressed in all cells) which stays on in the adult ENS mainly in ENC1-2,4. Bnc2 is also indicative of excitatory MN (although some IN also express Bnc2 in the adult). The Drokhylyansky human atlas also indicate Bnc2 as an excitatory MN marker. I would suggest that both Branch B clusters at pcw 6-11 are denoted excitatory motor neurons (they likely correspond most closely to ENC1-3). They could be called BranchB1 (eMN), Branch B2 (eMN). It is also clear that a subset of your Branch B is immature still, so it is understandably hard to fit them into a neuron class. Studies in the mouse (Morarach et al., 2021) suggest that there is significant differentiation at the postmitotic stage, even leading to phenotypic switches - hence when considering newly born neurons - the final identity is not trivial to ascertain. Especially at the older stages, some of newly born neurons might be developing submucosal neurons that have not started to express their unique markers yet. But it is fine to leave this information out from the manuscript. Adjust the nomenclature in all relevant documents including the supplementary figures and tables. Remove or adjust markings of excitatory marker genes in Supplementary Fig5 (perhaps this is not necessary information, especially as functional studies in human to link these markers to functional groups are still lacking).

c) With regards to enriched genes indicated as Branch specific: Is Ramp1 a very selective marker for the whole Branch B, and how wide-spread is Kcnj5 in Branch A? Bnc2 and Etv1 appear selective and robust for almost the whole branches (although Bnc2 is downregulated in Branch B IPAN, as in mouse). If you want to compare to murine ENS, Kcnj5 is not Branch-specific in this organism. See above for alternative markers to be considered if they indeed are enriched sufficiently. Ramp1 appears however rather specific to the beginning of Branch B and could be kept as a marker for this cluster (which I suggest you rename eMN or immature eMN as discussed above).

d) Branch A at 12-17 pcw contains the Neurod6+ population, a suggested interneuron. Think about the nomenclature in relation to above, whether some numbering would make things clearer (for example denoting these Branch A4 (IN)).

e) At 12-17pcw you do not indicate any Branch A IPAN/IN or eMN. Those cells could be too few and was outnumbered by the increasing glia and MN populations. Should this be mentioned somewhere (perhaps in figure legend in the lack of space in main text)? It is otherwise a bit confusing to the reader what happens with these neuron types at older ages.

Comment 11) Regarding glia/progenitors, it looks much cleaner with the separation of datasets into two ranges and easier to interpret. The further validation of the three classes of glia in tissue is

greatly appreciated.

Comment 12) Validation of genes in tissue in previous developing human ENS atlas: the reference is not included in the revised manuscript, but you also have removed the discussion on Gdf10, Tgfb and so on in the text. Please consider though that this gene expression atlas contains the validated immunohistochemical expression of 31 transcription factors and 19 ligand/receptors in the ENS of both intestine and stomach at week 5-6 and 7-10, matching your first range of scRNA-seq data-analysis. Referring to this paper as validating complementing resource could be very valuable for the reader.

Comment 13) The presentation and discussion regarding Hirschsprung's disease genes are also now more sound and interesting. Please cite a more general HSCR review or paper for the selection of HSCR risk genes in addition to the already selected. For instance Bondurand and Southard-Smith 2016 Dev. Biol.

Comment 14) Thank you for the up-to-date list of current scRNA-seq datasets of human gut cells. I would expect this summary to be extremely useful for the field - consider to include it as an extra supplementary Table that you could cite in relation to the sentence at Page 2, row 79-81. The extensive description of all datasets as well as the precise list of unique tissues/celltypes/analysis only obtained from your study is very valuable for the people in the gut field in guiding them to the right data-set for their downstream analysis.

Additional issues in the revised manuscript:

R2:1) Page 5, Row 193-194. "Premature differentiation of ENCCs leads to Hirschsprung's disease".

There are many mechanisms that appear to contribute to the HSCR phenotype, one of them premature differentiation, but also aberrant migration, proliferation, survival and neurite outgrowth – all together leading up to insufficient cell numbers in the distal colon. While you don't need to explain in detail, please widen the causes to HSCR.

The rationale to choose the two references are also not clear. One refers to migrational defect following an Ernb-mutation, and the other characterize a plausible risk gene for Wardenburg syndrome (which have HSCR amongst others). I suggest to cite a recent review for example: Bondurand and Southard-Smith 2016 Dev. Biol.

R2:2) The title for Figure 3 could be broader phrased. The detailed description of ENS differentiation in human is more interesting and novel than the expression of HSCR risk genes (the genes/cells

behind the dominant HSCR phenotype aganglionosis is well understood, although gene expression shown here could also explain additional subtler defects manifesting at later stages). In addition, please define IN: Interneuron

R2: 3) It is unclear how the genes were picked to denote the different clusters (Fig 3A-B). In many cases the genes are not unique and equally much expressed in other clusters. For instance, Branch B is labelled with "Bnc2", while Branch B (eMN) also express this gene (although it is lower at later stages, its not a good distinguishing gene between the two clusters). Likewise, the distinction between Branch A (eMN) and Branch A (IPAN/IN) is not portrayed well in the Figure. Both clusters show Grp, and eMN even at higher levels, despite this Grp is denoted as an IPAN/IN marker. See above also discussion about these two clusters and alternative interpretations. A suggestion are: Ntng1/Grp/Tac1/Penk and Ntng1/Grp/Nxph2 as marker genes. As Ntng1 and Etv1 are common marker for the two, but are not expressed in the Branch A they would additionally help you separate the Branch A Tac1/Penk/Ntng1(Etv1) from Branch B Tac1/Penk/Bnc2 populations. Thus, in the cases where a key gene is expressed in several clusters but anyway worth high-lighting it would be better to show it in several clusters and instead focusing on the combinatorial gene-code differences. These are just examples I detected - please, carefully scrutinize the marker genes indicated in Figure/mentioned in the text and pick as selective genes as possible (and/or show combinatorial profiles to avoid confusion).

R2: 5) Suggested rephrasing: Page 5; 216-217: "While differentiated neuronal cell types were abundant at 6-11 PCW, glial cell types were enriched at later stages (12-17 PCW)."

R2: 6) Suggested rephrasing Page 6; 246-258: "Our analysis shows broad expression of Hirschsprung's disease-associated genes across neural and ICC/SMC cells, implicating these gut cell types in disease pathogenesis." In addition, this is not a summary of the whole section and could be moved up as an ending of the previous paragraph.

R2: 7) Does Figure 3e show expression in the myenteric plexus or mesentery? The explanation "outside the myenteric plexus" (in figure legend) is a bit unclear.

R2: 8) Please correct the citing of Figure 3 in the results section, most are not updated.

R2: 9) Carefully check the nomenclature in Figure 3 and make sure the same names are used in a-b as f (for example ENCC is not used in the a-b, but in f). Also recheck the supplementary tables for

the nomenclature used.

R2:10) The difference in Ret expression between small (SI) and large intestine (LI) is with the revised clusters less pronounced, and it is doubtful whether it is worth high-lighting. Branch A (IN) appears even to have higher expression in SI than LI. The most interesting is probably your first sentence, that Ret remains in Branch A, but is downregulated in Branch B, progenitors and glia. The comparison between SI and LI with regards to Zeb2 and Ednrb is ok, but rephrase the sentence so that it is clear that you compare SI and LI (it was not in the latest revision). Note however that LI is always going to lag behind SI in its development due to the rostral-caudal colonization of the gut, and general differentiation of the embryo in an rostral to caudal fashion, so the changes might be related to this phenomena. Although then Sox10 would also be enriched in LI, which it is not so your observed LI-SI differences may be unrelated to the general differentiation.

R2:11) Some mistakes in Figure 3:

Please go through the gene names to make sure there are no typos.

-Branch B says Dlx1 at pcw12-17 while it Dlx5 at pcw 6-11. Is this correct or should it be Dlx5 at both stages.

-Neurog6 in Branch A (IN) – should it be Neurod6?

Referee #4 (Remarks to the Author):

The investigators have performed a basic re-analysis of some of the front part of the manuscript, added and clarified sample numbers and added new spatial transcriptomics and smFISH data. In general, moderate changes to their data visualization have simplified and/or clarified some of their major messages, including improved clarity wrt their choices for their ligand receptor mappings. They have restricted their Hirschsprung's associated gene list, and re-formulated/re-named some of their clusters. I agree with the author's statement that tracking fetal time-course with respect to proximal to distal migration likely should be held for a more detailed Hirschsprung's focused effort, given the difficulty of fetal tissue capture. The relegation of the (often rare subsets) in CD vs. UC is appropriate (given the substantial overlap between many clusters), with a space-efficient summarizing heat-map kept in Fig 5. This comprehensive and comprehensively referenced gut atlas will represent a "must-review" primer for GI, neurologic and immunologic development and disease for advanced level trainees. I have a few remaining questions and concerns.

Major:

1. Figure 1 overview

1. 1b can likely be compressed for space. Much of this real estate is not that informative.
2. In reviewing the fascinating complexity of the enteroendocrine cells, I wonder to what extent the very early fetal timepoints might reflect contamination from the pancreatic bud (acknowledged for insulin-producing beta, endocrine cells in line 148-9) for exocrine cell subsets as well. In very early development, not sure how easy to dissect apart these organs.
3. 1d. It seems rather unlikely that in the adult samples the fractions of epithelial cells range from < 2% (duodenum, cecum, rectum) to well over 15% (Sigmoid). I think the general concept that transmural vs. biopsy samples results in lower fractions of epithelial cells seems right, however, as presented, I feel that this bar-graph is misleading especially for the within adult comparisons. Is it possible that differences in sampling capture efficiencies account for this? The doublet removals (especially problematic for epithelial cells and given the high fraction of cell removal generally)?
4. (Minor) 1d. Consider removing the color (or restricting to epithelial green), so as to not confuse with 1b/c
5. 1f. Legend doesn't specify what the red highlights designate. Not sure what the basis for transcript inclusion is for this gene list

2. Figure 2. The RNA velocity analyses substantially strengthen the opening to this Figure. Figure 2a (fetal) vs. b (ped/adult). I may be mis/over-interpreting the scVelo analyses, but how do the authors interpret the M cell differences between pre- vs. post-natal cells? While the post-natal analyses appear to make sense (M cells from stem cells), the M cells in fetal appear far from the stem cells. This would appear to be in conflict with de Lau et al's findings (PMID 22778137). Does this reflect pre- vs. post-natal differences?

3. ILC origins. This may be beyond the scope of this manuscript, given the small number of ILC2 cells identified, but does appear generally relevant given the tie-in to IBD in Fig 5. Bielecki et al., (PMID 33536623) show in skin that tissue resident innate lymphoid cells are sufficient to drive inflammation/pathogenicity, implying conversion from ILC2 to ILC3. Do any of the present data support this model (small numbers of ILC2 due to transition toward ILC3), or do the authors think this merely reflects organ differences between skin and gut?

4. Data Sharing and Supplementary Materials. The authors have provided exhaustive figures and tables for the community, which will be incredibly valuable for the community at large.

Minor

1. Consistency wrt number of different anatomic sites sampled: 10 (line 51) vs. 11 (line 84). Figure

1a designates even more locations. Should be consistent or qualify in place the number of sites.

2. Line 457: change Nerog3+ to Neurog3+

3. Line 260: Change parentheses to (beta chain of gut-specific $\alpha 4\beta 7$ integrins)

Author Rebuttals to First Revision:

We thank the editor and reviewers for their time in considering our manuscript. A point-by-point response to reviewers' comments is detailed below. Additional text is highlighted in blue and revised changes are highlighted in red.

	Comments	Responses
1.1	Referee #3 (Remarks to the Author): The authors have responded satisfactory to most concerns. However some issues still remain as summarized below. Overall I commend the authors for their extensive efforts in assembling, analyzing and presenting this extensive human gut cell atlas, which is likely to become a valuable resource. Comment 3) The authors have been responsive and presented their data with more sound interpretation. -It could be wise to revise the result section slightly to improve clarity. For instance, the detection of plcg2 downstream mediators are only mentioned “matter of factly” and not exemplified with gene names in the text or with reference to picture (Page 4 175-177). -In figure legend, please amend to: i) “Summary schematic of proposed signaling pathways in tuft cells”. No experiments have verified any of the suggested interactions in tuft-cells yet. -For the experiment showing increased tuft cells, Fcgr2a and Plcg2 in differentiated human organoid line, please show the individual data-points in the bar	We thank the reviewer for recognising the effort we have made in revising the manuscript and for their additional comments. We have now exemplified the genes involved in PLCG2 signaling in tuft cells in text as follows: “... the expression of FCGR2A and downstream signalling mediators including RAC2, ITPR2, PRKCA and TRPM5 (Figure 2h & j) led us to wonder whether tuft cells specifically could be responsive to signals from immune cells.” We have changed the legend for Figure 2i as suggested by the reviewer. We also provide individual data-points for the human organoid experiments in Supplementary Fig. 4 k-l and in Figure R1. The experiments were performed in triplicates using two different organoid lines from two healthy pediatric patients.  Figure R1: k-l) Bar plot of relative expression of LGR5 and MUC2 (k) and Tuft cell key mRNA (l) by intestinal differentiated or undifferentiated organoids.

	plot and indicate how many replicates/experiments (n) were included in the calculation.	
1.2	Comment 5 (ENS) The separation of the datasets into two ranges of stages have led to clearer trajectories/clusters. The nomenclature into Branch A and B according to previous murine characterization is good for the future clarity in the ENS field. Some other neuron class nomenclature appear to have been solved nicely. Thank you for the extensive gene expression on UMAPs for validation of your reasoning/claims. I would only like to make a few changes: Comment 8/9 a) Branch A –two different clusters are specified (denoted IPAN/IN and eMN). Both these clusters are defined as ENC12 neurons in the murine dataset (number of cells was too few to subcluster). As in mouse, both your cell groups express Ntng1. It was demonstrated in Morarach et al., 2021 (Fig. 4o) that no Ntng1 fibers are innervating the muscle - hence none of these clusters are likely excitatory motor neurons. If this is the case in human remains to be explored, but there is also no evidence to classify this group as excitatory MN. These genes are for example expressed in many types of interneurons in the CNS where they modify excitation of other neurons. In the mouse both	We have renamed the neuronal populations to Branch A1 (iMN), Branch A2 (IPAN/IN), Branch A3 (IPAN/IN) as suggested. We also amended the relevant text, Figures (Supplementary Fig. 2, Fig. 5) and Tables. For Branch A subsets, we amended the highlighted genes in Figure 3a,b as follows: Branch A1 (iMN)- ETV1/ GAL/ SCGN/ NOS1/ SPOCK1 Branch A2 (IPAN/IN)- ETV1/ NTNG1/ GRP/ CHRM3/ NXPH2. Branch A3 (IPAN/IN)- ETV1/ NTNG1/ GRP/ TAC1/ RUNX1/ ONECUT3. Branch A4 (IN)- ETV1/ SCGN/ GCGR/ NEUROD6 and provide relevant plots in Figure R2 for your convenience:  Figure R2: Neural subsets during development at a) 6-11pcw and b) 12-17 pcw with revised

	these enteric neuron groups express Piezo2 (partly) and could thus be IPANs/INs. The equivalent neuron types in the Drokhylyansky atlas are PIN3 and PSN3 - as you can see they form a similar UMAP pattern as in juvenile mouse ENS. Please amend this nomenclature to IPAN/IN for both Branch A “tip-clusters” if you want to assign function to these neurons. Perhaps you could use some numbering system to distinguish the clusters within the branches (for instance: Branch A1 (iMN), Branch A2 (IPAN/IN), Branch A3 (IPAN/IN)). Please also amend the result text, supplementary figures and tables accordingly. Finally – the high-lighted genes for these tip clusters do not correspond with the dotplots/differentially expressed gene lists (for instance Grp is stronger in IPAN/IN – see also R2:3 below).	highlighted marker genes. c) Dotplot with genes highlighted in a-b.
1.3	b) The most typical markers in mouse for excitatory motor neurons are Calretinin (in juvenile/adults), and Enkephalin (Penk), but its known that both enkephalin and Calretinin are expressed in subsets, not all eMN and may not be exclusively expressed in eMN (Reiche et al., 1998 Dtsch Tierarztl Wochenschr 105; and Qu et al., 2008 Cell and Tissue Research). The recent human ENS atlas also suggest that there are Penk/Tac high (e.g. PEMN2) and low (e.g. PEMN3) excitatory motoneurons. Hence, not only the strong Penk+ neurons are to be considered as excitatory	We have amended branch B neuron nomenclature as Branch B1 (eMN), Branch B2 (eMN) and Branch B3 (IPAN), as suggested, and adjusted the relevant text, figures and tables. We also clarify Branch B1 and B2 immaturity in text as follows: “Branch B further differentiated into immature excitatory motor neuron (eMN) subsets (Branch B1 and B2) that correspond most closely to murine ENC1-3 (Figure 3a).” We also revised genes for Branch B as follows and in Figure R2: Branch B1 (eMN)- BNC2/ RAMP1/ NXPH4/ NDUFA4L2. Branch B2 (eMN)- BNC2/ PENK/ HTR7. Branch B3 (IPAN)- BNC2/ DLX3/ DGKG.

motor neurons. Morarach et al suggest that *Ndufa4l2* is an early marker of excitatory motor neurons (although not expressed in all cells) which stays on in the adult ENS mainly in ENC1-2,4. *Bnc2* is also indicative of excitatory MN (although some IN also express *Bnc2* in the adult). The Drokhylyansky human atlas also indicate *Bnc2* as an excitatory MN marker. I would suggest that both Branch B clusters at pcw 6-11 are denoted excitatory motor neurons (they likely correspond most closely to ENC1-3). They could be called BranchB1 (eMN), Branch B2 (eMN). It is also clear that a subset of your Branch B is immature still, so it is understandably hard to fit them into a neuron class. Studies in the mouse (Morarach et al., 2021) suggest that there is significant differentiation at the postmitotic stage, even leading to phenotypic switches - hence when considering newly born neurons - the final identity is not trivial to ascertain. Especially at the older stages, some of newly born neurons might be developing submucosal neurons that have not started to express their unique markers yet. But it is fine to leave this information out from the manuscript. Adjust the nomenclature in all relevant documents including the supplementary figures and tables. Remove or adjust markings of excitatory marker genes in Supplementary Fig5 (perhaps this is not necessary information, especially as functional studies in human to

Figure R2: Neural subsets during development at a) 6-11pcw and b) 12-17 pcw with revised highlighted marker genes. c) Dotplot with genes highlighted in a-b.

	link these markers to functional groups are still lacking).	
1.4	c) With regards to enriched genes indicated as Branch specific: Is Ramp1 a very selective marker for the whole Branch B, and how wide-spread is Kcnj5 in Branch A? Bnc2 and Etv1 appear selective and robust for almost the whole branches (although Bnc2 is downregulated in Branch B IPAN, as in mouse). If you want to compare to murine ENS, Kcnj5 is not Branch-specific in this organism. See above for alternative markers to be considered if they indeed are enriched sufficiently. Ramp1 appears however rather specific to the beginning of Branch B and could be kept as a marker for this cluster (which I suggest you rename eMN or immature eMN as discussed above).	RAMP1 and BNC2 were both similarly enriched in Branch B, and RAMP1 was more specific to Branch B neurons than BNC2 (Figure R3). We have decided to highlight BNC2 based on their similarity to murine Branch B neurons (Monarch et al., 2021) and humans (Drokhylansky et al., 2020). Similarly, we highlighted ETV1 for species similarity and KCNJ5 as a more selective marker for Branch A neurons (Figure R3). However, because the KCNJ5 gene is not conserved across species as suggested by the reviewer, we do not mention this gene in the Figure 3 (Figure R2).  Figure R3: Expression of BNC2, RAMP1, ETV1 and KCNJ5 in developing neurons 6-11 and 12-17PCW samples.

		Figure R2: Neural subsets during development at a) 6-11pcw and b) 12-17 pcw with revised highlighted marker genes. c) Dotplot with genes highlighted in a-b.
1.5	d) Branch A at 12-17 pcw contains the Neurod6+ population, a suggested interneuron. Think about the nomenclature in relation to above, whether some numbering would make things clearer (for example denoting these Branch A4 (IN)).	Thank you for this suggestion, we have renamed this population to Branch A4 (IN) in the revised manuscript.
1.6	e) At 12-17pcw you do not indicate any Branch A IPAN/IN or eMN. Those cells could be too few and was outnumbered by the increasing glia and MN populations. Should this be mentioned somewhere (perhaps in figure legend in the lack of space in main text)? It is otherwise a bit confusing to the reader what happens with these neuron types at older ages.	Thank you, we address this in the legend of revised Figure 3: “Branch A2 and A3 (IPAN/IN) subsets were not observed at 12-17 PCWs, possibly because these subsets were outnumbered by the increasing glial populations.”

1.7	Comment 11) Regarding glia/progenitors, it looks much cleaner with the separation of datasets into two ranges and easier to interpret. The further validation of the three classes of glia in tissue is greatly appreciated.	We agree with the reviewer, many thanks for their useful suggestion.
1.8	Comment 12) Validation of genes in tissue in previous developing human ENS atlas: the reference is not included in the revised manuscript, but you also have removed the discussion on Gdf10, Tgfb and so on in the text. Please consider though that this gene expression atlas contains the validated immunohistochemical expression of 31 transcription factors and 19 ligand/receptors in the ENS of both intestine and stomach at week 5-6 and 7-10, matching your first range of scRNA-seq data-analysis. Referring to this paper as validating complementing resource could be very valuable for the reader.	Thank you, we include this reference in text as follows: “Expression of transcription factors (ETV1) and other signalling molecules were previously validated in situ in a complementary resource of the human gut. ²³ ”
1.9	Comment 13) The presentation and discussion regarding Hirschsprung’s disease genes are also now more sound and interesting. Please cite a more general HSCR review or paper for the selection of HSCR risk genes in addition to the already selected. For instance Bondurand and Southard-Smith 2016 Dev. Biol.	Thank you, we include Bondurand and Southard-Smith 2016 Dev. Biol. reference as follows: “To identify neural cells involved in Hirschsprung’s disease, we screened for expression of known Hirschsprung’s disease-associated genes (Tang et al. 2018 ; Zhang et al. 2017 ; Bondurand and Southard-Smith 2016).”
1.10	Comment 14) Thank you for the up-to-date list of current scRNA-seq datasets of human gut cells. I would expect this summary to	Thank you, we have included this table as Table 1 and cite it as suggested by the reviewer.

	be extremely useful for the field - consider to include it as an extra supplementary Table that you could cite in relation to the sentence at Page 2, row 79-81. The extensive description of all datasets as well as the precise list of unique tissues/celltypes/analysis only obtained from your study is very valuable for the people in the gut field in guiding them to the right data-set for their downstream analysis.	
1.11	Additional issues in the revised manuscript: R2:1) Page 5, Row 193-194. “Premature differentiation of ENCCs leads to Hirschprung’s disease”. There are many mechanisms that appear to contribute to the HSCR phenotype, one of them premature differentiation, but also aberrant migration, proliferation, survival and neurite outgrowth – all together leading up to insufficient cell numbers in the distal colon. While you don’t need to explain in detail, please widen the causes to HSCR. The rationale to choose the two references are also not clear. One refers to migrational defect following an Ernb-mutation, and the other characterize a plausible risk gene for Wardenburg syndrome (which have HSCR amongst others). I suggest to cite a recent review for example: Bondurand and Southard-Smith 2016 Dev. Biol.	Thank you, we have included a wider description of causes of HSCR in the text, and have updated the associated reference: “Insufficient numbers of ENCCs in the distal colon due to numerous differentiation, migration, proliferation or survival defects can lead to Hirschsprung's disease.” We have replaced the two cited papers with Bondurand and Southard-Smith 2016 Dev. Biol. as suggested.

1.12	R2:2) The title for Figure 3 could be broader phrased. The detailed description of ENS differentiation in human is more interesting and novel than the expression of HSCR risk genes (the genes/cells behind the dominant HSCR phenotype aganglionosis is well understood, although gene expression shown here could also explain additional subtler defects manifesting at later stages). In addition, please define IN: Interneuron	Thank you for this suggestion. We re-titled the Figure 3 to “Cells of the developing enteric nervous system”. We have also defined IN as an interneuron in this figure legend.
1.13	R2:3) It is unclear how the genes were picked to denote the different clusters (Fig 3A-B). In many cases the genes are not unique and equally much expressed in other clusters. For instance, Branch B is labelled with “Bnc2”, while Branch B (eMN) also express this gene (although it is lower at later stages, its not a good distinguishing gene between the two clusters). Likewise, the distinction between Branch A (eMN) and Branch A (IPAN/IN) is not portrayed well in the Figure. Both clusters show Grp, and eMN even at higher levels, despite this Grp is denoted as an IPAN/IN marker. See above also discussion about these two clusters and alternative interpretations. A suggestion are: Ntng1/Grp/Tac1/Penk and Ntng1/Grp/Nxph2 as marker genes. As Ntng1 and Etv1 are common marker for the two, but are not expressed in the Branch A they would additionally help you separate the Branch A Tac1/Penk/Ntng1(Etv1) from	We provide an updated list of highlighted genes including the genes suggested by the reviewer and their expression in the dotplot in Supplementary Fig 2. d) and Figure R2 below.  Figure R2: Neural subsets during development at a) 6-11pcw and b) 12-17 pcw with revised highlighted marker genes. c) Dotplot with genes highlighted in a-b.

	Branch B Tac1/Penk/Bnc2 populations. Thus, in the cases where a key gene is expressed in several clusters but anyway worth high-lighting it would be better to show it in several clusters and instead focusing on the combinatorial gene-code differences. These are just examples I detected - please, carefully scrutinize the marker genes indicated in Figure/mentioned in the text and pick as selective genes as possible (and/or show combinatorial profiles to avoid confusion).	
1.14	R2:5) Suggested rephrasing: Page 5; 216-217: “While differentiated neuronal cell types were abundant at 6-11 PCW, glial cell types were enriched at later stages (12-17 PCW).”	Thank you, we have now re-phrased this sentence as suggested by the reviewer.
1.15	R2:6) Suggested rephrasing Page 6; 246-258: “Our analysis shows broad expression of Hirschsprung’s disease-associated genes across neural and ICC/SMC cells, implicating these gut cell types in disease pathogenesis.” In addition, this is not a summary of the whole section and could be moved up as an ending of the previous paragraph.	Thank you we updated this as follows: “Here we mapped the differentiation of neural cells from the precursor cell states, showing that neurogenesis precedes gliogenesis. In addition, our analysis shows broad expression of Hirschsprung’s disease-associated genes across neural and ICC/SMC cells, implicating these cell subsets in pathogenesis.”
1.16	R2:7) Does Figure 3e show expression in the myenteric plexus or mesentery? The explanation “outside the myenteric plexus” (in figure legend) is a bit unclear.	Figure 3e imaging shows DHH+ cells in the mesentery. We have clarified this in the figure legend. Thank you.

1.17	R2:8) Please correct the citing of Figure 3 in the results section, most are not updated.	We corrected it in text, thank you.
1.18	R2:9) Carefully check the nomenclature in Figure 3 and make sure the same names are used in a-b as f (for example ENCC is not used in the a-b, but in f). Also recheck the supplementary tables for the nomenclature used.	We have now unified the nomenclature in Figure 3 and related Supplementary Figures, thank you.
1.19	R2:10) The difference in Ret expression between small (SI) and large intestine (LI) is with the revised clusters less pronounced, and it is doubtful whether it is worth high-lighting. Branch A (IN) appears even to have higher expression in SI than LI. The most interesting is probably your first sentence, that Ret remains in Branch A, but is downregulated in Branch B, progenitors and glia. The comparison between SI and LI with regards to Zeb2 and Ednrb is ok, but rephrase the sentence so that it is clear that you compare SI and LI (it was not in the latest revision). Note however that LI is always going to lag behind SI in its development due to the rostral-caudal colonization of the gut, and general differentiation of the embryo in an rostral to caudal fashion, so the changes might be related to this phenomena. Although then Sox10 would also be enriched in LI, which it is not so your observed LI-SI differences may be unrelated to the general differentiation.	Thank you. We have changed the main text as follows: “Interestingly, ZEB2 and EDNRB were more highly expressed across colonic glia and neuroblast cells compared to equivalent small intestinal subsets (Figure 3f). Any differences in expression between regions may also be due to the developmental lag of the large intestines. In addition, key ligands implicated in Hirschsprung’s disease, including GDNF, NRTN and EDN3 were primarily expressed by mesothelium, smooth muscle cells (SMC) and interstitial cells of Cajal (ICC) (Supplementary Fig. 5j), as described previously.”

1.20	R2:11) Some mistakes in Figure 3: Please go through the gene names to make sure there are no typos. -Branch B says Dlx1 at pcw12-17 while it Dlx5 at pcw 6-11. Is this correct or should it be Dlx5 at both stages. -Neurog6 in Branch A (IN) – should it be Neurod6?	Apologies for the typos, we have now corrected this in Figure 3.
2.1	Referee #4 (Remarks to the Author): The investigators have performed a basic re-analysis of some of the front part of the manuscript, added and clarified sample numbers and added new spatial transcriptomics and smFISH data. In general, moderate changes to their data visualization have simplified and/or clarified some of their major messages, including improved clarity wrt their choices for their ligand receptor mappings. They have restricted their Hirschsprung's associated gene list, and re-formulated/re-named some of their clusters. I agree with the author's statement that tracking fetal time-course with respect to proximal to distal migration likely should be held for a more detailed Hirschsprung's focused effort, given the difficulty of fetal tissue capture. The relegation of the (often rare subsets) in CD vs. UC is appropriate (given the substantial overlap between many clusters), with a space-efficient summarizing heat-map	We thank the reviewer for their positive comments about the revised manuscript. We appreciate the suggestion to reorganise Figure 1 to contain essential information. We do think that overview of the dataset and cells covered in the manuscript helps to orientate the readers less familiar with single-cell sequencing analysis, have have now compressed this plot and provide the updated Figure 1 below.  Figure 1: Intestinal cellular census throughout life a) Schematic of tissue sampling of healthy human gut. Number of individual donors sampled for scRNA-seq throughout the paper are shown in parentheses. b) UMAP visualisation of cellular landscape of the human intestinal tract coloured by cellular lineage. c) Relative proportions of cell lineage as in (b) per developmental stage. d) Proportions of BEST4+ enterocytes among all epithelial cells

	kept in Fig 5. This comprehensive and comprehensively referenced gut atlas will represent a "must-review" primer for GI, neurologic and immunologic development and disease for advanced level trainees. I have a few remaining questions and concerns. Major:  Figure 1 overview 1b can likely be compressed for space. Much of this real estate is not that informative. 	in scRNA-seq data per tissue region and developmental stage. e) Expression of BEST4 (antibody: HPA058564) and CSTE (antibody: CAB032687) in gut histological sections from Human Protein Atlas (proteinalas.org). f) Dotplot with relative expression of selected marker genes within BEST4+ epithelial cells from small and large intestines and different ages as in (d). Genes highlighted in red are discussed in text. The transcripts for f) were selected from the Milo analysis. Full list of BEST4+ small vs large intestinal transcripts are shown in the Supplementary Fig. 3c.
2.2	2. In reviewing the fascinating complexity of the enteroendocrine cells, I wonder to what extent the very early fetal timepoints might reflect contamination from the pancreatic bud (acknowledged for insulin-producing beta, endocrine cells in line 148-9) for exocrine cell subsets as well. In very early development, not sure how easy to dissect apart these organs.	Beta cells were observed only in the first trimester proximal ileum samples (BRC2258-26 cells, BRC2043 - 6 cells and BRC2199- 1 cell) (Table 2; Supplementary Fig. 4c), while differentiated endocrine cells were observed across second-trimester and pediatric/adult samples. In the first trimester samples we also observe a population of CLDN10+ cells that could be an early pancreatic exocrine cell subset as it expresses PDX1. The majority of CLDN10+ cells were observed in BRC2258 and BRC2043 samples, coinciding with a higher number of beta cells found in these samples. We were not able to dissect pancreatic bud in the first trimester samples due to the size of the bud at that time, low availability of human samples and the fragile nature of fetal tissues (manipulation of tissues risks damaging the proximal ileal region). However, we decided not to remove these cells from our analysis and the dataset as we believe it opens avenues for further integration of this data with other data sets (e.g. adult pancreatic datasets for better resolution of pancreatic development in humans). Instead, our solution is now in the revised

manuscript to **clearly outline pancreatic populations with a dashed line in the Figure 2 a and e** and explain it in the figure legend:

“Cells in grey outlined with a black dashed line are likely developing pancreatic bud contamination.”

Figure 2: FCGR2A signalling in tuft cells; a-b) UMAP visualisation of a) fetal and b) postnatal (pediatric and adult) epithelial lineage from scRNA-seq data coloured by cell type. Key cell types discussed in text are circled with a dashed line and arrows depict scVelo paths of differentiation towards secretory and absorptive enterocytes. c) Relative proportions of cell subtypes within total epithelial lineage as in (a-b) separated by donor age (row). Units of age is years unless specified as weeks (Wk). d) Dot plot of *TMPRSS2* and *ACE2* expression by epithelial cell types in the fetal intestine as in (a-b). Here and in later figures, color represents maximum-normalized mean expression of marker genes in each cell group, and size indicates the proportion of cells expressing marker genes. e) UMAP visualisation of enteroendocrine cells (EECs) from fetal, pediatric and adult scRNA-seq datasets, where arrows depict differentiation paths inferred from scVelo analysis. **Cells in grey outlined with a black dashed line are likely developing pancreatic bud contamination.** f) UMAP visualisation as in (e)

showing key genes of NPW+ enterochromaffin (EC) cells. g) Heatmap of genes that change along the differentiation trajectory from NEUROG3+ progenitors to EC cells as highlighted by red arrow in (e). Arrows indicate known gene associations with EC differentiation in mice and organoids. hi) Dotplot of expression of key molecules upstream and downstream of PLCG2 activation in tuft cells and pooled absorptive (TA and enterocytes) and secretory (paneth, goblet and enteroendocrine (EECs)) epithelial cells as in (a-b). h) Percent expression of FCy receptor by SiglecF+EpCAM+ and SiglecF-EpCAM+ cells in wild type mice determined by flow cytometry (biological replicates are presented by individual points; n=4). i) Summary schematic of proposed signaling pathways in tuft cells. P values were calculated using 2-way ANOVA with Sidak multiple comparisons. ****<0.0001.

2.3

3. 1d. It seems rather unlikely that in the adult samples the fractions of epithelial cells range from < 2% (duodenum, cecum, rectum) to well over 15% (Sigmoid). I think the general concept that transmural vs. biopsy samples results in lower fractions of epithelial cells seems right, however, as presented, I feel that this bar-graph is misleading especially for the within adult comparisons. Is it possible that differences in sampling capture efficiencies account for this? The doublet removals (especially problematic for epithelial cells and given the high fraction of cell removal generally)?

Figure 1d shows the fraction of BEST4+ cells amongst all epithelial cells between the anatomical regions. Sample capture may contribute to the difference in the proportion of these cells between regions, however we believe this would be minimal since all epithelial cell types would incur the same bias in capture. We also do not observe a large technical variability in BEST4+ cells (Figure R4). We suggest that this difference is a true reflection of the makeup of the epithelial layer between tissue regions.

Figure R4: Dotplot where the fold change represents the enrichment (or depletion, low

		fold change in blue) of epithelial cell subsets compared with baseline. The local true sign rate (LTSR) value represents statistical significance of the fold change estimate ranging from 0 to 1, where 1 represents a confident estimate. BEST4+ enterocytes and colonocytes are highlighted in red.
2.4	4. (MInor) 1d. Consider removing the color (or restricting to epithelial green), so as to not confuse with 1b/c	Thank you, we have changed the color of the 1d barplot to epithelial green. Please see comment 2.1, Figure 1.
2.5	5. 1f. Legend doesn't specify what the red highlights designate. Not sure what the basis for transcript inclusion is for this gene list	Thank you, we have addressed this in text: “ f) Dotplot with relative expression of selected marker genes within BEST4+ epithelial cells from small and large intestines and different ages as in (d). Genes highlighted in red are discussed in text. The transcripts for f) were selected from the Milo analysis. Full list of BEST4+ small vs large intestinal transcripts are shown in the Supplementary Fig. 3c. “
2.6	2. Figure 2. The RNA velocity analyses substantially strengthen the opening to this Figure. Figure 2a (fetal) vs. b (ped/adult). I may be mis/over-interpreting the scVelo analyses, but how do the authors interpret the M cell differences between pre- vs. post-natal cells? While the post-natal analyses appear to make sense (M cells from stem cells), the M cells in fetal appear far from the stem cells. This would appear to be in conflict with de Lau et al's findings (PMID 22778137). Does this reflect pre- vs. post-natal differences?	scVelo analysis is influenced by the position of the cells within the UMAP space and it is possible that the differences seen in pre- vs. post-natal M cells are due to cell type numbers and composition differences at these timepoints. Being close in the UMAP space also does not necessarily reflect differentiation relationships. To add to the scVelo analysis, we have performed analysis to resolve potential M cell position along the crypt-villus axis. For this, we use curated genes that are differentially expressed along the crypt-villus axis defined by Moor et al. (2018) and Parikh et al. (2019) (SEPP1, CEACAM7, PLAC8, CEACAM1, TSPAN1, CEACAM5, CEACAM6, IFI27, DHRS9, KRT20, RHOC, CD177, PKIB, HPGD, LYPD8, APOBEC1, APOB, APOA4, APOA1, NPC1L1, EGFR, KLF4, ENPP3, NT5E, SLC28A2, ADA). We restrict this analysis to the small intestinal epithelial cells only (Figure R5).

Figure R5: Violin plots with crypt-villus axis score in fetal (top) and adult (bottom) small intestinal epithelial subsets.

Pre- and post-natal M cells scored similarly for the crypt-villus axis score (0.4-0.5), while we observed differences in other secretory cells (Goblet cells, Enteroendocrine) that scored lower in fetal samples, possibly reflecting their immaturity.

Based on this analysis, we believe that the difference between pre- and post- natal scVelo results reflect the differences in composition (low number of M cells in the fetus) rather than alternative differentiation paths.

2.7 3. ILC origins. This may be beyond the scope of this manuscript, given the small number of ILC2 cells identified,

Thank you for this interesting suggestion. Unfortunately, this is beyond the scope of the current manuscript due to the low number of ILC2 cells in our data.

	but does appear generally relevant given the tie-in to IBD in Fig 5. Bielecki et al., (PMID 33536623) show in skin that tissue resident innate lymphoid cells are sufficient to drive inflammation/pathogenicity, implying conversion from ILC2 to ILC3. Do any of the present data support this model (small numbers of ILC2 due to transition toward ILC3), or do the authors think this merely reflects organ differences between skin and gut?	
2.8	4. Data Sharing and Supplementary Materials. The authors have provided exhaustive figures and tables for the community, which will be incredibly valuable for the community at large.	Thank you.
2.9	Minor 1. Consistency wrt number of different anatomic sites sampled: 10 (line 51) vs. 11 (line 84). Figure 1a designates even more locations. Should be consistent or qualify in place the number of sites.	Thank you. We have now clarified that in the adult donors we have sampled 11 distinct locations, which are duodenum (two locations DUO1 and DUO2 that were pooled in the analysis), jejunum (JEJ), ileum (two locations ILE1 and ILE2 that were pooled in the analysis), appendix (APD), caecum (CAE), ascending colon (ACL), transverse colon (TCL), descending colon (DCL), sigmoid colon (SCL), rectum (REC), and mesenteric lymph nodes (MLN). We clarify this in methods and main text as follows: “To comprehensively map cell lineages we used single-cell RNA sequencing (scRNA-seq) and V(D)J analysis of almost half a million cells from up to 5 anatomical regions in the developing and up to eleven distinct anatomical regions in healthy pediatric and adult human gut. “ “Here, we create a single-cell census of the healthy human gut, encompassing around 428,000 cells from small and large intestines as

		well as associated lymph nodes during in utero development, childhood and adulthood.” Methods: “Samples were collected from 11 distinct locations, including duodenum (two locations DUO1 and DUO2 that were pooled in the analysis), jejunum (JEJ), ileum (two locations ILE1, ILE2 that were pooled in the analysis), appendix (APD), caecum (CAE), ascending colon (ACL), transverse colon (TCL), descending colon (DCL), sigmoid colon (SCL), rectum (REC), and mesenteric lymph nodes (mLN). “
2.10	2. Line 457: change Nerog3+ to Neurog3+	Changed, thank you.
2.11	3. Line 260: Change parentheses to (beta chain of gut-specific a4b7 integrins)	Changed, thank you.

Reviewer Reports on the Second Revision:

Referee #3 (Remarks to the Author):

All concerns have been dealt with satisfactory and I just found a few remaining typos and small errors that needs attention.

Figure 1B. Please change “Neuronal” to “Neural” as you show developing glia and neurons.

Figure S2d. Please change “Neuronal” to “Neural” as you show developing glia and neurons.

Figure 3a. Both *Onecut2* and *3* appear rather specific to Branch A3, You could mention both (*Onecut2/3*)

Table 1 Morarach et al corrections:

i) Developing (E15.5 and E18.5) and **juvenile (P21)** ENS from mouse small intestine.

ii) For E15.5 and E18.5, *Wnt1-Cre x R26Tom* mice were used (to capture whole ENS including all stages of differentiation), while juvenile datasets were produced from *Baf53b-Cre* to enrich for neurons only.

iii) Clarify the new model: “Define a molecular taxonomy of 12 enteric neuron classes within the myenteric plexus of the adult mouse small intestine. Embryonic ENS uncovers a novel principle of neuronal diversification, where two neuron classes arise through a binary neurogenic branching (Branch A and B) **and all other classes via postmitotic differentiation.**”

Table 1 Current study corrections:

i) Neuronal – Change to “Neural”

ii) Comparable clusters observed in Fawkner-Corbett et al., *Cell*, 2021. However, the annotations are based on the mouse data focusing on **gliogenesis** and neurogenesis branching (Morarach et al., *Nature Neuroscience*, 2021).

Author Rebuttals to Second Revision:

Referee #3:

Remarks to the Author:

All concerns have been dealt with satisfactory and I just found a few remaining typos and small errors that needs attention.

1.	Figure 1B. Please change “Neuronal” to “Neural” as you show developing glia and neurons.	Thank you, changed.
2.	Figure S2d. Please change “Neuronal” to “Neural” as you show developing glia and neurons.	Thank you, changed.
3.	Figure 3a. Both Onecut2 and 3 appear rather specific to Branch A3, You could mention both (Onecut2/3)	Thank you, updated with both genes.
4.	Table 1 Morarach et al corrections: i) Developing (E15.5 and E18.5) and juvenile (P21) ENS from mouse small intestine.	Thank you, the table is now updated.
5.	ii) For E15.5 and E18.5, Wnt1-Cre x R26Tom mice were used (to capture whole ENS including all stages of differentiation), while juvenile datasets were produced from Baf53b-Cre to enrich for neurons only.	Thank you the table is now updated.
6.	iii) Clarify the new model: “Define a molecular taxonomy of 12 enteric neuron classes within the myenteric plexus of the adult mouse small intestine. Embryonic ENS uncovers a novel principle of neuronal diversification, where two neuron classes arise through a binary neurogenic branching (Branch A and B) and all other classes via postmitotic differentiation. ”	Thank you, it has been updated according to reviewer’s suggestion.
7.	Table 1 Current study corrections: i) Neuronal – Change to “Neural”	Thank you, changed.
8.	ii) Comparable clusters observed in Fawkner-Corbett et al., Cell , 2021. However, the annotations are based on the mouse data focusing on gliogenesis and neurogenesis branching (Morarach et al., Nature Neuroscience , 2021).	This has been updated, thank you.